

# AGRIDE-c, a conceptual model for the estimation of flood damage to crops: development and implementation

Daniela Molinari[1], Anna Rita Scorzini[2], Alice Gallazzi[1], Francesco Ballio[1]

[1] Department of Civil and Environmental Engineering, Politecnico di Milano, Piazza Leonardo da Vinci 32, 20133, Milano, Italy
[2] Department of Civil, Environmental and Architectural Engineering, Università degli Studi dell'Aquila, Via Gronchi, 18, 67100, L'Aquila, Italy

*Correspondence to*: Daniela Molinari (daniela.molinari@polimi.it)

**Abstract.** This paper presents AGRIDE-c, a conceptual model for the assessment of flood damage to crops. All available knowledge on damage mechanisms triggered by inundation phenomena is systematised in a usable and consistent tool, with the main strength represented by the integration of physical damage assessment with the evaluation of its economic consequences on farmers' gross product. This allows AGRIDE-c to be used to guide the flood damage assessment process in different geographical and economic contexts, as demonstrated by the example provided in this study for the Po Plain (North of Italy). The development and implementation of the model highlighted that a thorough understanding and modelling of damage mechanisms to crops allows for comprehensive cost-benefit analyses of risk mitigation actions, and is a powerful tool to orient farmers' behaviour towards more resilient damage alleviation practices.

## 1 Introduction

The implementation of the European Floods Directive (Directive 2007/60/EC) requires Member States (and, in particular, River Basin Districts) to define and select mitigation actions to be included in Flood Risk Management Plans, on the basis of reliable and inclusive cost-benefit analyses (CBAs). The latter should consider all the direct and indirect costs and benefits that a specific measure brings to the society (Jonkman et al., 2004; Mechler, 2016), with benefits consisting in the avoided damage with respect to a null action.

In this framework, this paper deals with the estimation of flood damage to the agricultural sector. Indeed, the inclusion of damage to agriculture in CBAs is critically needed, especially when risk mitigation measures involve floodplains devoted to agricultural activities: this is typically the case of river restoration actions, as usually included in "integrated river basin management" projects (Morris and Hess, 1988; Brémond et al., 2013; Massaruto and De Carli, 2014; Guida et al., 2016). Moreover, a thorough understanding of flood damage mechanisms may increase farmers' resilience to floods, by supporting them in identifying the most proper damage alleviation and coping strategies.

In the literature on flood damage modelling, agriculture has received so far less attention than other exposed sectors, as demonstrated in Table 1, showing the number of papers in the Scopus database for different research keywords. Reasons may





include: (i) the (perceived) minor importance of agricultural losses compared to that of other sectors, especially because flood damage assessments are usually carried out in urban areas (Förster et al. 2008), (ii) the paucity of data for understanding damage mechanisms to agriculture and deriving prediction models and, finally, (iii) the insurance coverage for damage to farms, strongly incentivised at national level in many countries since the late nineties, that led most of public authorities

responsible for damage compensation to be less interested in the agricultural sector. Nonetheless, available damage models for agriculture are not only few in number but are also affected by many limitations, the major related to their local characteristics (i.e. the strong linkage with the context under investigation) and limited transferability to different contexts as wells as the lack of information/data for their validation. Accordingly, the first requirement for new models is their possible application in a wide variety of geographical and economic contexts. Experience gained in flood damage assessment for other sectors (typically

residences) highlighted that a broad generalisation is often not possible, as damage models must be able to capture the specificities of the investigated area, both in terms of hazard and vulnerability features (Cammerer et al., 2013). Still, a general conceptualisation of the problem at stake is conceivable in terms of main variables influencing the damage mechanisms, cause-effect relationships, etc.

Based on these considerations, this paper presents AGRIDE-c (AGRiculture DamagE model for Crops), a conceptual model

for the estimation of flood damage to crops. AGRIDE-c has the ambition of generality, i.e. to be valid in different geographical and economic contexts, supplying a useful framework to be followed any time the estimation of flood damage to crops is required, in which the main components of the problem at stake are identified as well as its relevant control parameters. While the model structure is generally valid, the analytical expression of its components is necessarily specific to the physical characteristics of the area as well as to the standards of the agricultural practices and to the type of crops under analysis. The

implementation of the conceptual framework of AGRIDE-c is exemplified in this paper in relation to the Po Plain - North of Italy. The case study is completed with a spreadsheet (available at: http://www.floodimpatproject.polimi.it/?page_id=652) for the calculation of damage to crops, which can be adapted to other contexts.

The paper is organised in four parts. Section 2 reviews the state of art on flood damage modelling to crops, as the starting point of the research. Section 3 presents the AGRIDE-c model, while Section 4 describes in detail its implementation in the Po Plain.

Section 5 provides a critical discussion on limits and strengths for the effective application of AGRIDE-c and conclusions are finally drawn in Section 6.

**Table 1. Papers in the Scopus database for different research keywords (last access: January 2019)**

| Keyword search | Number of papers |
| --- | --- |
| "Flood damage" | 4036 |
| "Flood damage" AND "crop" | 81 |
| "Flood damage" AND "agriculture" | 71 |
| "Flood damage" AND "building" | 284 |
| "Flood damage" AND "infrastructure" | 122 |



## 2 State of art on flood damage modelling for crops

The main available damage models for crops are reported in Table 2. As an overall consideration arising from the analysis of the literature, it can be first underlined that assumptions at the base of many models are not adequately described, leaving uncertainties in the interpretation of the approach and possibly leading to incorrect implementation of the procedure for damage

assessment.

The analysis of Table 2 shows that main differences among models are related to the input variables describing the inundation scenario (hazard) as well as the response of the exposed elements to flooding (vulnerability). Beyond hazard parameters usually considered in damage modelling for other exposed sectors (i.e., water depth, flow velocity, flood duration, sediment and contaminant load), for crops a key role is played by the period of the year, generally the month, of the flood event, as damage

is strongly dependent on crop calendars (USACE, 1985; Morris and Hess, 1988; Hussain, 1996; RAM, 2000; Citeau, 2003; Dutta et al., 2003; Förster et al., 2008; Agenais et al., 2013; Shrestha et al., 2013; Vozinaki et al., 2015; Klaus et al., 2016) that, in their turn, depend on the climate of a region: this is one of the reasons which makes damage models for crops strongly context specific. Indeed, crop calendars delineate the vegetative stage of the plants at the time of the flood (which strongly affects the damage suffered by the plants) for any crop type, the latter being the only vulnerability parameter often considered

by the models. In the case of meso-scale models (Kok et al., 2005; Hoes and Schuurmans, 2006), this parameter is replaced by the agricultural land-use. No model in Table 2 considers instead the behaviour of farmers after the occurrence of the flood (e.g. the decision of abandoning the production or to continue with increasing production costs) which has been shown to strongly influence the damage sustained by the farm (Pangapanga et al., 2012; Morris and Brewin, 2014).

With respect to the approach, only few literature models are directly derived from field observations of flood consequences on

crops: this is mainly due to the scarcity of ex-post damage data (Brémond et al., 2013) for models derivation/calibration. In fact, most of the models adopt a synthetic approach based on the expert investigation of causes and consequences of damage. In this regard, some models in Table 2 are labelled as "physically based", i.e., damage is first described in terms of physical susceptibility of the crop and consequent yield reduction, and then converted into economic impact on farmer's income. Instead, in "cost based" models damage is assessed considering only production costs sustained by farmers during the year, by

implicitly assuming (according to our interpretation) that the yield is totally lost in case of flood. Whatever the adopted approach, a comprehensive model for damage to crops should consider the (inter)correlation between the two aspects: actual yield reduction, as a function of hazard and vulnerability variables, and saved/increased production costs due to the occurrence of the flood (Pivot and Martin, 2002; Posthumus et al., 2009; Morris and Brewin, 2014).

With respect to the monetary evaluation, damage can be expressed as percentage of the gross profit (USACE, 1985; RAM,

2000; Agenais et al., 2013; Shrestha et al., 2013) or of the turnover (Citeau, 2003; Dutta et al., 2003; Förster et al., 2008; Vozinaki et al., 2015; Klaus et al., 2016) for the farmer. From another point of view, some models express damage in absolute terms (thus depending on local prices and costs) while others in relative terms, as a percentage of a maximum exposed value.





| Study and country | Crop types | Hazard parameters | Vulnerability aspects | Modelling approach | | Monetary evaluation approach | Validation |
|---|---|---|---|---|---|---|---|
| | | | | Empirical vs. expert based | Cost vs. physically based | | |
| AGDAM/ Hazus (USACE 1985) - USA | Generic crop | Duration, time of occurrence (month) | Crop type | Not specified | Cost based (supposed) | Relative - Damage as a percentage of the gross profit | Not specified |
| Morris and Hess (1988) - UK | Grassland | Time of occurrence (expressed in terms of vegetative stage) | Vegetative stage | Expert based | Physically based (i.e. damage functions give yield reduction due to the flood + information on additional/saved costs) | Absolute | No |
| Hussain (1996) - Bangladesh | Rice | Water depth, duration, sediment concentration, time of occurrence (growing stage) | Vegetative stage | Expert based | Physically based (i.e. damage functions supply yield reduction because of the flood) | Relative - No monetary evaluation | No |
| RAM (Read Sturgess and Associates (2000)) - Australia | Grassland, generic crop | Duration, time of occurrence (month) | Crop type | Expert based | Cost based | Absolute - Damage as a percentage of the gross profit | Not specified |
| Citeau (2003) - France | Maize | Water depth, duration, velocity, time of occurrence (month) | Crop type | Expert based | Cost based (supposed) | Relative - Damage as a percentage of the turnover | No |
| Dutta et al. (2003) - Japan | Beans, Chinese cabbage, dry crops, melon, paddy, vegetable with roots, sweet potato, green leave vegetables | Water depth, duration, time of occurrence (month) | Crop type | Empirical | Not specified; in fact, the model can be adapted to both a cost based and a physically based approach by varying the loss factor related to the time of the year | Relative - Damage as a percentage of the turnover | No |





| Standard method (Kok et al. (2005)) - The Netherlands | Generic agricultural land | Water depth | Agricultural land use | Expert based | Not specified | Relative - Not specified | Not specified |
|---|---|---|---|---|---|---|---|
| Hoes and Schuurmans (2006) - The Netherlands | Maize, orchards, cereals, sugar beet, potatoes, other crops, | Water depth | Agricultural land use | Not specified | Not specified | Relative -Not specified | No |
| Förster et al. (2008), Klaus et al. (2016) - Germany | Grain crops (wheat, rye, barley, corn), oilseed plants (canola), root crops (potatoes and sugar beets) and grassland | Duration, time of occurrence (month) | Crop type | mixed (empirically-expert based) | Cost based (supposed) | Relative - Damage as a percentage of the turnover | Yes, for one flood event |
| Agenais et al. (2013) - France | Wheat, barley, canola, sunflower, maize, vegetables, grassland, alfalfa | Water depth, duration, time of occurrence (week) | Crop type, vegetative stage | expert based | Physically based (i.e. damage functions give yield reduction due to the flood + information is supplied on additional/saved cultivation costs) | Relative - Damage as reduction of the gross profit | No |
| Shrestha et al. (2013) – Mekong Basin | Rice | Water depth, duration, time of occurrence (expressed in terms of vegetative stage) | Vegetative stage | Not specified | Not specified | Relative - Damage as reduction of the turnover | Yes (partial) |
| Vozinaki et al. (2015) - Greece | tomatoes, green vegetables | Water depth, flow velocity, time of occurrence (month) | Crop type, vegetative stage | Expert based | Physically based (i.e. damage functions supply yield reduction due to the flood) | Relative - Damage as a percentage of the turnover | No |

**Table 2. Analysis of state-of-art flood damage models for crops**




 Finally, last column of Table 2 indicates that damage models for the agricultural sector are hardly validated, mainly due to the scarcity of ex-post damage data discussed before; a partial exception is represented by the models by Förster et al. (2008) and Shrestha et al. (2013).

Overall, the state of art depicts a fragmented scenario, characterised by the existence of few, case-specific and poorly
documented models, only partly capturing the available knowledge on flood damage to crops, due to several simplifying assumptions. In this context, the use of existing models for the assessment of flood damage outside the contexts for which they were proposed is not a feasible option. Indeed, limited information on the rationale behind model development, like for instance on the adopted approach (whether empirical or synthetic, and, in the second case, whether physically or cost based), on the components of the model (in terms, e.g., of included cost items, modelled physical processes), and on the characteristics of the
region for which the model was derived (in terms of crop calendars, standard agricultural practices, etc.) prevents the identification of those models that may be suitable to be applied in a given study area. Nonetheless, it is not possible to implement existing models as "black box" models" (for example, for a preliminary estimation of damage) due to the lack of ex-post damage data for their validation.

In order to exemplify possible problems arising in the application of existing models, we tested the approaches proposed by
Agenais et al. (2013) and Förster et al. (2008) to estimate the relative damage to a 1 ha maize plot. The implementation was quite straightforward as both models supply damage in relative terms. Although the models are theoretically comparable, as they refer to similar contexts (France and Germany), sharing both climate characteristics and crop calendars (for maize, seeding in April and harvest in September/October), they produced significantly different results, as reported in Figure 1, where the models are applied for three different values of the water depth and two different flood durations.

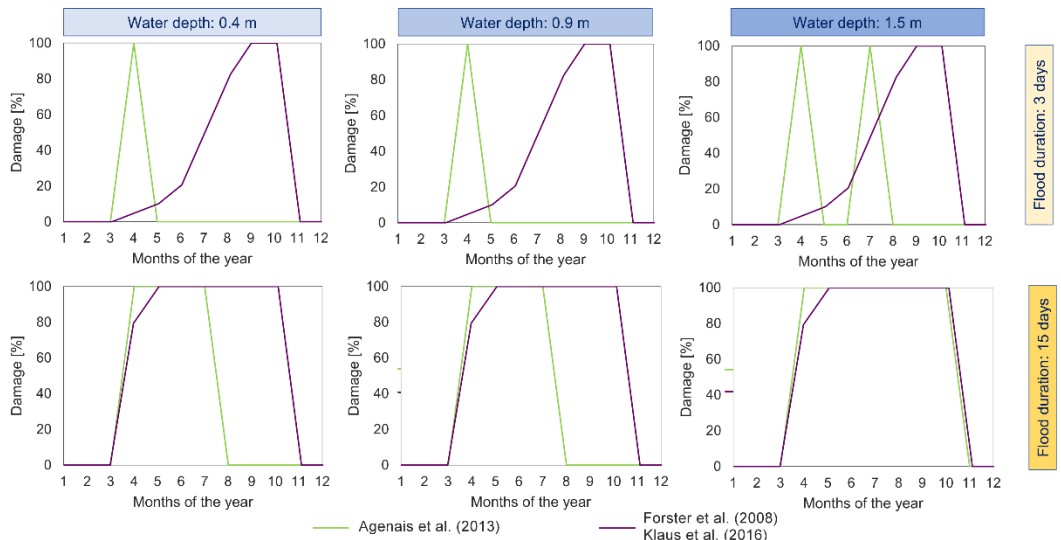

**Figure 1. Comparison between relative damage supplied by Agenais et al. (2013) and Forster et al. (2008) for a 1 ha maize plot, for two values of flood durations and three values of water depth**



For example, for short duration floods (d=3 days), Agenais et al. estimate the maximum damage in April for shallow water depths with a further peak of damage in July for higher water depths, while Förster et al. estimate the maximum damage in September-October, whatever is the value of the water depth.

The main reason for this inconsistency lays in the different modelling approach adopted by the two models: physically-based in the case of Agenais et al. and cost-based in the case of Förster et al.. Coherently, Agenais et al. estimate the maximum damage in correspondence of the most fragile vegetative phases of the crop, i.e. the growth (March-May) and the flowering (June-August), while Förster et al. well reproduce increasing costs sustained by farmers during the vegetative cycle, resulting in maximum damage at the harvesting phase (September-October). A further source of inconsistency among the two models is related to the different set of input variables, as Agenais et al. consider water depth as a control parameter, while Föster et al. do not, thus leading to different damage estimation even for a given flood duration. At last, a further source of error may be represented by the conversion from relative to absolute damage; indeed, while the relative model by Agenais et al. is derived by referring to the gross profit, the relative model by Förster et al. refers to the turnover. Given that conventions do not exist on how translating relative damage into absolute terms, the choice of the wrong reference value could amplify inconsistency between the two approaches.

In view of the above considerations, there is a need to organise available knowledge on flood damage mechanisms in a comprehensive and general framework that can be adapted to any context, by taking into account the specificities of the area under investigation. This was the main reason which led us to develop the AGRIDE-c model, described in detail in the next section.

## 3 Conceptual model of AGRIDE-c

AGRIDE-c has been developed by adopting an expert-based approach, encapsulating and systematising all the available knowledge on damage mechanisms triggered by inundation phenomena, as well as on their consequences in terms of income for the farmers. Information has been derived by a thorough investigation of the literature (Section 2) and by consultation with experts (i.e. agronomists and representatives of the authorities responsible for agricultural damage management and compensation). The result is a general, conceptual model, which identifies the different aspects to be modelled for the assessment of flood damage to crops, their (inter)connections as well as the variables at stake. Still, as stressed before, the implementation of the model (that is the derivation of an analytical expression for each of its components) must be context specific, as damage to crops depends on many local features that cannot be generalised. An example of the implementation of the model is supplied in Section 4.

The structure of AGRIDE-c is depicted in detail in Figure 2. Absolute damage (D) for an individual farmer is expressed as the difference between the reduction in the turnover ($\Delta T$) and the increase/decrease in production costs ($\Delta PC$), as a consequence of the flood of a specific crop. This is equal to consider absolute damage as the change in the gross profit (GP = T– PC, where T= turnover and PC = production costs) due to the flood, compared to the case when no flood occurs (i.e., Scenario 0):





**Figure 2. Conceptual model of AGRIDE-c**





$$D = GP_{noflood} - GP_{flood} = (T_{noflood} - T_{flood}) - (PC_{noflood} - PC_{flood}) = \Delta T - \Delta PC \qquad (1)$$

Accordingly, relative damage (d) can be obtained by dividing the absolute damage by the gross profit in the Scenario 0
($GP_{noflood}$)

$$d = D/GP_{noflood} \qquad (2)$$

AGRIDE-c includes a physical and an economic model to evaluate the absolute damage; the first provides information on the
physical damage, while the second converts the physical effects of the flood into monetary terms. In this way, the problems of
consistency among physically-based and cost-based models discussed in Section 2 are overcome, being both aspects taken into
account.

Physical damage to crops depends, on the one hand, on the direct contact of the flooding water with the plants; on the other
hand, damage to other components/elements of the farm may induce additional damage to crops, as, for instance, damage to
soil that may imply a reduction in soil fertility, and damage to machineries and equipment (e.g. the irrigation plant), that may
prevent cultivation for a while. Among these, AGRIDE-c considers only the damage to soil. This choice derives from the
evidence that, during a flood, damage to soil and plants occurs always at the same time differently from damage to the other
components which can occur or not, independently from the damage to plants; as a consequence, damage to soil and plants is
modelled together, while damage to the other components can be modelled as separated factors. The physical model of
AGRIDE-c (identified by the yellow dashed box in Figure 2) is therefore composed of two sub-models, for the evaluation of
physical damage to crops (i.e. the plants) and to soil, respectively.
The model for the assessment of the physical damage to crops calculates the reduction in the amount and quality of the harvest
due to the flood, as a function of the features of the flood (i.e. time of occurrence and intensity) and of the type of affected
crop. Indeed, the occurrence and the severity of damage mechanisms leading to yield decline (like root asphyxiation,
contamination, development of diseases and parasites) mainly depend on flood intensity, i.e. water depth, water velocity, flood
duration, sediment and contaminants load; still, different crops withstand flood impacts in different ways according to their
physical features as well as their vegetative stage at the time of occurrence of the flood (Rao and Li, 2003; Setter and Waters,
2003; Zaidi et al., 2004; Araki et al., 2012; Ren et al., 2016).
The model for the assessment of physical damage to soil calculates instead the amount of soil that is damaged and the kind(s)
of damage suffered by the soil, i.e. erosion, deposition of sediments, contamination (on which costs for soil restoration depend),
as well as the consequent reduction in soil fertility (which affects the quality and the quantity of the harvest), as a function of
the duration of the flood, the water velocity, the sediment and the contaminants loads.
The economic model of AGRIDE-c (identified by the green dashed box in Figure 2) consists of two sub-models as well: one
for the evaluation of the reduction in the turnover and one for the assessment of the increase/decrease in production costs,
compared to the Scenario 0 (i.e. no flood). The first model calculates $\Delta T$ as the reduction in the turnover due to a reduced yield





and to a decrease in the price of the crops because of a lower quality harvest; the second model evaluates ΔPC as the additional costs required to restore the flooded soil and to carry out additional cultivation practices for continuing the production (typically, reseeding), as well as saved costs in the case of abandoning. Indeed, farmers can react in different ways to alleviate flood damage, according to the vegetative stage of the plant at the occurrence of the flood, and of the physical damage suffered

by the plant. The first possible strategy is continuing when flood damage implies none or minor yield loss. The second strategy is reseeding a new (late) crop; this strategy is possible only in certain periods of the year according to the vegetative cycle of the crop under investigation. Finally, when the yield loss is severe, farmers can decide to abandon the production. ΔPC strongly depends on the strategy adopted by the farmer as well as the actual yield loss. For example, after an event causing a physical loss corresponding to 50% of the expected yield, a farmer can decide to continue the production or to abandon it; in the first

case, the yield reduction will be just 50% of the expected yield, while the farmer must sustain all the costs which are still necessary to conclude the vegetative cycle; the second case will result instead in a total crop loss (100%) and in the saving of part of the production costs.

## 4 Implementation of the model for the Po Plain

As previously discussed, while the conceptual structure of AGRIDE-c has a general validity for different geographical and

economic contexts, the analytical expression of its sub-models must be context specific. In this section, we provide an example of implementation for the Po Plain - North of Italy which can serve as guidance for the definition of the sub-models of AGRIDE-c in any other region.

The first step for the development of the model in a given area consists in the identification of the typical features of flood events occurring in the area as well as the main cultivated crops. The second step consists in the calculation of the gross profit

for the farmer in the Scenario 0, by considering the amount of production (yield), selling prices of the crops, time and costs of cultivation practices in the absence of any flood. Third, analytical expressions for all the processes shown in Figure 2 are derived and then, starting from the Scenario 0, flood effects on crops (i.e. the damage) are evaluated for different times of occurrence, flood intensities, and damage alleviation strategies.

### 4.1 Hazard and vulnerability features in the Po Plain

In order to identify the representative features of the floods and the main crops cultivated in the investigated area, we chose the Province of Lodi (Lombardia Region) as representative of hazard phenomena and agricultural activities in the Po Plain. As regards the hazard, the investigation of the last significant event occurred in the province, i.e. the flood of the Adda River in November 2002 (AdBPo, 2003; AdBPo, 2004; Rossetti et al., 2010; Scorzini et al,, 2018), highlighted riverine long-lasting floods, characterised by medium to high water depths (mean value: 0.9 m), low flow velocities (mean value: 0.2 m/s) and low

sediment and pollution loads in the flooded areas as typical of the region; accordingly, main hazard parameters to be included





in the analytical expression of AGRIDE-c for the Po Plain are limited to water depth, flood duration and time (month) of flood occurrence.

On the other hand, the analysis of the agricultural cadastral data (supplied by the Regional Authority) in a buffer of 1 km around the Adda River, indicated grain maize, wheat, barley and grassland as the most common crops in the area; still, for the

5 sake of being concise, only the model for maize is discussed hereinafter, while those related to other crops are reported in the supplement.

## 4.2 Characterisation of the Scenario 0

The Scenario 0 is characterised in terms of the annual gross profit for the farmer, per hectare, in the case no flood occurs; this implies the estimation of the annual turnover and the distribution of production costs over the year.

Given that only one vegetative cycle of grain maize is possible in the Po Plain in one year, the turnover is estimated as the product between the average yield and price for grain maize over the last five years (data sources: Regione Lombardia and Borsa Granaria di Milano), equal to 175 q/ha and 16.92 €/q, respectively. In addition, we also consider the annual EU contributions for agriculture as a further income for the farmer and, in detail, the subsidies given to agricultural activities in case of the application of conservation tillage (i.e. minimum tillage) and crop rotation, equal respectively to 300 and 150 €/ha

(data source: PSR - Programma di Sviluppo Rurale, Regione Lombardia: http://www.psr.regione.lombardia.it).

Concerning production costs, the type, period of the year and costs of the different cultivation practices for grain maize are identified with the support of discussions with experts and consultation of regional price books. The results of the survey are summarised in Figure 3, which reports the distribution of costs over the year, with indication of the corresponding vegetative stages of the plant.

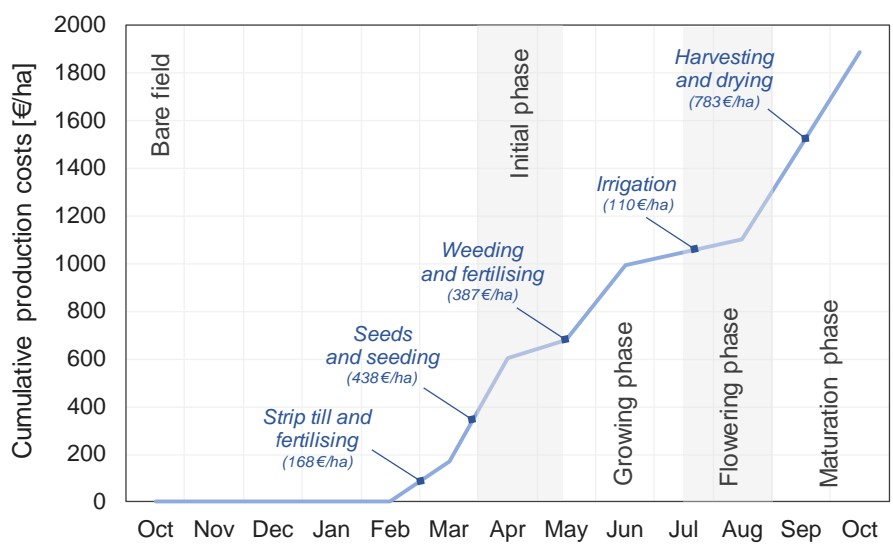

**Figure 3. Production costs over the year for grain maize, in case of minimum tillage technique**





Finally, fixed costs sustained by farmers (like management costs) are assumed to be a portion (5%) of the turnover. Based on these data, the analysis results in a gross profit for the famer in case of no flood equal to 1376 €/ha per year.

### 4.3 Damage to crops

Physical damage to crops is estimated by adopting the model developed in France by Agenais et al. (2013). This choice is supported by different considerations. First, the independent hazard variables considered by the authors (for maize: water depth and flood duration) are coherent with the typical flooding characteristics identified for the Po Plain (Section 4.1), i.e. riverine long-lasting floods. Second, their model can be easily transferred to other regions, independently from crop calendars, as they use the vegetative phases of the plant (and not the months of the year) as the time variable for the occurrence of the flood. Finally, local agronomists expressed a favourable opinion on the suitability of this model in the examined region.

An example of the physical damage model for maize is depicted in Figure 4 (adapted from Agenais et al., 2013). The model consists of susceptibility functions giving the yield reduction due to the flood (as a percentage of the yield in the Scenario 0), on the basis of water depth and flood duration, for four different vegetative stages (i.e. seeding, growing, flowering and maturation). Considering for example the growing stage, for a flood lasting less than 5 days the model gives a null yield loss, independently from the water depth; on the opposite, a flood lasting more than 12 days results in a total loss. For floods with intermediate duration, we assumed a linear yield reduction (from 0 to 100%) between 5 and 12 days. The use of this model implies that, at present, we do not take into account nor the reduction in the quality of the yield due to the flood nor the effect of damage to soil on yield quality and production; reason for such limitations is simply the lack of literature and data on these topics (see also Section 4.4).

### 4.4 Damage to soil

Concerning the physical damage to soil, no models were found in the literature investigating the complex chemical and mechanical processes leading to soil erosion, contamination and asphyxiation due to sediment deposition; we are, therefore, not able to parametrise the possible types of damage, the amount of damaged soil and the reduction in soil fertility as a function of hazard features. At present, the model is based on the assumption that soil always requires restoration in case of flood (consisting in the removal of sediments and in the levelling of terrain) and that no reduction in soil fertility occurs. Indeed, in the context under investigation, erosion and contamination are not expected because of the low velocity and contaminant load characterising typical floods in the region (see Section 4.2). According to regional price books, restoration costs have been estimated to be equal to 500 €/ha (see Table 3).

### 4.5 Alleviation strategies

After the recession of the flood, farmers make a choice among the possible strategies that can be adopted to alleviate damage; literature investigation and discussions with experts indicated three main strategies, their feasibility being necessarily linked to the damage suffered by the plants which, in its turn, depends on the flood intensity and the vegetative stage of the plants at





the occurrence of the flood: continuing the production, abandoning the production or reseeding. The strategy adopted by the farmer influences both yield reduction and production costs, because of cultivation practices which are additionally required or can be avoided when continuing or abandoning the production; such practices and related costs have been identified for the Po Plain, with the support of experts and regional price books (Table 3).

**Figure 4. Physical damage to maize as a function of vegetative stage, flood depth and duration (adapted from Agenais et al., 2013)**



**Table 3. Yield reduction and change in production costs for grain maize on the basis of damage alleviation strategy adopted by farmer**

| Time of the flood | Vegetative stage | Alleviation strategy | Yield reduction [%] | Additional costs | €/ha | Avoided costs | €/ha |
|---|---|---|---|---|---|---|---|
| November - March | Bare field | Continuation | 0 | Soil restoring (sediment removal and terrain levelling) | 500 | | |
| April - May | Initial phase | Abandoning | 100 | Soil restoring (sediment removal and terrain levelling) | 500 | Weeding and fertilising | 387 |
| | | | | | | Irrigation | 110 |
| | | | | | | Harvesting and drying | 783 |
| | | Reseeding | 0 | Soil restoring (sediment removal and terrain levelling) | 500 | | |
| | | | | Strip till and fertilising | 168 | | |
| | | | | Seeds and reseeding | 438 | | |
| June | Growing phase | Continuation | see Fig. 4 | Soil restoring (sediment removal and terrain levelling) | 500 | | |
| | | Abandoning | 100 | Soil restoring (sediment removal and terrain levelling) | 500 | Irrigation | 110 |
| | | | | | | Harvesting and drying | 783 |
| | | Reseeding | 0 | Soil restoring (sediment removal and terrain levelling) | 500 | | |
| | | | | Strip till and fertilising | 168 | | |
| | | | | Seeds and reseeding | 438 | | |
| July - August | Flowering phase | Continuation | see Fig. 4 | Soil restoring (sediment removal and terrain levelling) | 500 | | |
| | | Abandoning | 100 | Soil restoring (sediment removal and terrain levelling) | 500 | Irrigation | 55 |
| | | | | | | Harvesting and drying | 783 |
| September - October | Maturation phase | Continuation | see Fig. 4 | Soil restoring (sediment removal and terrain levelling) | 500 | | |
| | | Abandoning | 100 | Soil restoring (sediment removal and terrain levelling) | 500 | Harvesting and drying | 783 |

Continuing the flooded crops is suggested when flood damage implies none or minor yield loss; in this case, yield reduction

is equivalent to that supplied by the physical model of Figure 4 as a function of hazard features, while additional costs are only

due to soil restoring (see Section 4.4). Abandoning the production can be an option when flood damage is severe. This strategy

always leads to a 100% yield reduction; soil restoration is still required, but some production costs can be avoided according

to the time of the occurrence of the flood (i.e. remaining time to harvest). Reseeding is an alternative strategy to abandoning

when flood damage is severe, but it is possible only until June, by using late maize crops. At present, our model adopts the

simplified assumption that late reseeding does not imply a yield reduction, neither in quantity nor in quality. In fact, the use of

late crops generally implies a yield reduction with respect to traditional crops, reduction that increases as the time of reseeding

approaches the maturation phase, and that varies with the different species of late crops. Given the high variability of yield

loss with these two variables (i.e. time and species), a reference value was not identified in the literature neither in discussion

with experts; still, analysts can set the right value for the context under investigation in the available spreadsheet. In terms of

production costs, beyond additional costs required to restore the flooded soil, reseeding implies further additional costs related

to the preparation of the terrain, the purchase of new seeds and the seeding operations.



### 4.6 Damage estimation

According to the conceptual model in Section 3 and assumptions described in the previous sub-sections, damage (D) is estimated for different times of occurrence of the flood (i.e. month), flood intensities (i.e. water depth and flood duration) and damage alleviation strategies, as the difference between $\Delta T$ and $\Delta PC$:

$D = D$ (month, water depth, flood duration, alleviation strategy) $= \Delta T - \Delta PC$           (3)

In detail, $\Delta T$ and $\Delta PC$ are calculated on the basis of yield reduction and additional and avoided costs, as reported in Table 3 for maize. To be noted that, according to our model, absolute damage includes always costs related to soil restoration, which is required every time a flood occurs, irrespective of its intensity, the time of occurrence or the reaction of the farmer. All the other damage components depend, instead, on the four variables in Equation 3.

As an example of damage estimation, in Figure 5 we have reported changes in production costs and turnover for maize cultivation, for three different flood scenarios. More specifically, the value of the annual turnover and of cumulative production costs are reported for both Scenario 0 and the flood scenario under investigation, with respect to every alleviation strategy farmers can implement according to the intensity of the flood, its time of occurrence and the physical damage suffered by the plant. Differences of production costs and turnover between "flood" and "no flood" scenarios allow calculating the damage D

for the farmer.

The first scenario (Figure 5a) refers to a November flood. In this month, the plant is in the break stage, so no yield loss is expected for any flood intensity (Table 3). Farmers will then continue the production with additional costs limited to those required to restore the flooded soil for a total of 500 €/ha (Table 3), which is the absolute damage sustained by farmers.

The second scenario (Figure 5b) refers to a flood in June, when the plant is in the growing stage. According to the physical

model described in Figure 4, in this phase damage depends only on flood duration, while water depth has no effect on it. Figure 5b refers to a 5 days flood which leads, as given by the physical model, to a yield reduction of 12.5%. Given the low physical damage, farmers can decide to continue the production or to reseed. In the first case (green line), the turnover decreases by 12.5% (due to yield reduction), while production costs increase due to additional costs for soil restoration, resulting in an absolute damage for the farmer equal to about 870 €/ha. In the second case (blue line), no reduction in the turnover occurs

because reseeding would allow 100% of the yield, while additional production costs include both soil restoration and reseeding costs (Table 3), resulting in an absolute damage of 1106 €/ha. Figure 5b shows that, although possible in theory, abandoning the production is not a reasonable choice because of the low physical damage: in this case, in fact, absolute damage equals 2568 €/ha, due to a yield reduction of 100% (the only income for the farmer consists in the EU contributions for cultivation) against a saving of production costs of about 389 €/ha.

At last, Figure 5c refers to a flood occurring in September; in this period (i.e. maturation phase of the plant), damage depends on both water depth and flood duration. Figure 5c refers in particular to a 10 days flood with a water depth above 1.30 m. According to the physical model (Figure 4), this flood scenario leads to a 50% yield loss. Farmers have then two choices.




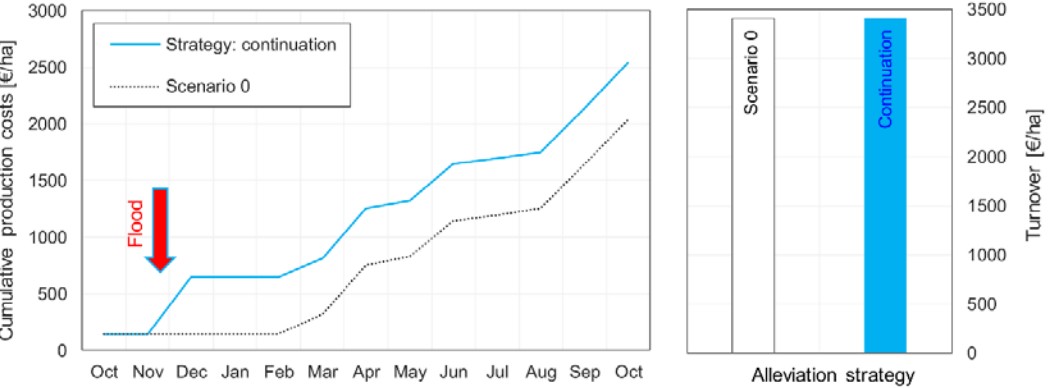

a)  November flood (break): any flood depth and duration

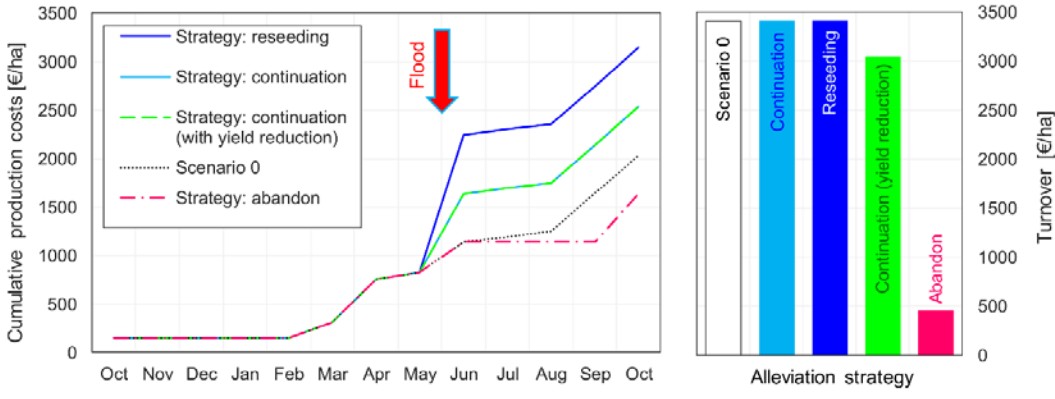

b)  June flood (growing): any flood depth and 5 days duration (yield loss 12.5%)

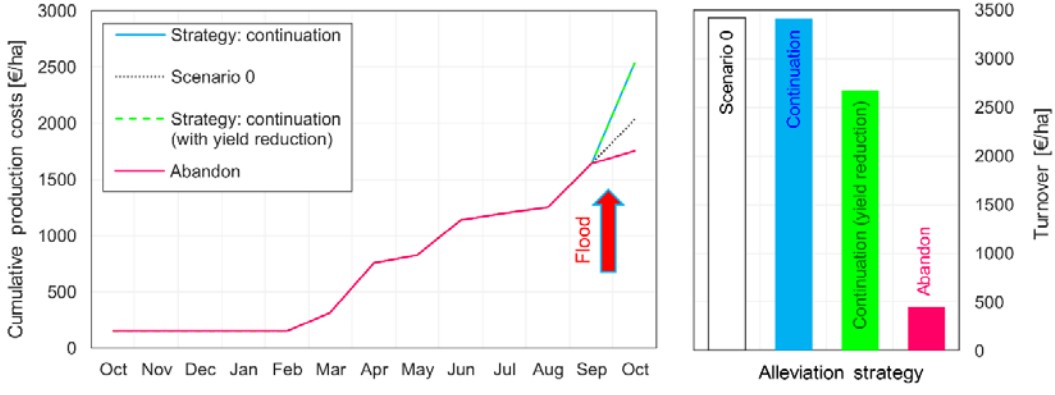

c)  September flood (maturation): flood depth > 1.30 m and 10 days duration (yield loss 50%)

**Figure 5. Distribution of cumulative production costs for grain maize during the year and annual turnover in the scenario 0 and in the case of a flood occurring in different months. Colours refers to the different possible strategies the farmer can adopt according to: the time of occurrence of the flood, intensity (water depth and duration) and physical damage.**





If production is continued the turnover decreases by 50% and additional costs are required to restore the flooded soil, resulting in an absolute damage equal to 1980 €/ha. In case of abandoning, absolute damage equals 2677 €/ha, because of a yield reduction of 100% and saving of production costs of 283 €/ha.

Previous considerations can be repeated for the different months of the year and hazard scenarios. Figure 6 displays the ensemble of the results of damage estimation for all the investigated cases, thus defining the AGRIDE-c model for the Po Plain, for grain maize crops. In particular, the figure reports the relative damage with respect to the gross profit in case of no inundation, $d=D/GP_{noflood}$, estimated by the model, for the different months of flood occurrence, flood intensities (i.e. water depth and flood duration) and damage alleviation strategies. The "dash" symbol means that the corresponding strategy cannot be adopted or is not reasonable in the flood scenario under investigation. For example, in the "bare field" season, reseeding is not possible because of climatic reasons, nor it is continuation as no cultivation is in place; continuation does not make sense when a 100% yield loss is expected as in the "initial phase" or in the "flowering" stage when h≥ 1.3 m (see Figure 4); reseeding with late crops is possible only until June, etc. Equivalent tables for the other investigated crops are reported in the Supplement.

## 5 Discussion

By enabling the estimation of the expected direct damage to crops in case of flood, AGRIDE-c is a powerful tool to increase the comprehensiveness of present CBAs of risk mitigation strategies. Indeed, while costs estimation is based on quite consolidated practices, available flood damage models do not allow for a comprehensive estimation of avoided damage, including both direct and indirect damage to all exposed sectors (Meyer et al., 2013); as a consequence, CBAs are presently limited to the direct avoided damage to people and some exposed items (see e.g. Ballesteros-Cánovas et al., 2013; Saint-Geours et al., 2015; Meyer et al., 2013; Shreve and Kelman, 2014; Arrighi et al., 2018). On the opposite, the importance of developing new and reliable models for comprehensive flood damage assessments has been highlighted in recent investigations of past flood events (Pitt, 2008; Jongman et al., 2012; Menoni et al., 2016), showing that losses to the different sectors weigh differently according to the type of the event and the affected territory. Still, as for other damage models, the variability of parameters required by AGRIDE-c together with the limited availability of data for its validation (see Section 2) suggest the use of the model not in absolute terms (i.e. to evaluate the effectiveness of a specific measure), but as a tool to compare among several alternatives (Molinari et al. 2019).

The development of AGRIDE-c and its implementation in the Po Plain highlighted that a thorough understanding and modelling of damage mechanisms to crops is also useful to orient farmers' behaviour towards more resilient practices. For example, for the context and crop investigated in the case study, Figure 6 highlights that abandoning the production is always the worst strategy, leading to a relative damage greater than 100% in any vegetative stage and for any flood intensity, due to the combined effect of the total loss of the turnover (apart from EU contributions) and of the costs sustained by the farmer before the flood. On the other hand, when flood intensity implies significant yield loss, reseeding (if possible) must be preferred to continuation, limiting the relative damage to 80%.

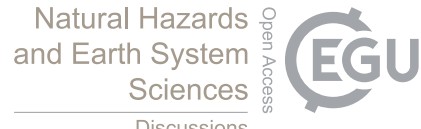



**Water depth < 130 cm**

| Phase | Month | Strategy | <5 | 5 | 6 | 7 | 8 | 9 | 10 | 11 | >11 |
|---|---|---|---|---|---|---|---|---|---|---|---|
| Bare field | Jan | c | | | | | 36% | | | | |
| | | r | | | | | - | | | | |
| | | a | | | | | - | | | | |
| | Feb | c | | | | | 36% | | | | |
| | | r | | | | | - | | | | |
| | | a | | | | | - | | | | |
| | Mar | c | | | | | 36% | | | | |
| | | r | | | | | - | | | | |
| | | a | | | | | - | | | | |
| Initial phase | Apr | c | | | | | - | | | | |
| | | r | | | | | 80% | | | | |
| | | a | | | | | 158% | | | | |
| | May | c | | | | | - | | | | |
| | | r | | | | | 80% | | | | |
| | | a | | | | | 158% | | | | |
| Growing | Jun | c | 36% | 63% | 90% | 117% | 144% | 171% | 198% | 225% | - |
| | | r | | | | | 80% | | | | |
| | | a | - | | | | 187% | | | | |
| Flowering | Jul | c | 36% | 90% | 144% | 198% | - | | | | |
| | | r | | | | | - | | | | |
| | | a | - | | | | 191% | | | | |
| | Aug | c | 36% | 90% | 144% | 198% | - | | | | |
| | | r | | | | | - | | | | |
| | | a | - | | | | 191% | | | | |
| Maturation | Sep | c | | | | | 36% | | | | |
| | | r | | | | | - | | | | |
| | | a | | | | | - | | | | |
| | Oct | c | | | | | 36% | | | | |
| | | r | | | | | - | | | | |
| | | a | | | | | - | | | | |
| Bare field | Nov | c | | | | | 36% | | | | |
| | | r | | | | | - | | | | |
| | | a | | | | | - | | | | |
| | Dec | c | | | | | 36% | | | | |
| | | r | | | | | - | | | | |
| | | a | | | | | - | | | | |

**Water depth ≥ 130 cm**

| Phase | Month | Strategy | <5 | 5 | 6 | 7 | 8 | 9 | 10 | 11 | >11 |
|---|---|---|---|---|---|---|---|---|---|---|---|
| Bare field | Jan | c | | | | | 36% | | | | |
| | | r | | | | | - | | | | |
| | | a | | | | | - | | | | |
| | Feb | c | | | | | 36% | | | | |
| | | r | | | | | - | | | | |
| | | a | | | | | - | | | | |
| | Mar | c | | | | | 36% | | | | |
| | | r | | | | | - | | | | |
| | | a | | | | | - | | | | |
| Initial phase | Apr | c | | | | | - | | | | |
| | | r | | | | | 80% | | | | |
| | | a | | | | | 158% | | | | |
| | May | c | | | | | - | | | | |
| | | r | | | | | 80% | | | | |
| | | a | | | | | 158% | | | | |
| Growing | Jun | c | 36% | 63% | 90% | 117% | 144% | 171% | 198% | 225% | - |
| | | r | | | | | 80% | | | | |
| | | a | - | | | | 187% | | | | |
| Flowering | Jul | c | | | | | - | | | | |
| | | r | | | | | - | | | | |
| | | a | | | | | 191% | | | | |
| | Aug | c | | | | | - | | | | |
| | | r | | | | | - | | | | |
| | | a | | | | | 191% | | | | |
| Maturation | Sep | c | | | 36% | | | | 90% | 144% | 198% | - |
| | | r | | | | | - | | | | |
| | | a | | | - | | | | 195% | | |
| | Oct | c | | | 36% | | | | 90% | 144% | 198% | - |
| | | r | | | | | - | | | | |
| | | a | | | - | | | | 195% | | |
| Bare field | Nov | c | | | | | 36% | | | | |
| | | r | | | | | - | | | | |
| | | a | | | | | - | | | | |
| | Dec | c | | | | | 36% | | | | |
| | | r | | | | | - | | | | |
| | | a | | | | | - | | | | |

**Figure 6: Relative damage to maize crops (in case of minimum tillage) for the different combinations times of occurrence of the flood (i.e. month), flood intensities (i.e. water depth and flood duration) and damage alleviation strategies ("c"=continuation; "r"=reseeding; "a"=abandoning**




Another strength of the implemented approach is the possibility of investigating the effect on damage of possible changes in the physical and economic context in which the farm is located (or, in another terms, to perform a sensitivity analysis of input variables). For example, for maize, the model reveals that even a reduction of 10% of the yield in the Scenario 0 (with respect to the value adopted in the analysis) impacts the damage scenarios, leading to a relative damage greater than 100%, even in

the case of reseeding in April and June (Figure 7) and continuation in July and September (when yield loss is expected). The same occurs (not shown here) if the selling price decreases more than 12.5%, or EU contribution for the minimum tillage is not considered or production costs increase more than 10%; all of these scenarios are realistic in the context under investigation (i.e. lower yields have been observed in other regions of the Po Plain, prices and costs are highly variable, while only few farmers apply for EU contributions for the minimum tillage) highlighting, in particular, the importance of EU contributions

for damage alleviation.

From another perspective, the development of AGRIDE-c highlighted some challenges for the hydrology and the hydraulic community. In fact, application of the model requires a relatively detailed set of hazard input variables which are often not supplied in existing flood hazard maps (de Moel et al., 2009). Such knowledge would require a shift from traditional 1D steady hydraulic models to 2D unsteady hydraulic models - coupled with suitable sediment and contaminant transport models - in all

flood prone areas, which is not easily achievable in a short time, both for technical and economic constraints. Thus, rapid approximate methods for the estimation of hydraulic variables of interest must be developed (e.g. Scorzini et al., 2018). In addition, a further problem arises with respect to the estimation of the probability of occurrence of the different inundation scenarios. Given the importance of the time of the year, risk estimates should be based no more on annual probabilities, but on seasonal probabilities (Förster et al., 2008; Klaus et al., 2016; Morris and Hess, 1999; USACE, 1985); this would imply

changing present conceptualisation of flood return periods. It is worth noting that the key role played by the time of the event affects also the identification of crops of interest, that should take into account which crops are actually in place when the event occurs. In fact, because of rotation techniques, it may happen that several different crops can exist on the same plot at different times of the year.

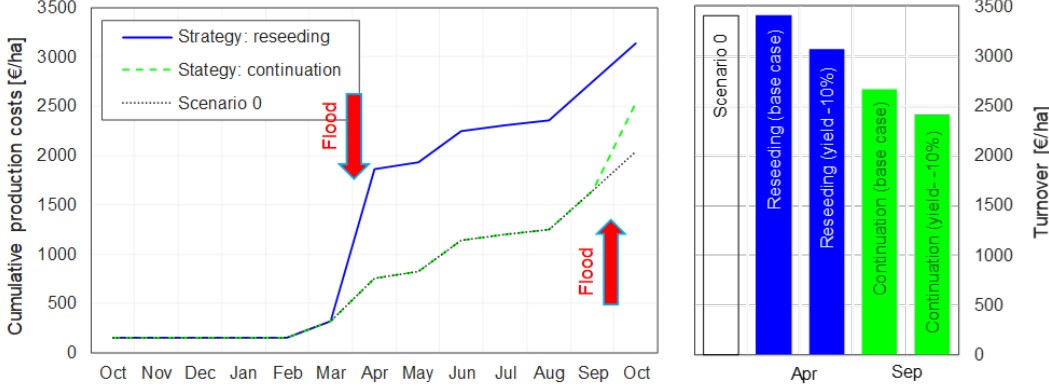

**Figure 7. Distribution of cumulative production costs for grain maize during the year and annual turnover in the scenario 0 and in the case of a flood occurring in June, in case of reseeding (blue) and September, in case of continuation (green). The figure compares the situation related to the baseline scenario 0 and the scenario 0 with a yield reduction of 10%**



## 6 Conclusions

This paper presented AGRIDE-c, a conceptual model for assessing flood damage to crops and its implication for farmers. The model has been then exemplified in the Po Plain – North of Italy, for which a spreadsheet (partly customizable by users) for the calculation of damage has been also developed.

According to authors' knowledge, AGRIDE-c represents the first attempt to organise all the available knowledge on flood damage to crops in a usable and consistent tool (i.e. the model integrates physical and economic approaches) that can be implemented to guide the flood damage assessment process, in different geographical and economic contexts. This aspect is the main strength of the model, given the fragmented and not consolidated literature on the topic. On the other hand, the development of the model highlighted different challenges that the scientific community still needs to face in order to allow

reliable estimations of flood damage to crops. Indeed, the exercise carried out for the Po Plain pointed out that further investigations on the modelling of damage mechanisms are required to fully implement AGRIDE-c in a specific context: at present, (over)simplifications are made, for instance, regarding the physical damage to soil and its effect on crops or the influence of flood intensity on yield quality reduction.

Despite current limitations, the case study demonstrates the usability of the conceptual model; at the same time, it represents

an example of how the model can be adapted to different geographical or economic contexts, given that all the assumptions and hypotheses made in the sub-models are clearly described; importantly, the model is based on the vegetative cycle of the crops, allowing its transferability to contexts characterised by different crop calendars or climate conditions.

Finally, according to our knowledge, the model represents the first tool for the estimation of flood damage to crops in the Italian context, and in particular in the Po Plain region.

Further research efforts will be focused on three directions: (i) a better understating of damage mechanisms, (ii) the validation of the model, even for other contexts of implementation and (iii) the extension of the model to the other components of a farm. Indeed, while most of the available literature on flood damage to agriculture focuses on crops (Table 1), comprehensive flood damage assessments would require considering all the damageable components of a farm, being perennial plants, soil, livestock, stock, equipment and machineries, buildings, permanent equipment and farm roads (Brémond et al., 2013;

Posthumus et al., 2009; Morris and Brewin, 2014).

### Acknowledgements

This work has been funded by Fondazione Cariplo, within the project "Flood-IMPAT+: an integrated meso & micro scale procedure to assess territorial flood risk".


### Author Contributions

Conceptualization, D.M., A.R.S. and F.B.; Methodology, D.M., A.R.S., F.B. and A.G.; Data Management, D.M., A.R.S. and A.G.; Analysis, D.M., A.R.S. and A.G.; Investigation of the results, D.M., A.R.S. and F.B.; Development of the spreadsheet, A.G., Writing – Original Draft, D.M.; Writing - Review, D.M., A.R.S., and F.B.





**Competing interests**

The authors declare that they have no conflict of interest

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
