# Peer review of "AGRIDE-c, a conceptual model for the estimation of flood damage to crops: development and implementation"

_Natural Hazards and Earth System Sciences, 2019_

## Referee Comment (RC1) · Patric Kellermann (Referee) · 29 Mar 2019

The paper presents a conceptual flood damage model for the agricultural sector – the AGRIDE-c model. AGRIDE-c is characterized by a general framework designed to be flexible in application in different context with marginal adaptation or consideration of local, context-specific damage variables, respectively. The model approach is based on expert judgement and complemented by information from literature. The research contributes to overcome the lack of knowledge and tools to assess flood damage and risk in the agricultural sector. The main strength of the presented model framework is its transferability, i.e. the flexibility to apply the model in contexts characterized by

different crop calendars and/or climate conditions.

After introducing the research need, the state-of-the-art of flood damage modeling in the agricultural sector is summarized. Subsequently, the AGRIDE-c model framework is described and applied in a case study in the Po River Plain. Finally, experiences from the case study are discussed and conclusions are drawn.

The paper follows a logical structure. The significant relevance of the research is substantiated by the authors in a reasonable and transparent manner. The used data and methods are sufficiently described. The authors give an adequate overview of existing literature and list the references appropriately. I recommend publishing the paper after minor revisions by the authors. In this respect, please consider the following comments:

General comments

- Please briefly discuss and justify the consideration of the element "damage to soil" in the model framework against the background that no approach to estimate this damage type as yet exists. From a theoretical point of view the implementation of this damage type is fully comprehensible and reasonable as 1) it ensures a comprehensive view of potential consequences of flooding in the agricultural sector, and 2) damage to soil can significantly contribute to overall flood damage in this sector. However, from a practitioner's perspective, the fact that the consideration of damage to soil is suggested on the one hand, but no concrete approach for such an estimation is provided (since not existent) on the other hand, can cause ambiguities. Further, a consideration of damage to soil in the model application using rough assumptions and proxies for this variable could introduce noise to the overall loss estimation rather than valuable information.

- The AGRIDE-c spreadsheet plays a central role in the model concept. It is currently provided to the reader via a hyperlink to a project website in Italian language. Due to language constraints of non-Italian-speakers as well as potential expiry of the hyperlink I suggest to additionally provide the spreadsheet in the supplement of this paper, if technical requirements of NHESS can be met or bypassed (Excel sheets cannot be

uploaded to NHESS supplements). This would ensure unlimited availability and better access of the spreadsheet.

Specific comments

Page 2, l. 6-8: The given characteristics of limited model transferability and applicability are not exclusive for agricultural sector, but rather represent general difficulties in flood damage modeling, i.e. often also apply to models for e.g. the residential or the commercial sector. I suggest to rephrase the sentence to avoid the impression that these aspects are exclusive problems of agricultural models.

Page 2, l. 23: "The paper is organized in four parts" is a bit confusing. To match this number, the exclusion of the sections "introduction" and "conclusion" is required. Moreover, in the subsequent sentences you list five different sections. Please rephrase the sentence towards a more unambiguous statement. For example, "the paper is organized as follows".

Page 3, l. 1: "The main available damage models [. . .]". This statement is unclear to me. Do you mean "prominent examples of damage models"? Please clarify.

Page 9, l. 27-30: Although in a European context floods usually have a negative effect on soils, the studies of e.g. Hein et al. (2003) and Tockner et al. (1999) show that such events can also have clearly positive effects, namely in the form of an increase of soil fertility. The fertility increase is explained by a (re-)distribution of river sediments and organic matter in the course of flooding. These river sediments replenish carbon and nutrients in topsoil and, hence, can make agricultural lands more fertile. I suggest to briefly discuss this aspect in the paper. An adaptation of Figure 2, where the box "damage to soil" currently states only the negative effect of flooding, could also be considered.

Technical corrections

Page 12, l. 16, w. 11: Grammar issue. "nor" should be replaced by "neither".

[Figure]

Page 19, l. 2, w. 13-15: Consider rephrasing "in another terms". For example, into "in other words".

Suggested references

Hein, T., C. Baranyi, et al. (2003). "Allochthonous and autochthonous particulate organic matter in floodplains of the River Danube: the importance of hydrological connectivity." Freshwater Biology 48(2): 220-232.

Tockner, K., D. Pennetzdorfer, et al. (1999). "Hydrological connectivity, and the exchange of organic matter and nutrients in a dynamic river–floodplain system (Danube, Austria)." Freshwater Biology 41(3): 521-535.

I look forward to the further progress of the model development and application.

Best regards, Patric Kellermann
* * *

---

## Referee Comment (RC2) · Anonymous Referee #2 · 9 Jul 2019

The authors present a conceptual model for the assessment of flood damage to crops, offering a novel systematic and consistent approach that can be universally applied. They demonstrate the use of the model through a case application in the Po Valley, northern Italy, focusing on flood damage to the maize crop. The paper is generally well written and argued. There is scope to improve the structure of the paper by separating the introduction of context and rationale, including statement of research objectives, and statement of methods to cover literature search, review of knowledge and construction of the analytical framework. There should be a critique of the approach. The case study then becomes results (reordering some results that currently occur in discussion). Discussion can then follow on both the case and the validity or otherwise of

generic framework. Some items currently in the conclusion, particularly on gaps/further development, can go in discussion (they appear to be recommendations). Conclusion on what has gone before can focus whether the objectives (regarding the tool, its application and its prospects ) have been met, rather than introducing new elements into discussion. The work has merit in its approach and application. However, it probably would be best to be more cautious and modest about the claims made about the comprehensiveness and novelty of the approach, and its suitability to all circumstances and contexts. The grassland /livestock and flooding complex is not referred to, nor is land drainage (see below). Further clarity on its potential application, either in cost benefit analysis of (publically funded) investments at the landscape scale in flood risk management, or in guiding individual farm-scale responses would be appropriate. The two applications are different in purpose and detail of approach. There is a difference between, for example, economic and financial appraisals. There is also a difference between ex ante appraisal and ex post evaluation, which is implied. This will support the important point made that insufficient ex post evaluation is undertaken to provide sound ex ante decisions. One particular issue requires attention, namely the importance, especially in temperate climates, of agricultural land drainage. The control of water levels in the soil, and particularly the removal of excess water and below surface 'flooding' , including during the post flood phase before field return to 'normal' is an important aspect of agricultural flood risk management and assessment . Impacts and land management responses are often driven by seasonal waterlogging and drainage problems as much as they are by surface flooding. This is certainly the case in northern Europe and North America. There should be coverage of this aspect, and the implications of not explicitly allowing for it in this model framework. Many areas of strategic importance are pump drained. Water quality, notably associated with saline flooding, a major issue in coastal and tidal areas, should be referred to with implications for costs, especially regarding remediation and subsequent year impacts. Surprising the authors do not mention climate change as a driver of concern or a factor affecting damage costs and responses. This seems an omission given the topic. Further clarity is required regarding the definition of measurements of damage. A more detailed listing, upfront, of the revenue and cost related parameters would help: these emerge in the case application later on. A table would be good to summarise the main elements of cost estimation processes /assumptions/ algorithms and where they come from. In the main, the methods draws on published data from Sub-sector models of crop damage or additional costs, such as Agenias et al. What other ones are used to transfers changes in yield, revenue and cost responses? Further clarity would help regarding the use of the terms 'turnover' and 'gross profit', ie exactly what is in these terms? They are not universally applied in farm business accounting, where the terms gross output (or gross revenue), gross margin and net margin are often used. (Turnover can for example include sales from previous production periods – just to be clear). And the definition of gross profit may or may not include elements of farm level fixed costs, such as machinery and buildings costs (again to be clear, so that the methods can be generally applied). The use of 'relative' Gross profit measured at negative % values is difficult to interpret and doesn't mean a lot. On flood scenarios, the treatment presumably here is for one-off relatively infrequent flooding on a land use that is not hitherto constrained by flood exposure. An increase in flood frequency, associated with climate change for example, or withdrawal of flood defences, could lead to increased flooding with a range of outcomes, permanent abandonment, repeat annual losses or a switch to more flood tolerant land use. How are these to be handled by the model? The paper refers to spreadsheets and supplementary data containing both data and estimation methods. I had difficulty locating the spreadsheets and understanding them when I did. This is probably my fault. It would however be good to explain what is in them and how they can be reliably accessed. There is a need to strengthen the treatment of inherent variation and uncertainty in the estimates. Most are given as single values. There is some passing reference to variation in yields in the case. How is variation modelled and reported? Linked to the last point, there is a need to provide a more systematic critique of the model and the resultant damage estimates, and implications for use and improvements . At the moment this is mainly confined to the last paragraph on page

19. The authors report that their work draws on systematic review of multiple sources, including expert judgement. This aspect, especially the latter, is under reported. Did the research approach follow a particular methodology that can be supported by literature, especially engaging experts? I think the paper can make a useful contribution and the authors should be encouraged to further develop the paper in the light of review and discussion, especially regarding the following :.

Some reordering of contents Greater clarity on context and purpose of the model, Some extensions to the literature reviewed Explicit reference to agricultural land drainage as its association with flood risk management (where drainage addresses below surface flooding), Critical review of the approach and its advantages and limitations as the basis for improving decision support in this important area (and hence holding back on some of the claims made)

Detailed comments are provided in the attached supplement

Please also note the supplement to this comment:
https://www.nat-hazards-earth-syst-sci-discuss.net/nhess-2019-61/nhess-2019-61-RC2-supplement.pdf

[Figure]

**Supplement:**

**Referee comment on NHESS 2019 61: AGRIDE-c, a conceptual model for the estimation of flood damage to crops: development and implementation**

The authors present a conceptual model for the assessment of flood damage to crops, offering a novel systematic and consistent approach that can be universally applied. They demonstrate the use of the model through a case application in the Po Valley, northern Italy, focusing on flood damage to the maize crop.

The paper is generally well written and argued. There is scope to improve the structure of the paper by separating the introduction of context and rationale, including statement of research objectives, and statement of methods to cover literature search, review of knowledge and construction of the analytical framework. There should be a critique of the approach. The case study then becomes results (reordering some results that currently occur in discussion). Discussion can then follow on both the case and the validity or otherwise of generic framework. Some items currently in the conclusion, particularly on gaps/further development, can go in discussion (they appear to be recommendations). Conclusion on what has gone before can focus whether the objectives (regarding the tool, its application and its prospects ) have been met, rather than introducing new elements into discussion.

The work has merit in its approach and application. However, it probably would be best to be more cautious and modest about the claims made about the comprehensiveness and novelty of the approach, and its suitability to all circumstances and contexts. The grassland /livestock and flooding complex is not referred to, nor is land drainage (see below).

Further clarity on its potential application, either in cost benefit analysis of (publically funded) investments at the landscape scale in flood risk management, or in guiding individual farm-scale responses would be appropriate. The two applications are different in purpose and detail of approach. There is a difference between, for example, economic and financial appraisals. There is also a difference between ex ante appraisal and ex post evaluation, which is implied. This will support the important point made that insufficient ex post evaluation is undertaken to provide sound ex ante decisions.

One particular issue requires attention, namely the importance, especially in temperate climates, of agricultural land drainage. The control of water levels in the soil, and particularly the removal of excess water and below surface 'flooding' , including during the post flood phase before field return to 'normal' is an important aspect of agricultural flood risk management and assessment . Impacts and land management responses are often driven by seasonal waterlogging and drainage problems as much as they are by surface flooding. This is certainly the case in northern Europe and North America. There should be coverage of this aspect, and

the implications of not explicitly allowing for it in this model framework. Many areas of strategic importance are pump drained.

Water quality, notably associated with saline flooding, a major issue in coastal and tidal areas, should be referred to with implications for costs, especially regarding remediation and subsequent year impacts.

Surprising the authors do not mention climate change as a driver of concern or a factor affecting damage costs and responses. This seems an omission given the topic.

Further clarity is required regarding the definition of measurements of damage. A more detailed listing, upfront, of the revenue and cost related parameters would help: these emerge in the case application later on.

A table would be good to summarise the main elements of cost estimation processes /assumptions/ algorithms and where they come from. In the main, the methods draws on published data from Sub-sector models of crop damage or additional costs, such as Agenias et al. What other ones are used to transfers changes in yield, revenue and cost responses?

Further clarity would help regarding the use of the terms 'turnover' and 'gross profit', ie exactly what is in these terms? They are not universally applied in farm business accounting, where the terms gross output (or gross revenue), gross margin and net margin are often used. (Turnover can for example include sales from previous production periods – just to be clear). And the definition of gross profit may or may not include elements of farm level fixed costs, such as machinery and buildings costs (again to be clear, so that the methods can be generally applied). The use of 'relative' Gross profit measured at negative % values is difficult to interpret and doesn't mean a lot.

On flood scenarios, the treatment presumably here is for one-off relatively infrequent flooding on a land use that is not hitherto constrained by flood exposure. An increase in flood frequency, associated with climate change for example, or withdrawal of flood defences, could lead to increased flooding with a range of outcomes, permanent abandonment, repeat annual losses or a switch to more flood tolerant land use. How are these to be handled by the model?

The paper refers to spreadsheets and supplementary data containing both data and estimation methods. I had difficulty locating these and understanding them when I did. This is probably my fault. It would however be good to explain what is in them and how they can be reliably accessed.

There is a need to strengthen the treatment of inherent variation and uncertainty in the estimates. Most are given as single values. There is some passing reference to variation in yields in the case. How is variation modelled and reported?

Linked to the last point, there is a need to provide a more systematic critique of the model and the resultant damage estimates, and implications for use and improvements . At the moment this is mainly confined to the last paragraph on page 19.

The authors report that their work draws on systematic review of multiple sources, including expert judgement. This aspect, especially the latter, is under reported. Did the research approach follow a particular methodology that can be supported by literature, especially engaging experts?

I think the paper can make a useful contribution and the authors should be encouraged to further develop the paper in the light of review and discussion, especially regarding the following :.

> Some reordering of contents

> Greater clarity on context and purpose of the model,

> Some extensions to the literature reviewed

> Explicit reference to agricultural land drainage as its association with flood risk management (where drainage addresses below surface flooding),

> Critical review of the approach and its advantages and limitations as the basis for improving decision support in this important area (and hence holding back on some of the claims made)

**Abstract**

I think the abstract would better begin with a statement of context and purpose, and how the proposed model seeks to make a contribution to decision support. I think it best to avoid giving the paper an identity by using 'this paper….' as a writing style here and in the manuscript itself ; it is the authors who are reporting their work. As above, I think some cautious modesty would be advisable. CBA implies welfare assessment. Farmer decision support is something else.

Manuscript.

| Page/line | Comment |
|---|---|
| 1/20 | What are flood risk management plans, and what is the implication of CBA ?. This implies public investment at the landscape scale, often funded through the public purse, as implied by CBA |
| 1/23 | I would avoid, 'in this paper', here and elsewhere |
| 1/25 | River restoration usually implies rejoining the river to its floodplain and set back of (previously installed) flood defences in the conventional sense.
see
Morris J, Bailey AP, Lawson CS, Leeds-Harrison PB, Alsop D, Vivash R (2008) The economic dimensions of integrating flood management and agri-environment through washland creation: A case study from Somerset, England. J Environ Manage 88:372-381
Rouquette JR, Posthumus H, Morris J, Hess TM, Dawson, QL, Gowing DJG (2011) Synergies and trade-offs in the management of lowland rural floodplains: an ecosystem services approach. Hydrol Sci J 56(8):1566-1581

Is the context to justify of guide decisions in flood risk management infrastructure and operations made at the landscape/sub catchment/shoreline scale , with support from the public purse. This is the case in many parts of northern Europe and north America. Getting a handle on damage costs to agriculture is part of this ? |
| 1/29 | I think this is partly reflecting a limitation of the use of selected key literature search terms and also confinement to formal academic, rather than grey literature and institution-based activities and outputs.
There is a history here in this topic : Since the 1930s, and probably up to the mid1980s, the focus in this area in northern Europe was on 'land drainage' of which flood protection , (rather than 'flood risk management'), was a part. Major investments, including large scale pumping schemes, were made to control /remove excess soil water and simultaneously alleviate surface flood from river, tidal and shore line sources. Many of these investments were 'land reclamation (for agric) projects' often involving major river works (and not river restoration) . Thus land drainage and flood control were and are inextricably integrated (just as irrigation and drainage are). The authors should in my view show an understanding of this nexus, and consider how, without undermining what they have done, it can be incorporated here.

Including the terms agricultural/land drainage in the search would go some way towards this, as would 'flood risk '. Much of the work was carried out by research |

| | |
|---|---|
| | institutions as part of national programmes and is reported in sources that are not as easy to access.

A bit dated , but see for example, Morris, J. 1992.

Agricultural land drainage, land use change and economic performance: Experience in the UK. Land Use Policy
Volume 9, Issue 3, July 1992, Pages 185-198

And for decision support :

See Chapter 9 Flood Risk Management for Agriculture, in:
• Penning-Rowsell, E., Priest, S., Parker, D., Morris, J., Tunstall, S., Viavattene, C., Chatterton, J. and Owen D. (2013) Flood and Coastal Erosion Risk Management: A Manual for Economic Appraisal, Routledge, Abingdon, Oxford |
| 2/5 | I think also there has been a policy shift, especially in Europe post 1980s when agricultural surpluses increased under EU CAP and the subsides to agric were being challenged , and urban flood damage increased in absolute as well as relative importance.
Also the drainage link is important here : the emphasis in Europe and N America was on drainage land and reclamation. |
| 2/13-14 | Suggest avoid etc , and 'this paper' |
| 2/15 | Some of the comments here seem premature: we haven't yet explained the approach and the model, but seem to be drawing conclusions , unless these are objectives . The authors might want to consider a clear statement of the objectives of the work reported here, and then subsequently review the extent to which they have been able to meet them . |
| 2/25 | Should table 1 be part of methods ?
What of 'flood risk' and 'drainage' as key search terms ?
And using experts to identify sources ? |
| 3/ 3 | Would be good to clarify the perspective and purpose of the assessment of damage costs: ex ante or ex post, and the implications : the term ex post is used later without explanation. |
| 3/10 | Agree there is paucity of data on actual flood impact costs , recorded during and post flood. This observation is not confined to the agricultural sector (Chatterton et al, for the English cases for example, including agricultural damage ).

• Chatterton, J; Clarke, C; Daly, E; Dawks, S; Elding, C; Fenn, T; Hick, E; Miller, |

| | |
|---|---|
| | J; Morris, J; Ogunyoye, F; Salado R. .2016. The costs and impacts of the winter 2013 to 2014 floods. Report SC140025/R1. Environment Agency, Bristol. http://rpaltd.co.uk/uploads/report_files/the-costs-and-impacts-of-the-winter-2013-to-2014-floods-report.pdf

There is a large, albeit now dated literature on drainage /water logging impacts on agricultural production that should be referred to, with modelling of the link between soil- water, crop growth and yields, and particularly linked to water level management in the context of land drainage and associated flood control measures. |
| 3/10 | See Chapter 9, section 9.5, p336 in Penning-Rowsel, opcit
For FLOOD$_{FARM, that}$ assesses the cost of flooding at the farm scale

Where FLOODFARM = (costs associated with flood impacts on) ARABLE+ +GRASS+LIVESTOCK+OTHER.

See also

Dunderdale J A L and Morris J. 1997. The Benefit: Cost Analysis of River Maintenance. Water and Environment Journal. Volume11, Issue 6

Pages 423-430 https://doi.org/10.1111/j.1747-6593.1997.tb01375.x |
| 3/25 | I am not sure the assumption of full loss is true here.

The Posthumus, and the Morris and Brewin examples, based on farmers reported assessment of damages, incorporated 'partial' losses, and also losses in the following years.

And also on farms adapting to flood risk:
     Pivot J.M., Josien E. & Martin P. Farms adaptation to changes in flood risk: a management approach. J Hydrol 2002, 267, 12–25.

The ex-ante estimation methods described in Penning Rowsel above, for use in the appraisal of flood investments for agriculture, explicitly build in allowance for seasonal variation in yield loss between different crops (including grass) and livestock. |

| | |
|---|---|
| 3/29 | Should define Gross profit as gross output minus direct costs. The term Gross Margin is widely used in agricultural /farm business accounting circles.

(there is an interesting accounting challenge here : what is considered a direct, avoidable cost in the context of flood impacts, especially when lots of field operations are carried out by contractors) |
| 6/1 | The agricultural flood damage estimation |
| 6/7 | Is this a tautology ? |
| 6/7 | Should this be' and/or': with respect to data source, estimation and valuation methods: eg some models have both physical quantities and unit monetary values. |
| 6/10 | Implies that this would be good idea?

Again need to set in context of the purpose of the 'modelling', high level or detailed assessment ? A number of Environmental bodies use very high level 'cost calculators' to derive quick assessments of flood impacts at the large scale , eg using 'standardised' damage costs $/ha, for example to respond to immediate questions by politicians post flood .

There is guidance on this > The UK Environment Agency use a Flood Cost Calculator ,
 European Commission are promoting a standard approach to disaster observation , see for example
 http://publications.jrc.ec.europa.eu/repository/bitstream/JRC110489/loss-database-architecture-jrc110489.pdf |
| 6/7 | well reproduce increasing costs? |
| 6/12 | Says Agenais model is physically, presumably yield based , but then says it uses gross profit (gross output (turnover) less direct costs : isn't this monetary (cost) based .

Some further clarity of the distinction between physical and monetary estimation would be useful with definition of terms used |
| | They both imply that duration is probably more important than depth ? |
| 7/20 | Some more detail on the methods used to define the boundary of investigation , and the methods used to elicit important parameters and values from experts and other sources. Was a formal research method used? Was the research review for example formally a 'systematic' review, and were the experts 'systematically' engaged? Would be good explain how the research topic was framed and bounded , and the issues arising. What is the implication of an expert based approach here? This is an important methodological aspect, and liable to bias that needs to be managed ? |

| | |
|---|---|
| 7/30 | How is turnover defined . For the purpose here is it Gross Output (Q x P) specifically for the damage to crop outputs in a given period. Turnover in an accounting sense can be something else. Need to explain. |
| | Need to be explicit on definition of production costs here. Presumably the concern with a costs across the farm business (non revenue items), including replacement and remedial costs, net of savings in uncommitted costs Gross profit is usually after direct costs (or the cost of good sold) , but much depends on how overheads/fixed costs are categorized . How are changes in machinery operating costs , or 'other' damage costs to machinery, buildings and infrastructure being assessed, or are they not included here , given the implied focus on 'field' scale costs?

 I think a table to support equation 1 should show the revenue and cost items that are used in the assessment : what is in and what is not ? Lots of jobs are done by contractors : how are these valued ? what of within season reseeding costs, reduction in gross output or profit associated with crop substitution , clean up and remedial works, following year impacts? A list would be good . I see these come later for the Po example, but a classification for the model would be useful; Elements are suggested in figure 1 , but it is not clear which are explicitly measured revenue and cost items |
| 8/figure | Useful diagram. Where would salinity fit, and field flooding/waterlogging as it affects field access and timing of operations both within and beyond the immediate flood period ?
 Not all elements are 'valued' in the model

 Pri'c'es. |
| 8 | Does the model include grassland and associated grassland management and livestock systems? If so, how are flood impacts assessed? |
| 9 | A summary of estimation parameters and algorithms would be helpful, possibly linked to the table of estimation items referred to earlier, summarizing the estimation basis . Presumably these are listing in the supporting spreadsheets: I tried but had difficulty accessing

 See my comment on the Po case later : the approach is one of 'estimation transfer' . And there are some implicit criteria for transfer that could be made more explicit

 It would be good to say what is not in there : are damage costs to farm infrastructure, crops in store, included ? |

| | |
|---|---|
| 10/10 | Are there thresholds for assumptions on crop switching/reseeding ? |
| 10/25 | So the scenario is for a single freshwater flood occurring in a given production year ? |
| 11/5 | Implications of grassland? |
| 11/10 | What year price base is used ? Were annual price series inflation adjusted to a common year ? similarly with costs? |
| | 'annual EU contributions for agriculture as a further income for the farmer and, in detail, the subsidies given to agricultural activities in...'

 Not clear how these are being treated. Presumably farmers get decoupled income support at the farm scale under CAP and these are unaffected by the flood, so can be left out for a single flood event. What of production subsidies: will not these also continue for the year of the flood, so from a farmers viewpoint costs (and cost savings) are net of subsidies? |
| | consultation of regional price books: reference? |
| 11 | Is the assumption that all the costs shown in Fig 3 are direct costs (and therefore included in Gross profit as defined here) and are potentially 'avoidable' . This might be the case if farmers are using contractors , but if they are using own equipment and labour, how much of these are avoidable costs. Some explanation of the treatment of field operations and related costs would be useful. Some costs are more direct than others. The reference to fixed costs on the next page suggests that most costs are regarded as direct. The estimates are very sensitive to assumptions about the treatment and behaviour of costs : a tricky subject.

 I don't quite follow: I got E927 using the numbers presented , but there may be other costs not shown.

 Even so, the gross profit as defined for maize seems high > maize farmers in the Po Valley are doing well. |
| 12/10 | This approach should be more fully explained in describing the model above , that algorithms are judiciously 'transferred' from research applications elsewhere according to suitability/relevance, and availability |
| 12/16 | Delete first 'nor' |
| 12/25 | According to regional price books, restoration costs have been estimated to be equal to 500 €/ha (see Table 3). Would be good to reference these sources: Were contractors contacted? These seem very high unit costs . As for that matter do field operating costs , eg Harvesting at almost E800 /ha? |
| 12/25 | So the damage to soil box in Figure is aspirational? |
| 14/10 | I am surprised that a yield (and possibly price) penalty is not included in the assessment of reseeded crops, given the importance of timing of |

| | operations. There are generic yield functions available for timeliness that would support a relative estimate of yield and gross output loss. This is one topic where local experts and farmers would have an empirically based view . |
|---|---|
| | The comment about variation and uncertainty in the estimates is valid for the modelling as a whole, and should be made as part of the method critique |
| 14/15 | yes |
| 14/16 | Break stage? There is no crop in the field? Presumably also depends on crop rotation . |
| 15/22 | In my view gross output or gross revenue would be a better term than turnover, throughout . (Turnover refers to total sales in a period, sales may include items from other production periods) |
| 15/24 | Seems unlikely that there would be no yield penalty for delayed planting. Furthermore, reseeding would probably not be feasible immediately post flood because of field conditions . Penalty delay functions could be used . |
| 15/30 | Finally? |
| 16/figure | Would be good to make the axes consistent amongst the graphs , and for cost and turnover estimates. |
| | Would also be good to indicate net margin (or gross profit) , although this might complicate the graph. If a read it correctly, for a june flood, reseeding will not make sense , especially if there is (likely) yield penalty: I note for this graph the two 'y' scales are common |
| 17/9 | This raises the question about likely average annual damaged according to the likelihood of a flood occurring within given months : where information is available on annul flood probability, and seasonal distribution, and to complicate further, whether seasonal distributions vary according the severity of the flood ? I see this is raised later |
| 18/figure 6 | Is this really a table. The title does not explain that it is relative gross profit : this is difficult to interpret when the preceding assessment was made with respect to turnover ad costs, so some clear explanation is required. Is a relative loss of gross profit greater than 100% a helpful measure? |
| 18/15 | The use of the term CBA needs explanation: it implies public choice and assessment of welfare change associated with public investments . |
| 18/16 | Quite consolidated practices. Meaning? |
| | limited to the direct avoided damage to people and some exposed items . this is not clear |
| | 18/19, Conversely? See Penning Rowsel et al for Guidance, and for the UK approach |
| 18/24 | The points here are not clear. I suggest the whole paragraph might be recrafted |

| | |
|---|---|
| | to advantage, with some examples to support the argument |
| 18/26 and para | I am not convinced Figure 6 does this. What does the greater than 100% refer to: is this the gross profit estimate in Figure 6. Assumption of no yield loss with (delayed) reseeding probably underestimates losses . There may be opportunities for reseeding with a different crop, especially between winter sown and spring sown crops |
| 17/30 | Apart from EU contributions? Not clear |
| 17/30 | Sustained? Already committed/incurred |
| 19/0-10 | These are valid and critical points., and fundamentally concern the underlying variation and uncertainty in the estimates (that have been single values so far). In my view it would be more appropriate to include the treatment of variation and uncertainty in the description of methods and the presentation of results of the case, rather than raise it for the first time here in discussion, where the purpose is to critical discuss the methods and results. |
| 19/figure 7 | This is results and should go there above. The figure is presumably for the Po case? The likely effect of a 10% penalty that would most likely arise due to (delayed) planting is apparent : negative gross profit. A figure showing absolute changes in gross profit (as defined here ) might be useful in the results section. |
| 19/16 | Rather than saying 'must' it would be better to say why, identifying the advantage of doing so |
| 19/17 | Perhaps rather than 'no more' , 'not only…. but also' seasonal probabilities Is the Morris and Hess ref 1988? |
| 20/0 | This paper? The reference to the spreadsheet and to supplementary data needs further support : these are mentioned in passing |
| 20/5 | It depends how far the Authors have looked, and with the information presented here it is difficult to judge whether they can substantiate the claim. It might be fair to say they see advantage in developing a generic framework that can potentially be applied across different geographical and economic contexts , and they have made progress in this respect. For example, in more temperate part of Europe, land drainage is a particularly critical component of the land use: flooding nexus, and is particularly critical during post flood periods . |
| 20/16 | It would be useful to have a description of the sub models used , as referred to earlier . A summary table showing the estimation methods and sources would be |

| | particularly helpful , linked to supplementary data. |
|---|---|
| 20/20 | Damage mechanisms- Meaning ?
Drainage and soils might be important also. And also salinity issues in coastal areas, as referred earlier

. |

---

## Referee Comment (RC3) · F. GRELOT (Referee) · 12 Jul 2019

In its first part, this paper presents a "conceptual model" for estimating damage induced by flooding of cultivated agricultural plots. This "conceptual model" shall be seen as what should be considered for someone that wish to implement a model that assess "damage to crops" in some given context. This first part is mainly based on a litterature review of past studies. In its second part, the paper presents an implementation of this "conceptual model" on a specific context (in the Po plain, in Province of Lodi). This implementation relies partly on the expertise of some experts, and on the reuse of some models coming from the litterature. This implementation has been made through the

development of a "spreadsheet" that can be find on internet, and has been developped for 4 types of crops (maize grain, wheat, barley and grassland), a special focus being done on maize grain within the article.

General comments

My main concern is on the innovation provided by the article compared to precedent studies. From precedent studies, referred in the state of art section, it can be seen that many bricks presented in the article were already existing. For instance, a "conceptual model" has yet been formalized since the 80's in the USA, combining in a certain way "physical damage" and their "economic implications" in terms of loss of added value. Even if all detail are not given, many specifications are provided (see the user's manual of AGDAM, link provided below). Part of those details may be available from other studies. For instance in our works (Agenais et al., 2013), we explicit completely how we link "physical damages" with "economic implications". Thus, I recommend that the article should be more specific on what it gets from previous studies, and what it added.

This is not specific to the "conceptual model". For instance, in their conclusion (page 20), the authors state that "According to authors' knowledge, AGRIDE-c represents the first attempt to organize all the available knowledge on flood damage to crops in a usable and consistent tool (i.e. the model integrates physical and economic approaches) that can be implemented to guide the flood damage assessment process, in different geographical and economic contexts." I do not master totally the American approach, but it has been developed to be used in different context (USA is a large country...) and has been developed as a tool used by USACE (ADGDAM). Another example is coming from Hess and Morris (1988), that also organized their work on grasslands in a framework comparable to that of AGRIDE-c and included it in a tool (SCADE). Last example, concerning our work, I can specify that in chapter 3, the presentation of the methodological framework gives a clear explanation that damage are considered as loss of added value and how to link them to physical effects of flood. In chapter 4,

we give some precision on modelling of damage to sub-component of farms (which include crops, vegetal material, soil, equipment). For crops, we explain how to take into account farmers' strategies (continuation, abandoning, reseeding for instance), depending on when the flood occurs compared to the crop calendar. In chapter 5 we explain that all this has been implemented in a tool called floodam (now floodam-agri), which aim is to help to adapt damage modelling to different context, including prices, crops calendars, and even also the question of the typology of culture to best fit with typology of GIS. I want also to point that the authors present results for 4 types of crops, whereas both the American and the French approaches deal with many more types (including permanent crops).

Secondly, I feel uncomfortable with the articulation of the two parts of the article, "conceptual model" and "implementation". From figure 2, it is expected that implementation of AGRIDE-c shall take into account all the phenomea described. But when coming to the implementation, it appears that many of those phenomena are not taken into account (loss of fertility due to sedimentation or loss of quality for instance). This shall be exposed in a clearer way not to induce false expectations on the scope of the study.

Third, I think that the conceptual model is incomplete, at least for perennial crops (such as vineyard, but also grassland). First, it does not seem clear that the authors get that for some culture it may be useful to separate crops (fruits) and vegetal material (trees). This is not restricted to vineyard and orchards, but may also be important for asparagus, and even certain type of grasslands. If vegetal material is affected by flood, there may be at least two types of effect that last more than one year. For a given plot, if some plants are to be "destroyed" by a flood, it is expected that yield reduction and thus of products, but also variations of charges occur during the following years. This type of effects are for instance implicitly taken into account in Agenais et al. Secondly

Forth, I think there are not sufficient description of the role of the direct consultation of experts for the current work. It is only said (page 7) that some experts were consulted (agronomists and representatives of the authorities responsible for agricultural damage

management and compensation), that this expertise were used to produce the production costs for normal activity (page 11, figure 3), the 3 possible strategies after a flood occur (pages 12-13) and an opinion on the suitability of yield reduction model coming from our works for maize grain (page 12). As many of the implementation seem to rely on the consultation of experts, I think a much more detailed description of this consultation shall be provided: how many experts has been consulted? What were their precise expertise, especially concerning flood impacts? What were their opinions on the data they provided? Have they been consulted on the results? What were their opinion on those results? What were their opinion on the transferability of those results on other context? This would strengthen this part of the work, that is almost invisible at the moment.

I would therefore recommend to be less ambitious in terms of interest of the article and to reorient it on the question of what has been necessary to adapt from previous works to a specific context. I invite the authors to be more precise on what they really include in their model, not on what they would have liked to include, because this makes things unclear for the reader. Another perspective would be to make a clear list of what that have not included. I also invite the authors to be more specific on how they have really implement their "local" model, by precising all the steps about consultations of experts. This seems important, especially for a expert-based approach. For those reasons I recommend a major revision.

Other comments

Given the scope of the article, I am not totally convinced that the state of the art analysis should be done in such a detailed way. This is more convenient for a kind of review articles. If this section should stay in such a detail version, some imprecision needs to be corrected. For instance: - P3-L16. "No model in Table 2 considers instead the behavior of farmers after the occurrence of the flood (e.g. the decision of abandoning the production or to continue with increasing production costs)..." Agenais et al. does (see chapter 4) - P3-L22. The distinction between "physically based" and "cost based"

is not clear. As formulated "cost based" models appears as simpler models where yield reduction is always total, which is not the case for "physical based" models. But for "physical based" models consequences in terms of production costs are considered. - Table 2. I am not convinced by the way some works are classified. For instance AGDAM cannot be said as "cost based", as it also consider some physical aspects on flood for yield reduction. I have not the time to check for all the works. Thus, I am not confident by what is presented in table 2. - P5-Table2 Agenais et al. present damage functions for 14 crops type, based on a detailed model of 50 crops types.

I haven't seen the demonstration of what is promised in the abstract about "comprehensive cost-benefit analyses of risk mitigation actions". What is said in the discussion (page 17) is just that AGRIDE-c provide a way to estimate direct damage to crops, but in fact, it is only one contributions among others. I also feel that the authors did not get that for CBA purpose, it should be considered a "collective perspective", without considering possible transfers. This is not clear (see remarks on the "spreadsheet" tools). I think that should be reformulated.

I haven't seen neither the demonstration that AGRIDE-c is a "a powerful tool to orient farmers' behavior towards more resilient damage alleviation practices". I do not know in detail what is the context of management of flood and agriculture in Italy. It is not presented in the article. But, this context may have some implications on what strategy would be follow by farmers, independently of what AGRIDE-c shall demonstrate. For instance, in France, if a farmer expect to receive some compensations from "Calamités Agricoles" (a State compensation scheme) or from "Assurances Récoltes" (Private insurance), he shall have to follow some recommendations concerning what he can follow as a strategy. If he does not, he may not receive any compensation. Also, it is not clear that the consequences presented are really those supported by the farmers. If there exists some compensations in Italia, this shall be included to provide a true "financial perspective" (point of view of the farmers).

I am not convinced by the the §starting line 19 on page 19, concerning the necessity of

"sediment and contaminant transport models" as the authors said before that they did not find available models to estimate the effects of sediment and contamination. This appears not coherent.

I think that some of the figures presenting the results shall be changed. In fact, in the title, the authors say that they analyze "damage to crops" but none of the figures clearly present a "net" damage. The reader has to make a mental effort to understand what are the damage from those figures (5 and 7): - make a difference between last point of the curve of scenario 0 and last points of three other curves for the production costs part - make a difference between a value given by a bar for scenario 0 and 3 other values for the "turnover" part - and then make a difference between the difference of turnover and the difference of production costs... Well, this should be done for the reader! This could allow to have a representation of the flood damage depending on the season of occurrence (as a function).

Concerning figure 1, the authors announce that relative damage is supplied by our works (Agenais et al. 2013), but this is not the case. Our results are expressed in absolute damage (see page 51 of our report). Thus, the figure 1 is an interpretation of what we have done, but this interpretation is not explained. Moreover this interpretation is necessary incorrect, as our results are presented on a seasonal time step (3 months) whereas the time scale in the figure 1 is one month. Moreover, as seen in our reports, relative damage are maximum only in summer, for long duration, and height over 130 cm, thus it is impossible to have relative damage of 100 % for any other case. I have not verified what is announced about the presentation of the results of Forster et al., but it shall be verified.

"Spreadsheet" tool

I had a look at the "spreadsheet" tool, and I share some comments on it, as I understood that all the application where made thanks to it: - There is not a manual to help people use the tool, it would be nice (necessary?). - Technically this tool is not designed

to produce damage function but to estimate damage for specific value of hazard. This is not very practical for a user interested in a "damage function". - I have some questions on how the value coming from "Agenais" were filled, as I cannot remember that the authors asked for those values, which are not detailed in our report. - For instance, in some cases (maize, germination) yield reduction may occur for flood with a duration of 0 day, and in other cases (maize, flowering) there is no yield reduction for such flood with duration of 0 day. This may be a misunderstanding of what we developed. This leads to possible damage for a flood of 0 day and 0 cm for maize, which has no sense in our works. Such a flood is typically a flood with no consequences. - Another thing is that it is not specified that all data used from Agenais et al. are only specified for negligible flow velocity. This is particularly important for maize, wheat, and barley, that are very sensitive to this parameter. This may induce a bad use of the tool. - One of the aspects that may change from site to site is the list of actions inside the crop management sequences. There are many reasons for which those actions may differ, not only in value but also in nature. This aspect is not taken into account, and doesn't seem to be easily taken into account with the provided tool. - In the tool, "EU contributions for agriculture" are included. This is more oriented for a financial analysis (point of view of a specific farmer, including transfers) than for a Cost-Benefit Analysis (collective point of view, excluding transfers). I think a clear precision on the usage of damage produced should be added. If a financial perspective is to be promoted, all insurance or compensations mechanisms should be also included to give a better view of net consequences for the farmer.

I think this tool should be added (if possible) as additional data to the final version of the article. In this case, I recommend adding a "manual user" sheet to the tool.

"Technical corrections"

At this stage, I have not focus on those aspects.

Some references (and access):

USACE. (1985). AGDAM, Agricultural Flood Damage Analysis – User's Manual (Provisionnal). CPD-48. Davis, CA, USA: US Army Corps of Engineers, Institute for Water Resources, Hydrologic Engineering Center (HEC). https://www.hec.usace.army.mil/publications/ComputerProgramDocumentation/AGDAM_UsersManual_(CPD-48).pdf

Agenais, A.-L. et al. (2013). Dommages des inondations au secteur agricole. Guide méthodologique et fonctions nationales. IRSTEA. https://irsteadoc.irstea.fr/cemoa/PUB00041499

---

## Author Comment (AC1) · 31 Jul 2019

We would like to thank the referee both for his appreciation of our paper and for the work he did on our manuscript; we greatly appreciate his comments as they may contribute to increase the manuscript robustness and, in general, to improve its quality and readability. In the following, we supply a point by point answer to the general and specific comments raised by the referee (see also attached file).

General comments:

RC1: Please briefly discuss and justify the consideration of the element "damage to

soil" in the model framework against the background that no approach to estimate this damage type as yet exists. From a theoretical point of view the implementation of this damage type is fully comprehensible and reasonable as 1) it ensures a comprehensive view of potential consequences of flooding in the agricultural sector, and 2) damage to soil can significantly contribute to overall flood damage in this sector. However, from a practitioners perspective, the fact that the consideration of damage to soil is suggested on the one hand, but no concrete approach for such an estimation is provided (since not existent) on the other hand, can cause ambiguities. Further, a consideration of damage to soil in the model application using rough assumptions and proxies for this variable could introduce noise to the overall loss estimation rather than valuable information.

Answer: We thank the reviewer for this comment and we fully agree with him on this point. Indeed, our choice to include the "damage to soil" component in AGRIDE-c, although in a simplified way, was driven by the two main reasons also raised by the reviewer: comprehensiveness of model structure and importance of this sub-component in the overall flood damage figure to agriculture; in particular, this last point clearly emerged during the interviews with local experts, who pointed out the occurrence of such damages even for flood events characterised by shallow water depths and not particularly high flow velocities. In the revised version of the manuscript, we will include these considerations in Section 4.4 in order to justify the necessity of modelling this sub-component and we will also include in Section 5 a critical discussion of possible impacts of the modelling assumptions and proxies for this component on the overall loss estimation.

RC2: The AGRIDE-c spreadsheet plays a central role in the model concept. It is currently provided to the reader via a hyperlink to a project website in Italian language. Due to language constraints of non-Italian-speakers as well as potential expiry of the hyperlink I suggest to additionally provide the spreadsheet in the supplement of this paper, if technical requirements of NHESS can be met or bypassed (Excel sheets cannot be uploaded to NHESS supplements). This would ensure unlimited availability

and better access of the spreadsheet.

Answer: We will check with the NHESS editorial support office whether the spreadsheet can be added as supplement material. Otherwise, we will upload it in a repository with an easier access.

Specific comments:

RC3: Page 2, l. 6-8: The given characteristics of limited model transferability and applicability are not exclusive for agricultural sector, but rather represent general difficulties in flood damage modeling, i.e. often also apply to models for e.g. the residential or the commercial sector. I suggest to rephrase the sentence to avoid the impression that these aspects are exclusive problems of agricultural models.

Answer: We agree with the reviewer that the transferability of damage models represents a general issue in flood damage modelling, affecting all exposed sectors. Agriculture is probably one of the most critical in terms of transferability, due to large variability of the features affecting damage mechanisms for this sector. For more clarity, in the revised version of the manuscript we will revise L.6-8 in P.2 by specifying: "Nonetheless, available damage models for agriculture are not only few in number, but are also affected by many limitations, the major related to the lack of information/data for their validation and to the large variability of local features affecting damage (i.e. strong linkage with the context under investigation) which limits their transferability to different contexts more than other exposed sectors, as the residential or commercial ones".

RC4: Page 2, l. 23: "The paper is organized in four parts" is a bit confusing. To match this number, the exclusion of the sections "introduction" and "conclusion" is required. Moreover, in the subsequent sentences you list five different sections. Please rephrase the sentence towards a more unambiguous statement. For example, "the paper is organized as follows".

Answer: The reviewer is right. In the revised version of the manuscript we will rephrase

the sentence accordingly in order to avoid confusion.

RC5: Page 3, l. 1: "The main available damage models [: : :]". This statement is unclear to me. Do you mean "prominent examples of damage models"? Please clarify Answer: Yes, we do. We will then revise the sentence as suggested.

RC6: Page 9, l. 27-30: Although in a European context floods usually have a negative effect on soils, the studies of e.g. Hein et al. (2003) and Tockner et al. (1999) show that such events can also have clearly positive effects, namely in the form of an increase of soil fertility. The fertility increase is explained by a (re-)distribution of river sediments and organic matter in the course of flooding. These river sediments replenish carbon and nutrients in topsoil and, hence, can make agricultural lands more fertile. I suggest to briefly discuss this aspect in the paper. An adaptation of Figure 2, where the box "damage to soil" currently states only the negative effect of flooding, could also be considered.

Answer: We thank the reviewer for this important comment. In the original version of the manuscript we only referred to negative flood effects on soils, because in Italy these are the most common impacts observed from past events. However, we fully concur with the reviewer on the importance of including also positive effects (e.g. increase of soil fertility) in the general conceptual model represented in Figure 2, which will be revised accordingly. In the revised version of the manuscript, we will also include some discussion on this point in subsection 4.4 (the title "Damage to soil" will be changed to "Impact to soil" in order to be more comprehensive).

Technical corrections:

Page 12, l. 16, w. 11: Grammar issue. "nor" should be replaced by "neither". Page 19, l. 2, w. 13-15: Consider rephrasing "in another terms". For example, into "in other words". Answer: These technical corrections will be fixed in the revised version of the manuscript

Please also note the supplement to this comment:
https://www.nat-hazards-earth-syst-sci-discuss.net/nhess-2019-61/nhess-2019-61-AC1-supplement.pdf

———————————————————

---

## Author Comment (AC2) · 31 Jul 2019

We would like to thank the referee both for appreciation of our work and for carefully reading our manuscript; we greatly appreciate the insightful comments as they may contribute to increase the manuscript robustness and, in general, to improve its quality and readability. In the following, we supply a point by point reply to the general and specific comments raised by the referee (see also attached file).

General comments:

RC1: There is scope to improve the structure of the paper: - by separating the introduction of context and rationale, including statement of research objectives, and statement of methods to cover literature search, review of knowledge and construction of the analytical framework. - There should be a critique of the approach. - The case study then becomes results (reordering some results that currently occur in discussion). - Discussion can then follow on both the case and the validity or otherwise of generic framework. Some items currently in the conclusion, particularly on gaps/further development, can go in discussion (they appear to be recommendations). Conclusion on what has gone before can focus whether the objectives (regarding the tool, its application and its prospects) have been met, rather than introducing new elements into discussion.

Answer: We would try to address at our best the suggestions provided by the referee, especially regarding a better ri-organisation of the contents of the introduction/literature review/methodological part, and discussion/conclusions. Still we think that the present organisation of the sections is appropriate for the explanation of the conceptual model and its exemplification. Of course, if the editor thinks that a change in the structure of the paper is required to better meet the journal standards, we will re-organise the paper as suggested.

RC2: It probably would be best to be more cautious and modest about the claims made about the comprehensiveness and novelty of the approach, and its suitability to all circumstances and contexts.

Answer: After reading the referee comment, we realised that the "scope of use" of our model was not well specified in the original version of the paper, which could lead to incorrect interpretations of our work. Indeed, without such specifications our claims appear as too wide, and we will be more specific in the revised version of the manuscript. In fact, the conceptual model has been designed to supply an estimation of flood damage: - to annual crops (i.e. not including perennial plants) - by considering one single culture (i.e. by not considering replacement of one culture with another one) - by limiting the time frame of the analysis to one "productive year" (i.e. not considering long

term damages, e.g. loss of soil productivity in the following years); - for infrequent flooding (i.e. effect of two, or more, consecutive floods is not considered) Nonetheless, as specified at page 9 line 11-19, AGRIDE-c do not consider damage to other components/elements of the farm that may induce additional damage to crops, as, for instance, damage to machineries and equipment (e.g. the irrigation plant) that may prevent cultivation for a while. Only damage to soil is considered from the evidence that, during a flood, damage to soil and plants occurs always at the same time, differently from damage to the other components which can occur or not, independently from the damage to plants; as a consequence, damage to soil and plants is modelled together, while damage to the other components could be modelled as separated factors, not included in the conceptual model. We will specify all these aspects in the new version of the paper, by hopefully clarifying what we mean with "generality" and "transferability" of the approach. We never referred in the paper to "novelty" or "comprehensiveness" of the approach, but we highlighted the fact that we are trying to encapsulate and systematise the available knowledge on damage mechanisms (to annual crops) triggered by inundation phenomena, as well as on their consequences in terms of income for the farmers.

RC3: Further clarity on its potential application, either in cost benefit analysis of (publically funded) investments at the landscape scale in flood risk management, or in guiding individual farm‐scale responses would be appropriate. The two applications are different in purpose and detail of approach. There is a difference between, for example, economic and financial appraisals. There is also a difference between ex ante appraisal and ex post evaluation, which is implied. This will support the important point made that insufficient ex post evaluation is undertaken to provide sound ex ante decisions

Answer: The CBA of flood risk mitigation strategies would require a comprehensive estimation of benefits linked to the different strategies, i.e. of the avoided loss to all exposed sectors and at different temporal scales (i.e. direct and indirect/long term

damages). Present damage modelling capacity prevents comprehensive flood damage assessments, which usually include only direct damage to people and some of the exposed assets (typically residential buildings). In such a context, by allowing the estimation of the expected loss to crops in a specific flood scenario, AGRIDE-c may support more comprehensive CBAs of public risk mitigation strategies. Of course, to meet such an objective, the tool must be critically used, e.g. by considering possible transfers of losses/gains between farmers in an economic perspective, according to the temporal and spatial scales of the analysis. Regarding individual responses, by supplying the expected damage for different types of crops and alleviation strategies (according to the expected yield reduction for different flood intensities and period of occurrence), AGRIDE-c may support individual farmers exposed to flood risk in preventing losses by supporting: the choice of the most appropriate crops to be cultivated, the choice of the best alleviation strategy to be followed once flooded, the evaluation of the opportunity to ask for a flood insurance scheme and the definition of the premium. The model was not designed to be used ex-post. This explanation will be added in the revised version of the manuscript.

RC4: One particular issue requires attention, namely the importance, especially in temperate climates, of agricultural land drainage. The control of water levels in the soil, and particularly the removal of excess water and below surface 'flooding', including during the post flood phase before field return to 'normal' is an important aspect of agricultural flood risk management and assessment. Impacts and land management responses are often driven by seasonal waterlogging and drainage problems as much as they are by surface flooding. This is certainly the case in northern Europe and North America. There should be coverage of this aspect, and the implications of not explicitly allowing for it in this model framework. Many areas of strategic importance are pump drained.

Answer: We thank the referee for highlighting these important aspects for flood risk/damage assessment and management (that were erroneously not included in the

original version of the paper), and for supplying some of the related literature. In the new version of the manuscript, the conceptual model will be modified in order to take into account of the effect of waterlogging and prolonged soil saturation on both the yield and the soil, as well as on the corresponding effect on the revenue and costs for the farmer, and then on the final flood damage. Still, these aspects will not be considered in the application of the model to the Po Valley because, according to experts' opinion, phenomena like waterlogging and prolonged soil saturation after floods are not common in the area.

RC5: Saline flooding, a major issue in coastal and tidal areas, should be referred to with implications for costs, especially regarding remediation and subsequent year impacts

Answer: The conceptual model was not conceived to cover coastal floods but we have decided to extend the context of applicability of the model in the revised version. However, the model will still be focused on "one" productive year. We will discuss limitations of this hypothesis in the discussion section.

RC6: Surprising the authors do not mention climate change as a driver of concern or a factor affecting damage costs and responses. This seems an omission given the topic.

Answer: Given that the model is focused on "one" productive year, long term effects of climate change are not considered in the model. Anyway, we will add a sentence in the introduction on the importance of climate change in exacerbating future flood damage to agriculture.

RC7: Further clarity is required regarding the definition of measurements of damage. A more detailed listing, upfront, of the revenue and cost related parameters would help: these emerge in the case application later on.

Answer: A comprehensive list of all revenue and cost related parameters cannot be compiled in the framework of the conceptual model, as most of them, especially those related to costs, are context-specific. Still, we will add examples of such parameters in

the description of the conceptual model

RC8: A table would be good to summarise the main elements of cost estimation processes /assumptions/ algorithms and where they come from. In the main, the methods draws on published data from Sub‑sector models of crop damage or additional costs, such as Agenias et al. What other ones are used to transfers changes in yield, revenue and cost responses?

Answer: a table will be added summarising the main elements and sources of revenue and cost estimation processes

RC9: Further clarity would help regarding the use of the terms 'turnover' and 'gross profit', ie exactly what is in these terms? They are not universally applied in farm business accounting, where the terms gross output (or gross revenue), gross margin and net margin are often used. (Turnover can for example include sales from previous production periods – just to be clear). And the definition of gross profit may or may not include elements of farm level fixed costs, such as machinery and buildings costs (again to be clear, so that the methods can be generally applied). The use of 'relative' Gross profit measured at negative % values is difficult to interpret and doesn't mean a lot.

Answer: we really thank the referee for the suggestion. According to the literature suggested, we will change the terms "gross profit" and "turnover" in "net margin" and "gross output", respectively, by also specifying what is included in production costs.

RC10: On flood scenarios, the treatment presumably here is for one‑off relatively infrequent flooding on a land use that is not hitherto constrained by flood exposure. An increase in flood frequency, associated with climate change for example, or withdrawal of flood defences, could lead to increased flooding with a range of outcomes, permanent abandonment, repeat annual losses or a switch to more flood tolerant land use. How are these to be handled by the model?

Answer: As explained before (RC2) the model considers damage to one productive year for infrequent floods. Limits of these assumptions will be discussed in the new version of the paper.

RC11: The paper refers to spreadsheets and supplementary data containing both data and estimation methods. I had difficulty locating these and understanding them when I did. This is probably my fault. It would be good to explain what is in them and how they can be reliably accessed.

Answer: We will ask the NHESS editorial support office whether the spreadsheet can be added as supplement material; otherwise, we will upload it in a repository with an easier access We will also develop a user manual for it.

RC12: There is a need to strengthen the treatment of inherent variation and uncertainty in the estimates.

Answer: We will include a deeper discussion on model uncertainty in the discussion section.

RC13: there is a need to provide a more systematic critique of the model and the resultant damage estimates, and implications for use and improvements

Answer: see answer to RC2, RC3 and RC12

RC14: The authors report that their work draws on systematic review of multiple sources, including expert judgement. This aspect, especially the latter, is under reported. Did the research approach follow a particular methodology that can be supported by literature, especially engaging experts?

Answer: Experts were involved with two main objectives. The first one is to support the definition, and validate the quality, of the conceptual model. The second one is to give suggestions/information on the implementation of the model in the Po Valley, above all regarding expected physical damage and costs. With respect to the first objective, an iterative process was followed. First, a semi-structured interview was conducted,

by asking experts about the main damage mechanisms/phenomena in case of flood, possible interconnections among them, important explicative variables. In this phase, results from the literature review were proposed to experts for their judgment. In the following step, experts were asked to evaluate a draft version of the conceptual model we draw according to the literature review and results from first interviews. Then, there was an iterative revision process of improved versions of the model until an agreement on its final structure was reached. With respect to the second objective, several individual meetings were organised with the aim of asking experts about context-specific information on: crops calendars, yields and prices, type, timing and costs of cultivation practices. In this phase, the transferability of the model by Agenais et al. was also discussed. Three kinds of experts were involved. One representative of the Regional Authority responsible for agricultural damage management and compensation, with more than 20 years of expertise in the management and compensation of flood damage to farms in the Lombardy Region. Two agronomists of the local association of farmers (Coldiretti Lodi), with specific knowledge on the investigated context and with direct experience in managing floods in the last 20 years. During the work, the two agronomists asked for data/information also to individual local farmers that were flooded in the past years, including also their viewpoint in the process. Finally, an academic economist, with specific expertise in agriculture, has been involved in validating the final model. A new sub-section will be added to the revised version of the paper explaining the whole process of experts' involvement.

Specific comments (see attached file)

Please also note the supplement to this comment:
https://www.nat-hazards-earth-syst-sci-discuss.net/nhess-2019-61/nhess-2019-61-AC2-supplement.pdf

―――――――――――――

**Supplement:**

**Manuscript nhess-2019-61 "AGRIDE-c, a conceptual model for the estimation of flood damage to crops: development and implementation" – Final response to referee comment 2**

We would like to thank the referee both for appreciation of our work and for carefully reading our manuscript; we greatly appreciate the insightful comments as they may contribute to increase the manuscript robustness and, in general, to improve its quality and readability. In the following, we supply a point by point reply to the general and specific comments raised by the referee.

**General comments:**

**RC1:** There is scope to improve the structure of the paper:

- by separating the introduction of context and rationale, including statement of research objectives, and statement of methods to cover literature search, review of knowledge and construction of the analytical framework.
- There should be a critique of the approach.
- The case study then becomes results (reordering some results that currently occur in discussion).
- Discussion can then follow on both the case and the validity or otherwise of generic framework.

Some items currently in the conclusion, particularly on gaps/further development, can go in discussion (they appear to be recommendations). Conclusion on what has gone before can focus whether the objectives (regarding the tool, its application and its prospects) have been met, rather than introducing new elements into discussion.

**Answer:** We would try to address at our best the suggestions provided by the referee, especially regarding a better ri-organisation of the contents of the introduction/literature review/methodological part, and discussion/conclusions. Still we think that the present organisation of the sections is appropriate for the explanation of the conceptual model and its exemplification. Of course, if the editor thinks that a change in the structure of the paper is required to better meet the journal standards, we will re-organise the paper as suggested.

**RC2:** It probably would be best to be more cautious and modest about the claims made about the comprehensiveness and novelty of the approach, and its suitability to all circumstances and contexts.

**Answer:** After reading the referee comment, we realised that the "scope of use" of our model was not well specified in the original version of the paper, which could lead to incorrect interpretations of our work. Indeed, without such specifications our claims appear as too wide, and we will be more specific in the revised version of the manuscript. In fact, the conceptual model has been designed to supply an estimation of flood damage:

- to annual crops (i.e. not including perennial plants)
- by considering one single culture (i.e. by not considering replacement of one culture with another one)
- by limiting the time frame of the analysis to one "productive year" (i.e. not considering long term damages, e.g. loss of soil productivity in the following years);
- for infrequent flooding (i.e. effect of two, or more, consecutive floods is not considered)

Nonetheless, as specified at page 9 line 11-19, AGRIDE-c do not consider damage to other components/elements of the farm that may induce additional damage to crops, as, for instance, damage to machineries and equipment (e.g. the irrigation plant) that may prevent cultivation for a while. Only damage to soil is considered from the evidence that, during a flood, damage to soil and plants occurs always at the same time, differently from damage to the other components which can occur or not,

independently from the damage to plants; as a consequence, damage to soil and plants is modelled together, while damage to the other components could be modelled as separated factors, not included in the conceptual model.

We will specify all these aspects in the new version of the paper, by hopefully clarifying what we mean with "generality" and "transferability" of the approach. We never referred in the paper to "novelty" or "comprehensiveness" of the approach, but we highlighted the fact that we are trying to encapsulate and systematise the available knowledge on damage mechanisms (to annual crops) triggered by inundation phenomena, as well as on their consequences in terms of income for the farmers.

**RC3**: Further clarity on its potential application, either in cost benefit analysis of (publically funded) investments at the landscape scale in flood risk management, or in guiding individual farm-scale responses would be appropriate. The two applications are different in purpose and detail of approach. There is a difference between, for example, economic and financial appraisals. There is also a difference between ex ante appraisal and ex post evaluation, which is implied. This will support the important point made that insufficient ex post evaluation is undertaken to provide sound ex ante decisions

**Answer:** The CBA of flood risk mitigation strategies would require a comprehensive estimation of benefits linked to the different strategies, i.e. of the avoided loss to all exposed sectors and at different temporal scales (i.e. direct and indirect/long term damages). Present damage modelling capacity prevents comprehensive flood damage assessments, which usually include only direct damage to people and some of the exposed assets (typically residential buildings). In such a context, by allowing the estimation of the expected loss to crops in a specific flood scenario, AGRIDE-c may support more comprehensive CBAs of public risk mitigation strategies. Of course, to meet such an objective, the tool must be critically used, e.g. by considering possible transfers of losses/gains between farmers in an economic perspective, according to the temporal and spatial scales of the analysis. Regarding individual responses, by supplying the expected damage for different types of crops and alleviation strategies (according to the expected yield reduction for different flood intensities and period of occurrence), AGRIDE-c may support individual farmers exposed to flood risk in preventing losses by supporting: the choice of the most appropriate crops to be cultivated, the choice of the best alleviation strategy to be followed once flooded, the evaluation of the opportunity to ask for a flood insurance scheme and the definition of the premium. The model was not designed to be used ex-post. This explanation will be added in the revised version of the manuscript.

**RC4:** One particular issue requires attention, namely the importance, especially in temperate climates, of agricultural land drainage. The control of water levels in the soil, and particularly the removal of excess water and below surface 'flooding', including during the post flood phase before field return to 'normal' is an important aspect of agricultural flood risk management and assessment. Impacts and land management responses are often driven by seasonal waterlogging and drainage problems as much as they are by surface flooding. This is certainly the case in northern Europe and North America. There should be coverage of this aspect, and the implications of not explicitly allowing for it in this model framework. Many areas of strategic importance are pump drained.

**Answer:** We thank the referee for highlighting these important aspects for flood risk/damage assessment and management (that were erroneously not included in the original version of the paper), and for supplying some of the related literature. In the new version of the manuscript, the conceptual model will be modified in order to take into account of the effect of waterlogging and prolonged soil saturation on both the yield and the soil, as well as on the corresponding effect on the revenue and costs for the farmer, and then on the final flood damage. Still, these aspects will not be considered in the

application of the model to the Po Valley because, according to experts' opinion, phenomena like waterlogging and prolonged soil saturation after floods are not common in the area.

**RC5:** Saline flooding, a major issue in coastal and tidal areas, should be referred to with implications for costs, especially regarding remediation and subsequent year impacts

**Answer:** The conceptual model was not conceived to cover coastal floods but we have decided to extend the context of applicability of the model in the revised version. However, the model will still be focused on "one" productive year. We will discuss limitations of this hypothesis in the discussion section.

**RC6:** Surprising the authors do not mention climate change as a driver of concern or a factor affecting damage costs and responses. This seems an omission given the topic.

**Answer:** Given that the model is focused on "one" productive year, long term effects of climate change are not considered in the model. Anyway, we will add a sentence in the introduction on the importance of climate change in exacerbating future flood damage to agriculture.

**RC7:** Further clarity is required regarding the definition of measurements of damage. A more detailed listing, upfront, of the revenue and cost related parameters would help: these emerge in the case application later on.

**Answer:** A comprehensive list of all revenue and cost related parameters cannot be compiled in the framework of the conceptual model, as most of them, especially those related to costs, are context-specific. Still, we will add examples of such parameters in the description of the conceptual model

**RC8:** A table would be good to summarise the main elements of cost estimation processes /assumptions/ algorithms and where they come from. In the main, the methods draws on published data from Sub-sector models of crop damage or additional costs, such as Agenias et al. What other ones are used to transfers changes in yield, revenue and cost responses?

**Answer:** a table will be added summarising the main elements and sources of revenue and cost estimation processes

**RC9:** Further clarity would help regarding the use of the terms 'turnover' and 'gross profit', ie exactly what is in these terms? They are not universally applied in farm business accounting, where the terms gross output (or gross revenue), gross margin and net margin are often used. (Turnover can for example include sales from previous production periods – just to be clear). And the definition of gross profit may or may not include elements of farm level fixed costs, such as machinery and buildings costs (again to be clear, so that the methods can be generally applied). The use of 'relative' Gross profit measured at negative % values is difficult to interpret and doesn't mean a lot.

**Answer:** we really thank the referee for the suggestion. According to the literature suggested, we will change the terms "gross profit" and "turnover" in "net margin" and "gross output", respectively, by also specifying what is included in production costs.

**RC10:** On flood scenarios, the treatment presumably here is for one-off relatively infrequent flooding on a land use that is not hitherto constrained by flood exposure. An increase in flood frequency, associated with climate change for example, or withdrawal of flood defences, could lead to increased flooding with a range of outcomes, permanent abandonment, repeat annual losses or a switch to more flood tolerant land use. How are these to be handled by the model?

**Answer:** As explained before (RC2) the model considers damage to one productive year for infrequent floods. Limits of these assumptions will be discussed in the new version of the paper.

**RC11:** The paper refers to spreadsheets and supplementary data containing both data and estimation methods. I had difficulty locating these and understanding them when I did. This is probably my fault. It would be good to explain what is in them and how they can be reliably accessed.

**Answer:** We will ask the NHESS editorial support office whether the spreadsheet can be added as supplement material; otherwise, we will upload it in a repository with an easier access We will also develop a user manual for it.

**RC12:** There is a need to strengthen the treatment of inherent variation and uncertainty in the estimates.

**Answer**: We will include a deeper discussion on model uncertainty in the discussion section.

**RC13:** there is a need to provide a more systematic critique of the model and the resultant damage estimates, and implications for use and improvements

**Answer:** see answer to RC2, RC3 and RC12

**RC14:** The authors report that their work draws on systematic review of multiple sources, including expert judgement. This aspect, especially the latter, is under reported. Did the research approach follow a particular methodology that can be supported by literature, especially engaging experts?

**Answer:** Experts were involved with two main objectives. The first one is to support the definition, and validate the quality, of the conceptual model. The second one is to give suggestions/information on the implementation of the model in the Po Valley, above all regarding expected physical damage and costs. With respect to the first objective, an iterative process was followed. First, a semi-structured interview was conducted, by asking experts about the main damage mechanisms/phenomena in case of flood, possible interconnections among them, important explicative variables. In this phase, results from the literature review were proposed to experts for their judgment. In the following step, experts were asked to evaluate a draft version of the conceptual model we draw according to the literature review and results from first interviews. Then, there was an iterative revision process of improved versions of the model until an agreement on its final structure was reached.
With respect to the second objective, several individual meetings were organised with the aim of asking experts about context-specific information on: crops calendars, yields and prices, type, timing and costs of cultivation practices. In this phase, the transferability of the model by Agenais et al. was also discussed.
Three kinds of experts were involved. One representative of the Regional Authority responsible for agricultural damage management and compensation, with more than 20 years of expertise in the management and compensation of flood damage to farms in the Lombardy Region. Two agronomists of the local association of farmers (Coldiretti Lodi), with specific knowledge on the investigated context and with direct experience in managing floods in the last 20 years. During the work, the two agronomists asked for data/information also to individual local farmers that were flooded in the past years, including also their viewpoint in the process. Finally, an academic economist, with specific expertise in agriculture, has been involved in validating the final model.
A new sub-section will be added to the revised version of the paper explaining the whole process of experts' involvement.

**Specific comments** (we noted that for some comments the reviewer made a wrong reference to page/line number of the original manuscript; in the following, we made our best to locate the comments in the proper point of the paper)

**RC15:** Abstract. I think the abstract would better begin with a statement of context and purpose, and how the proposed model seeks to make a contribution to decision support. I think it best to avoid giving the paper an identity by using 'this paper….' as a writing style here and in the manuscript itself ; it is the authors who are reporting their work. As above, I think some cautious modesty would be advisable. CBA implies welfare assessment. Farmer decision support is something else.

**Answer:** We will revise the abstract according to reviewer's suggestion. We would like to maintain the impersonal writing style within the paper as there are not specific indications about in the Journal guidelines, or a common trend in published articles. However, in this respect, we are available to follow editor's suggestions, if any. See also reply to general comments RC2-RC3.

**RC16 (P1.L20):** What are flood risk management plans, and what is the implication of CBA ?. This implies public investment at the landscape scale, often funded through the public purse, as implied by CBA.

**Answer:** According to the EU Floods Directive, Flood Risk Management Plans are the operational/normative tools by which Member States (and in particular River District Authorities within each State) must implement flood risk management, including a blend of structural and non-structural risk mitigation strategies, to be implemented at different spatial and temporal scales. Such measures must be identified on the bases of a reliable and comprehensive assessment of costs and benefits associated to alternative strategies. We will better clarify this point in the revised manuscript.

**RC17:** I would avoid, 'in this paper', here and elsewhere.

**Answer:** See reply to RC15.

**RC18 (P1.L25):** River restoration usually implies rejoining the river to its floodplain and set back of (previously installed) flood defences in the conventional sense. See:

*Morris J, Bailey AP, Lawson CS, Leeds-Harrison PB, Alsop D, Vivash R (2008) The economic dimensions of integrating flood management and agri-environment through washland creation: A case study from Somerset, England. J Environ Manage 88:372-381*

*Rouquette JR, Posthumus H, Morris J, Hess TM, Dawson, QL, Gowing DJG (2011) Synergies and trade-offs in the management of lowland rural floodplains: an ecosystem services approach. Hydrol Sci J 56(8):1566-1581*

Is the context to justify of guide decisions in flood risk management infrastructure and operations made at the landscape/sub catchment/shoreline scale , with support from the public purse. This is the case in many parts of northern Europe and north America. Getting a handle on damage costs to agriculture is part of this ?

**Answer:** We thank the reviewer for the suggested references, which support our statement regarding the importance of including damage costs to agriculture when dealing with floodplains devoted to agricultural activities. The two references will be included in the revised version of the paper.

**RC19 (P1.L29):** I think this is partly reflecting a limitation of the use of selected key literature search terms and also confinement to formal academic, rather than grey literature and institution-based activities and outputs. There is a history here in this topic: Since the 1930s, and probably up to the mid1980s, the focus in this area in northern Europe was on 'land drainage' of which flood protection , (rather than 'flood risk management'), was a part. Major investments, including large scale pumping

schemes, were made to control /remove excess soil water and simultaneously alleviate surface flood from river, tidal and shore line sources. Many of these investments were 'land reclamation (for agric) projects' often involving major river works (and not river restoration). Thus land drainage and flood control were and are inextricably integrated (just as irrigation and drainage are). The authors should in my view show an understanding of this nexus, and consider how, without undermining what they have done, it can be incorporated here. Including the terms agricultural/land drainage in the search would go some way towards this, as would 'flood risk'. Much of the work was carried out by research institutions as part of national programmes and is reported in sources that are not as easy to access.

A bit dated, but see for example, *Morris, J. 1992. Agricultural land drainage, land use change and economic performance: Experience in the UK. Land Use Policy Volume 9, Issue 3, July 1992, Pages 185-198.*

And for decision support: *See Chapter 9 Flood Risk Management for Agriculture, in: Penning-Rowsell, E., Priest, S., Parker, D., Morris, J., Tunstall, S., Viavattene, C., Chatterton, J. and Owen D. (2013) Flood and Coastal Erosion Risk Management: A Manual for Economic Appraisal, Routledge, Abingdon, Oxford*

**Answer:** Thank you. In the revised version of the manuscript we will try to expand the literature review, by including (both academic and grey) references dealing with the issue of land drainage.

**RC20 (P2.L5):** I think also there has been a policy shift, especially in Europe post 1980s when agricultural surpluses increased under EU CAP and the subsides to agric were being challenged , and urban flood damage increased in absolute as well as relative importance. Also the drainage link is important here : the emphasis in Europe and N America was on drainage land and reclamation.

**Answer:** Thank you for this comment. We will include these points in the revised version of the manuscript.

**RC21 (P2.L15):** Some of the comments here seem premature: we haven't yet explained the approach and the model, but seem to be drawing conclusions , unless these are objectives . The authors might want to consider a clear statement of the objectives of the work reported here, and then subsequently review the extent to which they have been able to meet them

**Answer:** In the lines indicated by the reviewer we briefly introduced AGRIDE-c, the need for its model structure and its usefulness. In order to avoid ambiguity, we propose to rephrase P2.L17-19 as follows: "While the model structure aims to be generally valid, the analytical expression of its components must necessarily be specific to the local physical characteristics of the area as well as to the standards of the agricultural practices and to the type of crops under analysis, given the large variability characterising the agricultural sector".

**RC22 (P2.L25):** Should table 1 be part of methods ? What of 'flood risk' and 'drainage' as key search terms ? And using experts to identify sources ?

**Answer:** We will include "flood risk" and "drainage" as key search terms and modify the results in Table 1 accordingly. However, we think Table 1 should not be moved in the methodological section of the paper, as we used it only to support our preliminary statement on the need to improve damage modelling for the agricultural sector. Experts were not involved in this literature research (almost all of them were not academics), but they were interviewed for model development and assessment (see also response to RC14).

**RC23 (P3):** Would be good to clarify the perspective and purpose of the assessment of damage costs: ex ante or ex post, and the implications : the term ex post is used later without explanation.

**Answer:** In the Introduction of the revised version of the manuscript we will better explain that AGRIDE-c is a tool for an ex-ante (i.e. expected) estimation of flood damage to agriculture, while we will replace the term "ex-post" within the paper with "observed" or "empirical".

**RC24 (P3.L20):** Agree there is paucity of data on actual flood impact costs , recorded during and post flood. This observation is not confined to the agricultural sector (Chatterton et al, for the English cases for example, including agricultural damage)

*Chatterton, J; Clarke, C; Daly, E; Dawks, S; Elding, C; Fenn, T; Hick, E; Miller, J; Morris, J; Ogunyoye, F; Salado R. .2016. The costs and impacts of the winter 2013 to 2014 floods. Report SC140025/R1. Environment Agency, Bristol. http://rpaltd.co.uk/uploads/report_files/the-costs-and-impacts-of-the-winter-2013-to-2014-floods-report.pdf*

There is a large, albeit now dated literature on drainage/water logging impacts on agricultural production that should be referred to, with modelling of the link between soil- water, crop growth and yields, and particularly linked to water level management in the context of land drainage and associated flood control measures.

**Answer:** Thank you. We will add the suggested reference in the revised version of the manuscript. In addition, we will include aspects related to drainage and waterlogging impacts in an enhanced version of the conceptual model represented in Figure 2. See also answer to comment RC4

**RC25:** See Chapter 9, section 9.5, p336 in Penning-Rowsel, opcit. For FLOODFARM, that assesses the cost of flooding at the farm scale Where FLOODFARM = (costs associated with flood impacts on) ARABLE+GRASS+LIVESTOCK+OTHER.

See also: *Dunderdale J A L and Morris J. 1997. The Benefit: Cost Analysis of River Maintenance. Water and Environment Journal. Volume11, Issue 6 Pages 423-430 https://doi.org/10.1111/j.1747-6593.1997.tb01375.x*

**Answer:** We thank the referee for the suggested literature that we will include in the new version of the manuscript. Still, our model is only focused on the crops component of flood damage to farms as explained in answer to RC2.

**RC26 (p.3, L25):** I am not sure the assumption of full loss is true here. The Posthumus, and the Morris and Brewin examples, based on farmers reported assessment of damages, incorporated 'partial' losses, and also losses in the following years. And also on farms adapting to flood risk:

*Pivot J.M., Josien E. & Martin P. Farms adaptation to changes in flood risk: a management approach. J Hydrol 2002, 267, 12–25.*

The ex-ante estimation methods described in Penning Rowsel above, for use in the appraisal of flood investments for agriculture, explicitly build in allowance for seasonal variation in yield loss between different crops (including grass) and livestock.

**Answer:** Thank you. In the revised version of the paper, we will include a comment based on the suggested references.

**RC27 (P7.L29):** Should define Gross profit as gross output minus direct costs. The term Gross Margin is widely used in agricultural /farm business accounting circles. (there is an interesting accounting challenge here: what is considered a direct, avoidable cost in the context of flood impacts, especially when lots of field operations are carried out by contractors)

**Answer:** See reply to comment RC9. All agricultural operations have been considered as direct, avoidable costs and priced based on contractors' price lists for the different operations (experts told us that in Lodi province most of field operations are carried out by contractors). This point will be made clearer in the revision of the manuscript and reference to the price books will be included.

**RC28 (P9.L7):** Is this a tautology ?

**Answer:** The sentence "the first provides information on the physical damage, while the second converts the physical effects of the flood into monetary terms" will be deleted in the revised manuscript.

**RC29 (P9.L9):** Should this be' and/or': with respect to data source, estimation and valuation methods: eg some models have both physical quantities and unit monetary values.

**Answer:** We will replace "and" with "and/or" in the revised paper.

**RC30 (P9.L10):** Implies that this would be good idea? Again need to set in context of the purpose of the 'modelling', high level or detailed assessment ? A number of Environmental bodies use very high level 'cost calculators' to derive quick assessments of flood impacts at the large scale , eg using 'standardised' damage costs $/ha, for example to respond to immediate questions by politicians post flood. There is guidance on this > The UK Environment Agency use a Flood Cost Calculator, European Commission are promoting a standard approach to disaster observation, see for example
*http://publications.jrc.ec.europa.eu/repository/bitstream/JRC110489/loss-database-architecture jrc110489.pdf*

**Answer**: We agree with the reviewer that the required level of detail of a model depends on the context and use. So, not always an ultra-detailed, multi-parameter model can be the best option. We think however that it is not appropriate to comment on this point in the methodological part of the paper, but we will include some comments on this in the discussion section of the revised paper.

**RC31 (P12.L12):** Says Agenais model is physically, presumably yield based , but then says it uses gross profit (gross output (turnover) less direct costs : isn't this monetary (cost) based. Some further clarity of the distinction between physical and monetary estimation would be useful with definition of terms used

**Answer:** We will better explain the Agenais model in the revised version of the manuscript.

**RC32 (P12.L15):** They both imply that duration is probably more important than depth ?

**Answer:** Yes, for the crop under investigation (maize). But this depends on crop type.

**RC33 (P13.L5):** Some more detail on the methods used to define the boundary of investigation, and the methods used to elicit important parameters and values from experts and other sources. Was a formal research method used? Was the research review for example formally a 'systematic' review, and were the experts 'systematically' engaged? Would be good explain how the research topic was framed and bounded , and the issues arising. What is the implication of an expert based approach here? This is an important methodological aspect, and liable to bias that needs to be managed ?

**Answer:** See response to comment RC14.

**RC34:** How is turnover defined . For the purpose here is it Gross Output (Q x P) specifically for the damage to crop outputs in a given period. Turnover in an accounting sense can be something else. Need to explain.

**Answer:** This will be better explained in the revised paper. See reply to comment RC9.

**RC35 (P9.L32):** Need to be explicit on definition of production costs here. Presumably the concern with a costs across the farm business (non revenue items), including replacement and remedial costs, net of savings in uncommitted costs Gross profit is usually after direct costs (or the cost of good sold) , but much depends on how overheads/fixed costs are categorized .How are changes in machinery operating costs, or 'other' damage costs to machinery, buildings and infrastructure being assessed, or are they not included here , given the implied focus on 'field' scale costs?

I think a table to support equation 1 should show the revenue and cost items that are used in the assessment : what is in and what is not ? Lots of jobs are done by contractors : how are these valued ? what of within season reseeding costs, reduction in gross output or profit associated with crop substitution , clean up and remedial works, following year impacts? A list would be good . I see these come later for the Po example, but a classification for the model would be useful; Elements are suggested in figure 1 , but it is not clear which are explicitly measured revenue and cost items

**Answer:** See reply to comment RC2, as AGRIDE-c assesses only damage to annual crops and not to other farm components (machinery, buildings, infrastructure), and RC7 regarding revenue and cost items. Prices of agricultural operations are based on contractors' price lists (experts told us that in Lodi province most of field operations are carried out by contractors and that this would have been the most suitable option for pricing the different operations).

**RC36 (Figure 2):** Useful diagram. Where would salinity fit, and field flooding/waterlogging as it affects field access and timing of operations both within and beyond the immediate flood period? Not all elements are 'valued' in the model

Pri'c'es.

**Answer:** See response to RC2, RC4 and RC5. Figure 2 will be amended with the correct spelling of "prices", thank you for noting that.

**RC37:** Does the model include grassland and associated grassland management and livestock systems? If so, how are flood impacts assessed?

**Answer:** No, AGRIDE-c only estimate damage to crops; this will be better clarified in the revised version of the manuscript (see response to RC2).

**RC38:** A summary of estimation parameters and algorithms would be helpful, possibly linked to the table of estimation items referred to earlier, summarizing the estimation basis . Presumably these are listing in the supporting spreadsheets: I tried but had difficulty accessing. See my comment on the Po case later : the approach is one of 'estimation transfer' . And there are some implicit criteria for transfer that could be made more explicit It would be good to say what is not in there : are damage costs to farm infrastructure, crops in store, included ?

**Answer:** Ok, a summary table will be included in the revised version of the manuscript. See also response to RC2.

**RC38 (P13.L2):** Are there thresholds for assumptions on crop switching/reseeding ?

**Answer:** Yes, this was implicitly reported in Table 3 (alleviation strategy vs month). We will better specify this point in the revised version of the manuscript.

**RC39:** So the scenario is for a single freshwater flood occurring in a given production year ?

**Answer:** Yes, this will be clarified in the revised manuscript (see response to comment RC2).

**RC40 (P11.L5):** Implications of grassland?

**Answer:** Only damage to crop are considered in AGRIDE-c. See response to RC2.

**RC41 (P11.L10):** What year price base is used ? Were annual price series inflation adjusted to a common year ? similarly with costs?

**Answer:** As stated in P11.L11 prices and costs were averaged over the last five years (2013-2017: this will be better specified in the revised paper) and were not adjusted for inflation (negligible change over the considered period).

**RC42 (P11.L15):** 'annual EU contributions for agriculture as a further income for the farmer and, in detail, the subsidies given to agricultural activities in…' Not clear how these are being treated. Presumably farmers get decoupled income support at the farm scale under CAP and these are unaffected by the flood, so can be left out for a single flood event. What of production subsidies: will not these also continue for the year of the flood, so from a farmers viewpoint costs (and cost savings) are net of subsidies?

**Answer:** Experts explained us that EU contributions do not depend on actual production. If a farmer abandons the production of a year due to a flood, he still receives the contribution.

**RC43 (P11.L17):** consultation of regional price books: reference?

**Answer:** Reference to regional price books will be added in the revised manuscript.

**RC44 (P11):** Is the assumption that all the costs shown in Fig 3 are direct costs (and therefore included in Gross profit as defined here) and are potentially 'avoidable' . This might be the case if farmers are using contractors , but if they are using own equipment and labour, how much of these are avoidable costs. Some explanation of the treatment of field operations and related costs would be useful. Some costs are more direct than others. The reference to fixed costs on the next page suggests that most costs are regarded as direct. The estimates are very sensitive to assumptions about the treatment and behaviour of costs : a tricky subject. I don't quite follow: I got E927 using the numbers presented , but there may be other costs not shown. Even so, the gross profit as defined for maize seems high > maize farmers in the Po Valley are doing well.

**Answer:** Yes, all field operations are considered as direct costs and priced based on contractors' price lists See also response to RC35. The reviewer is right in obtaining 927 Euro = 175x16.92 + 150+400 − (175x16.92)*0.05 − (sum of production costs). The results in terms of gross profit reflect the ones observed in the Province, as also confirmed by interviewed local experts and farmers.

**RC45 (P12.L10):** This approach should be more fully explained in describing the model above , that algorithms are judiciously 'transferred' from research applications elsewhere according to suitability/relevance, and availability

**Answer:** In the original manuscript we already stated that "local agronomists expressed a favourable opinion on the suitability of this model in the examined region". However, for more clarity, this point will be stressed in the new section regarding expert involvement in model development/validation (see also response to comment R14).

**RC46 (P12.L16):** Delete first 'nor'

**Answer:** Ok, thanks.

**RC47 (P12.L25):** According to regional price books, restoration costs have been estimated to be equal to 500 €/ha (see Table 3). Would be good to reference these sources: Were contractors contacted? These seem very high unit costs . As for that matter do field operating costs , eg Harvesting at almost E800 /ha?

**Answer:** See previous response to comments regarding reference to contractors' price lists and experts' opinions.

**RC48 (P12.L25):** So the damage to soil box in Figure is aspirational?

**Answer:** Yes. See also response to RC1 of Reviewer 1.

**RC49 (P14.L10):** I am surprised that a yield (and possibly price) penalty is not included in the assessment of reseeded crops, given the importance of timing of operations. There are generic yield functions available for timeliness that would support a relative estimate of yield and gross output loss. This is one topic where local experts and farmers would have an empirically based view. The comment about variation and uncertainty in the estimates is valid for the modelling as a whole, and should be made as part of the method critique

**Answer:** In the original version of the manuscript we did not consider a yield reduction for late planting in case of reseeding, because interviewed experts told us that this is very variable and dependent on many factors (among others, type of late hybrids used) and difficult to estimate based on few parameters. However, in the revised version of the paper and of the model, we will comment on this point and we will also include simplified functions for yield reduction (possible range, max and min) based on experimental results reported in the literature, as in Tsimba et al. (2013), in order to take into the possible effect of yield reduction on the results.

*Tsimba, R., Edmeades, G. O., Millner, J. P., & Kemp, P. D. (2013). The effect of planting date on maize grain yields and yield components. Field Crops Research, 150, 135-144.*

**RC49 (P15.L16):** Break stage? There is no crop in the field? Presumably also depends on crop rotation .

**Answer:** Yes, we are considering only a single crop type in field. This will be better explained in the revised version of the manuscript (see also response to comment RC2).

**RC50 (P15.L22):** In my view gross output or gross revenue would be a better term than turnover, throughout . (Turnover refers to total sales in a period, sales may include items from other production periods)

**Answer:** Ok. See response to RC9.

**RC51 (P15.L25):** Seems unlikely that there would be no yield penalty for delayed planting Furthermore, reseeding would probably not be feasible immediately post flood because of field conditions . Penalty delay functions could be used .

**Answer:** See response to comment RC49.

**RC52 (P15.L30):** Finally?

**Answer:** Ok, thanks.

**RC53 (P16. Figure5):** Would be good to make the axes consistent amongst the graphs , and for cost and turnover estimates. Would also be good to indicate net margin (or gross profit) , although this might complicate the graph. If a read it correctly, for a june flood, reseeding will not make sense , especially if there is (likely) yield penalty: I note for this graph the two 'y' scales are common

**Answer:** Figure 5 will be amended in the revised manuscript by taking into consideration reviewer's suggestions.

**RC54 (P17.L9):** This raises the question about likely average annual damaged according to the likelihood of a flood occurring within given months: where information is available on annul flood probability, and seasonal distribution, and to complicate further, whether seasonal distributions vary according the severity of the flood ? I see this is raised later

**Answer:** Yes. The importance of knowing "seasonal" return periods of floods is commented in the Discussion section.

**RC55 (P16. Figure5):** Is this really a table. The title does not explain that it is relative gross profit : this is difficult to interpret when the preceding assessment was made with respect to turnover ad costs, so some clear explanation is required. Is a relative loss of gross profit greater than 100% a helpful measure?

**Answer:** In the revised manuscript, the caption and the text referring to Figure 5 will be more explicative.

**RC56 (P17.L15):** The use of the term CBA needs explanation: it implies public choice and assessment of welfare change associated with public investments .

**Answer:** See response to comment RC3.

**RC57 (P17.L16):** Quite consolidated practices. Meaning

**Answer:** We mean that cost assessment in CBA is not very problematic, as all cost data can be easily determined. We will better explain this in the revised paper.

**RC58 (P17.L18):** limited to the direct avoided damage to people and some exposed items . this is not clear

**Answer:** We will change this sentence as follows: "CBAs currently consider only direct avoided damage to people and some exposed sectors as benefit of a flood mitigation measure".

**RC59 (P17.L24):** The points here are not clear. I suggest the whole paragraph might be recrafted to advantage, with some examples to support the argument

**Answer:** This will be better explained in the revised manuscript, by also including references supporting the statement of the necessity of using different sets of damage models for more reliable CBAs (e.g. Jongman et al. 2012).

**RC60 (P17.L30):** I am not convinced Figure 6 does this. What does the greater than 100% refer to: is this the gross profit estimate in Figure 6. Assumption of no yield loss with (delayed) reseeding probably underestimates losses . There may be opportunities for reseeding with a different crop, especially between winter sown and spring sown crops

**Answer:** See reply to comment RC9 and RC49.

**RC61 (P17.L30):** Apart from EU contributions? Not clear

**Answer:** With "apart from EU contributions" we meant "if excluding the EU contributions" (which will still be obtained by the farmer). We will make this point clearer in the revised version of the paper.

**RC62 (P17.L30):** Sustained? Already committed/incurred

**Answer:** Ok. This will be fixed in the revised manuscript.

**RC63 (P19.L0-10):** These are valid and critical points, and fundamentally concern the underlying variation and uncertainty in the estimates (that have been single values so far). In my view it would be more appropriate to include the treatment of variation and uncertainty in the description of methods and the presentation of results of the case, rather than raise it for the first time here in discussion, where the purpose is to critical discuss the methods and results.

**Answer:** See reply to general comments regarding the reorganization of the paper. In our opinion, this specific point on variability and uncertainty of estimations should remain in the Discussion section of the paper, as here, after the presentation of model structure, description of input parameters and data, we make comments of strengths and weaknesses of the adopted approach.

**RC64 (P19.Fig7):** This is results and should go there above. The figure is presumably for the Po case? The likely effect of a 10% penalty that would most likely arise due to (delayed) planting is apparent : negative gross profit. A figure showing absolute changes in gross profit (as defined here ) might be useful in the results section.

**Answer:** See reply to general comments regarding the reorganisation of the paper.
Yes, the results in the Figure refer to the Po case: this will be better clarified in the figure caption. See also response to comment RC49. Figure 7 will be amended in order to show also changes in gross profit (i.e. net margin in the revised paper, after reviewer's indication).

**RC65 (P19.L16) :** Rather than saying 'must' it would be better to say why, identifying the advantage of doing so.

**Answer:** We explained the need of developing rapid approximate methods just in previous lines of the original manuscript (P19.L11-16): "The development of AGRIDE-c highlighted some challenges for the hydrology and the hydraulic community. In fact, application of the model requires a relatively detailed set of hazard input variables which are often not supplied in existing flood hazard maps (de Moel et al., 2009). Such knowledge would require a shift from traditional 1D steady hydraulic models to 2D unsteady hydraulic models - coupled with suitable sediment and contaminant transport models - in all flood prone areas, which is not easily achievable in a short time, both for technical and economic constraints".

**RC66 (P19.L18):** Perhaps rather than 'no more' , 'not only…. but also' seasonal probabilities Is the Morris and Hess ref 1988?

**Answer:** The reviewer is right. The sentence and year of Morris and Hess paper will be fixed in the revised manuscript.

**C67:** This paper? The reference to the spreadsheet and to supplementary data needs further support : these are mentioned in passing

**Answer:** "This paper": it is a writing style preference, as already discussed in replies to previous comments. In revising the paper, we will better emphasize reference to the spreadsheet and supplementary data.

**RC68 (P20.L5):** It depends how far the Authors have looked, and with the information presented here it is difficult to judge whether they can substantiate the claim. It might be fair to say they see advantage in developing a generic framework that can potentially be applied across different geographical and economic contexts , and they have made progress in this respect. For example, in more temperate part of Europe, land drainage is a particularly critical component of the land use: flooding nexus, and is particularly critical during post flood periods .

**Answer**: We will revise the sentence according to reviewer's suggestion. The issue of land drainage will be discussed as well (see reply to general comments).

**RC69**: It would be useful to have a description of the sub models used , as referred to earlier. A summary table showing the estimation methods and sources would be particularly helpful, linked to supplementary data.

**Answer:** See answer to RC8.

**RC70**: Damage mechanisms- Meaning ? Drainage and soils might be important also. And also salinity issues in coastal areas, as referred earlier.

**Answer:** With damage mechanisms we mean the interaction between damage influencing factors and characteristics of exposed elements leading to a loss. We will better clarify the sentence in the revised manuscript. Issues related to soil drainage and salinity will also be included (see answer to general comment RC4-RC5).

---

## Author Response (AR1)

**Manuscript nhess-2019-61 "AGRIDE-c, a conceptual model for the estimation of flood damage to crops: development and implementation" – Point by point response to referee 1 comments**

We would like to thank the referee both for his appreciation of our paper and for the work he did on our manuscript; we greatly appreciate his comments as they may contribute to increase the manuscript robustness and, in general, to improve its quality and readability. In the following, we supply a point by point answer to the general and specific comments raised by the referee.

General comments:

**RC1:** Please briefly discuss and justify the consideration of the element "damage to soil" in the model framework against the background that no approach to estimate this damage type as yet exists. From a theoretical point of view the implementation of this damage type is fully comprehensible and reasonable as 1) it ensures a comprehensive view of potential consequences of flooding in the agricultural sector, and 2) damage to soil can significantly contribute to overall flood damage in this sector. However, from a practitioners perspective, the fact that the consideration of damage to soil is suggested on the one hand, but no concrete approach for such an estimation is provided (since not existent) on the other hand, can cause ambiguities. Further, a consideration of damage to soil in the model application using rough assumptions and proxies for this variable could introduce noise to the overall loss estimation rather than valuable information.

**Answer:** We thank the reviewer for this comment and we fully agree with him on this point. Indeed, our choice to include the "damage to soil" component in AGRIDE-c, although in a simplified way, was driven by the two main reasons also raised by the reviewer: comprehensiveness of model structure and importance of this sub-component in the overall flood damage figure to agriculture; in particular, this last point clearly emerged during the interviews with local experts, who pointed out the occurrence of such damages even for flood events characterised by shallow water depths and not particularly high flow velocities. We have included these considerations in Section 4.4 in order to justify the necessity of modelling this sub-component and we will also include in Section 5 a critical discussion of possible impacts of the modelling assumptions on the overall loss estimation.

**RC2:** The AGRIDE-c spreadsheet plays a central role in the model concept. It is currently provided to the reader via a hyperlink to a project website in Italian language. Due to language constraints of non-Italian-speakers as well as potential expiry of the hyperlink I suggest to additionally provide the spreadsheet in the supplement of this paper, if technical requirements of NHESS can be met or bypassed (Excel sheets cannot be uploaded to NHESS supplements). This would ensure unlimited availability and better access of the spreadsheet.

**Answer:** The NHESS editorial support office confirmed that spreadsheets cannot be uploaded as supplement material. For this reason, we have created an open folder including the spreadsheet and a new developed user manual. The tools are now easily accessible at: https://tinyurl.com/yyj2arhp

Specific comments:

**RC3:** Page 2, l. 6-8: The given characteristics of limited model transferability and applicability are not exclusive for agricultural sector, but rather represent general difficulties in flood damage modeling, i.e. often also apply to models for e.g. the residential or the commercial sector. I suggest to rephrase the sentence to avoid the impression that these aspects are exclusive problems of agricultural models.

**Answer:** We agree with the reviewer that the transferability of damage models represents a general issue in flood damage modelling, affecting all exposed sectors. Agriculture is probably one of the most

critical in terms of transferability, due to large variability of the features affecting damage mechanisms for this sector. For more clarity, in the new version of the manuscript we have revised the original L.6-8 in P.2 as follows: "Available damage models for agriculture are not only few in number, but are also affected by many limitations, the major being the paucity of information/data for their validation and the large variability of the local features affecting damage (i.e. the strong linkage with the context under investigation) which limit their transferability to different contexts more than other exposed sectors, as the residential and commercial ones".

**RC4:** Page 2, l. 23: "The paper is organized in four parts" is a bit confusing. To match this number, the exclusion of the sections "introduction" and "conclusion" is required. Moreover, in the subsequent sentences you list five different sections. Please rephrase the sentence towards a more unambiguous statement. For example, "the paper is organized as follows".

**Answer:** The reviewer is right. We have revised the sentence as suggested in the new version of the manuscript

**RC5:** Page 3, l. 1: "The main available damage models [: : :]". This statement is unclear to me. Do you mean "prominent examples of damage models"? Please clarify

**Answer:** Yes, we do. We have revised the sentence as suggested in the new version of the manuscript.

**RC6:** Page 9, l. 27-30: Although in a European context floods usually have a negative effect on soils, the studies of e.g. Hein et al. (2003) and Tockner et al. (1999) show that such events can also have clearly positive effects, namely in the form of an increase of soil fertility. The fertility increase is explained by a (re-)distribution of river sediments and organic matter in the course of flooding. These river sediments replenish carbon and nutrients in topsoil and, hence, can make agricultural lands more fertile. I suggest to briefly discuss this aspect in the paper. An adaptation of Figure 2, where the box "damage to soil" currently states only the negative effect of flooding, could also be considered.

**Answer:** We thank the reviewer for this important comment. In the original version of the manuscript we only referred to negative flood effects on soils, because in Italy these are the most common impacts observed from past events. However, we fully concur with the reviewer on the importance of including also positive effects (e.g. increase of soil fertility) in the general conceptual model represented in Figure 2, which has been revised accordingly. In the revised version of the manuscript, we have also included some discussion on this point in subsection 4.4 (in addition, the title "Damage to soil" has been changed to "Impact on soil" in order to be more comprehensive).

**Technical corrections:**

Page 12, l. 16, w. 11: Grammar issue. "nor" should be replaced by "neither".

Page 19, l. 2, w. 13-15: Consider rephrasing "in another terms". For example, into "in other words".

**Answer:** These technical corrections have been fixed in the revised version of the manuscript.

**Manuscript nhess-2019-61 "AGRIDE-c, a conceptual model for the estimation of flood damage to crops: development and implementation" – Point by point response to referee 2 comments**

We would like to thank the referee both for appreciation of our work and for carefully reading our manuscript; we greatly appreciate the insightful comments as they may contribute to increase the manuscript robustness and, in general, to improve its quality and readability. In the following, we supply a point by point reply to the general and specific comments raised by the referee.

**General comments:**

**RC1:** There is scope to improve the structure of the paper:

- by separating the introduction of context and rationale, including statement of research objectives, and statement of methods to cover literature search, review of knowledge and construction of the analytical framework.
- There should be a critique of the approach.
- The case study then becomes results (reordering some results that currently occur in discussion).
- Discussion can then follow on both the case and the validity or otherwise of generic framework.

Some items currently in the conclusion, particularly on gaps/further development, can go in discussion (they appear to be recommendations). Conclusion on what has gone before can focus whether the objectives (regarding the tool, its application and its prospects) have been met, rather than introducing new elements into discussion.

**Answer:** We tried to address at our best the suggestions provided by the referee, especially regarding a better ri-organisation of the contents of the introduction/methodological part, and discussion/conclusions. Still we think that the present organisation of the sections is appropriate for the explanation of the conceptual model and its exemplification. Of course, if the editor thinks that a change in the structure of the paper is required to better meet the journal standards, we will re-organise the paper as suggested.

**RC2:** It probably would be best to be more cautious and modest about the claims made about the comprehensiveness and novelty of the approach, and its suitability to all circumstances and contexts.

**Answer:** After reading the referee comment, we realised that the "scope of use" of our model was not well specified in the original version of the paper, which could lead to incorrect interpretations of our work. Indeed, without such specifications our claims appear as too wide and, therefore, we have been more specific in the revised version of the manuscript in presenting the conceptual model of AGRIDE-c. In fact, the conceptual model has been designed to supply an estimation of flood damage:

- to annual crops (i.e. not including perennial plants)
- by considering one single culture (i.e. by not considering replacement of one culture with another one)
- by limiting the time frame of the analysis to one "productive cycle" (i.e. not considering long term damages, e.g. loss of soil productivity in the following cycle/years);
- for infrequent flooding (i.e. effect of two, or more, consecutive floods is not considered)

Nonetheless, as specified at page 9 line 11-19 of the original manuscript, AGRIDE-c do not consider damage to other components/elements of the farm that may induce additional damage to crops, as, for instance, damage to machineries and equipment (e.g. the irrigation plant) that may prevent cultivation for a while. Only damage to soil is considered from the evidence that, during a flood, damage to soil and plants occurs always at the same time, differently from damage to the other components which can

occur or not, independently from the damage to plants; as a consequence, damage to soil and plants is modelled together, while damage to the other components could be modelled as separated factors, not included in the conceptual model.

We have specified all these aspects in the new version of the paper (Section 3), by hopefully clarifying what we mean with "generality" and "transferability" of the approach. We never referred in the paper to "novelty" or "comprehensiveness" of the approach, but we highlighted the fact that we are trying to encapsulate and systematise the available knowledge on damage mechanisms (to annual crops) triggered by inundation phenomena, as well as on their consequences in terms of income for the farmers.

**RC3**: Further clarity on its potential application, either in cost benefit analysis of (publically funded) investments at the landscape scale in flood risk management, or in guiding individual farm-scale responses would be appropriate. The two applications are different in purpose and detail of approach. There is a difference between, for example, economic and financial appraisals. There is also a difference between ex ante appraisal and ex post evaluation, which is implied. This will support the important point made that insufficient ex post evaluation is undertaken to provide sound ex ante decisions

**Answer:** The CBA of flood risk mitigation strategies would require a comprehensive estimation of benefits associated to the adoption of different strategies, i.e. of the avoided loss to all exposed sectors and at different temporal scales (i.e. direct and indirect/long term damages). However, present damage modelling capacity is mainly focused on direct damage to people and some exposed assets (typically residential buildings), thus preventing the possibility of performing comprehensive flood damage assessments. In such a context, by allowing the estimation of the expected loss to crops in a specific flood scenario, AGRIDE-c may support more comprehensive CBAs of public risk mitigation strategies. Clearly, to meet such an objective, the tool must be critically used, e.g. by considering possible transfers of losses/gains between farmers in an economic perspective, according to the temporal and spatial scales of the analysis. Regarding individual responses, by supplying the expected damage for different types of crops and alleviation strategies (according to the expected yield reduction for different flood intensities and period of occurrence), AGRIDE-c may support individual farmers exposed to flood risk in preventing losses by supporting: the choice of the most resilient crops to be cultivated, the choice of the best alleviation strategy to be followed once flooded, the evaluation of the opportunity to ask for a flood insurance scheme and the definition of the premium. The model was not designed to be used ex-post. This explanation has been included in the Introduction and Discussion sections of the revised manuscript.

**RC4:** One particular issue requires attention, namely the importance, especially in temperate climates, of agricultural land drainage. The control of water levels in the soil, and particularly the removal of excess water and below surface 'flooding', including during the post flood phase before field return to 'normal' is an important aspect of agricultural flood risk management and assessment. Impacts and land management responses are often driven by seasonal waterlogging and drainage problems as much as they are by surface flooding. This is certainly the case in northern Europe and North America. There should be coverage of this aspect, and the implications of not explicitly allowing for it in this model framework. Many areas of strategic importance are pump drained.

**Answer:** We thank the referee for highlighting these important aspects for flood risk/damage assessment and management (that were erroneously not included in the original version of the paper), and for supplying some of the related literature. In the new version of the manuscript, the conceptual model has been modified in order to take into account of the effect of waterlogging on both the yield and the soil, as well as on the corresponding effect on the revenue and costs for the farmer, and then

on the final flood damage (see Figure 2 and Section 3). Still, these aspects were not considered in the application of the model to the Po Valley because, according to experts' opinion, waterlogging after floods is not common in the area.

**RC5:** Saline flooding, a major issue in coastal and tidal areas, should be referred to with implications for costs, especially regarding remediation and subsequent year impacts

**Answer:** The conceptual model was originally not conceived to cover coastal floods but we have decided to extend the context of applicability of the model in the revised version, by including water salinity load among the hazard parameters, and salinization as possible effect on soil (see Figure 2 and Section 3). However, the model will still be focused on "one" productive cycle/year. We have discussed limitations of this hypothesis in the discussion section.

**RC6:** Surprising the authors do not mention climate change as a driver of concern or a factor affecting damage costs and responses. This seems an omission given the topic.

Given that the model is focused on "one" productive cycle, long term effects of climate change are not considered in the model. Anyway, we have included a paragraph in the introduction on the importance of climate change in exacerbating future flood damages.

**RC7:** Further clarity is required regarding the definition of measurements of damage. A more detailed listing, upfront, of the revenue and cost related parameters would help: these emerge in the case application later on.

**Answer:** A comprehensive list of all revenue and cost related parameters cannot be compiled in the framework of the conceptual model, as most of them, especially those related to costs, are context-specific. Still, some examples were already included in the description of the conceptual model (e.g. yield and price of the crops regarding the revenue, and soil restoration and reseeding regarding the costs) and we added more in the new version of the paper (e.g. land drainage costs). A detailed description of all parameters have been instead supplied with reference to the case study (see also answer to RC8).

**RC8:** A table would be good to summarise the main elements of cost estimation processes /assumptions/ algorithms and where they come from. In the main, the methods draws on published data from Sub-sector models of crop damage or additional costs, such as Agenias et al. What other ones are used to transfers changes in yield, revenue and cost responses?

**Answer:** A table have been added (Table 3 in the revised version of the manuscript) that summarises the main input data required by AGRIDE-c and its exemplification in the Po Plain

**RC9:** Further clarity would help regarding the use of the terms 'turnover' and 'gross profit', ie exactly what is in these terms? They are not universally applied in farm business accounting, where the terms gross output (or gross revenue), gross margin and net margin are often used. (Turnover can for example include sales from previous production periods – just to be clear). And the definition of gross profit may or may not include elements of farm level fixed costs, such as machinery and buildings costs (again to be clear, so that the methods can be generally applied). The use of 'relative' Gross profit measured at negative % values is difficult to interpret and doesn't mean a lot.

**Answer:** we really thank the referee for the suggestion. According to the literature suggested, we have changed the terms "gross profit" and "turnover" in "net margin" and "gross output", respectively. As specified in the revised manuscript (Section 4.2 and new Table 3), all agricultural operations have been

considered as direct, avoidable costs, as interviewed local experts indicated that in Lodi province most of field operations are carried out by contractors.

**RC10:** On flood scenarios, the treatment presumably here is for one-off relatively infrequent flooding on a land use that is not hitherto constrained by flood exposure. An increase in flood frequency, associated with climate change for example, or withdrawal of flood defences, could lead to increased flooding with a range of outcomes, permanent abandonment, repeat annual losses or a switch to more flood tolerant land use. How are these to be handled by the model?

**Answer:** As explained before (RC2) the model considers damage to one productive cycle for infrequent floods. Limits of these assumptions have been discussed in the new version of the paper (Section 5).

**RC11:** The paper refers to spreadsheets and supplementary data containing both data and estimation methods.  I had difficulty locating these and understanding them when I did. This is probably my fault. It would be good to explain what is in them and how they can be reliably accessed.

**Answer:** Because the journal does not allow to upload spreadsheets as supplementary material, we have created an open folder including the AGRIDE-c spreadsheet and a new developed user manual. The tools are easily accessible at:  https://tinyurl.com/yyj2arhp

**RC12:** There is a need to strengthen the treatment of inherent variation and uncertainty in the estimates.

**Answer**: We have included a deeper discussion on model uncertainty in the discussion section.

**RC13:** there is a need to provide a more systematic critique of the model and the resultant damage estimates, and implications for use and improvements

**Answer:** see answer to RC2, RC3 and RC12

**RC14:** The authors report that their work draws on systematic review of multiple sources, including expert judgement. This aspect, especially the latter, is under reported. Did the research approach follow a particular methodology that can be supported by literature, especially engaging experts?

**Answer:** Experts were involved with two main objectives. The first one is to support the definition, and validate the quality, of the conceptual model. The second one is to give suggestions/information on the implementation of the model in the Po Valley, above all regarding expected physical damage and costs. With respect to the first objective, an iterative process was followed.  First, a semi-structured interview was conducted, by asking experts about the main damage mechanisms/phenomena in case of flood, possible interconnections among them, important explicative variables. In this phase, results from the literature review were proposed to experts for their judgment. In the following step, experts were asked to evaluate a draft version of the conceptual model we draw according to the literature review and results from first interviews. Then, there was an iterative revision process of improved versions  of the model until an agreement on its final structure was reached.
With respect to the second objective, several individual meetings were organised with the aim of asking experts about context-specific information on: crops calendars, yields and prices, type, timing and costs of cultivation practices. In this phase, the transferability of the model by Agenais et al. was also discussed.
Three kinds of experts were involved. One representative of the Regional Authority responsible for agricultural damage management and compensation, with more than 20 years of expertise in the management and compensation of flood damage to farms in the Lombardy Region. Two agronomists of the local association of farmers (Coldiretti Lodi), with specific knowledge on the investigated context

and with direct experience in managing floods in the last 20 years. During the work, the two agronomists asked for data/information also to individual local farmers that were flooded in the past years, including also their viewpoint in the process. Finally, an academic economist, with specific expertise in agriculture, has been involved in validating the final model.

This information explaining the whole process of experts' involvement has been included in Section 3 and 5 of the revised manuscript.

**Specific comments** (we noted that for some comments the reviewer made a wrong reference to page/line number of the original manuscript; in the following, we made our best to locate the comments in the proper point of the paper)

**RC15:** Abstract. I think the abstract would better begin with a statement of context and purpose, and how the proposed model seeks to make a contribution to decision support. I think it best to avoid giving the paper an identity by using 'this paper….' as a writing style here and in the manuscript itself ; it is the authors who are reporting their work. As above, I think some cautious modesty would be advisable. CBA implies welfare assessment. Farmer decision support is something else.

We have revised the abstract according to reviewers' suggestion. We would like to maintain the impersonal writing style within the paper as there are not specific indications about in the Journal guidelines, or a common trend in published articles. However, in this respect, we are available to follow editor's suggestions, if any. See also reply to general comments RC2-RC3.

**RC16 (P1.L20):** What are flood risk management plans, and what is the implication of CBA ?. This implies public investment at the landscape scale, often funded through the public purse, as implied by CBA.

According to the EU Floods Directive, Flood Risk Management Plans are the operational/normative tools by which Member States (and in particular River District Authorities within each State) must implement flood risk management, including a blend of structural and non-structural risk mitigation strategies, to be implemented at different spatial and temporal scales. Such measures must be identified on the bases of a reliable and comprehensive assessment of costs and benefits associated to alternative strategies. We have better clarified this point in the revised manuscript (Section 1).

**RC17:** I would avoid, 'in this paper', here and elsewhere.

**Answer:** See reply to RC15.

**RC18 (P1.L25):** River restoration usually implies rejoining the river to its floodplain and set back of (previously installed) flood defences in the conventional sense. See:

*Morris J, Bailey AP, Lawson CS, Leeds-Harrison PB, Alsop D, Vivash R (2008) The economic dimensions of integrating flood management and agri-environment through washland creation: A case study from Somerset, England. J Environ Manage 88:372-381*

*Rouquette JR, Posthumus H, Morris J, Hess TM, Dawson, QL, Gowing DJG (2011) Synergies and trade-offs in the management of lowland rural floodplains: an ecosystem services approach. Hydrol Sci J 56(8):1566-1581*

Is the context to justify of guide decisions in flood risk management infrastructure and operations made at the landscape/sub catchment/shoreline scale , with support from the public purse. This is the case in

many parts of northern Europe and north America. Getting a handle on damage costs to agriculture is part of this ?

**Answer:** We thank the reviewer for the suggested references, which support our statement regarding the importance of including damage costs to agriculture when dealing with floodplains devoted to agricultural activities. The two references have been included in the revised version of the paper.

**RC19 (P1.L29):** I think this is partly reflecting a limitation of the use of selected key literature search terms and also confinement to formal academic, rather than grey literature and institution-based activities and outputs. There is a history here in this topic: Since the 1930s, and probably up to the mid1980s, the focus in this area in northern Europe was on 'land drainage' of which flood protection, (rather than 'flood risk management'), was a part. Major investments, including large scale pumping schemes, were made to control /remove excess soil water and simultaneously alleviate surface flood from river, tidal and shore line sources. Many of these investments were 'land reclamation (for agric) projects' often involving major river works (and not river restoration). Thus land drainage and flood control were and are inextricably integrated (just as irrigation and drainage are). The authors should in my view show an understanding of this nexus, and consider how, without undermining what they have done, it can be incorporated here. Including the terms agricultural/land drainage in the search would go some way towards this, as would 'flood risk'. Much of the work was carried out by research institutions as part of national programmes and is reported in sources that are not as easy to access.

A bit dated, but see for example, *Morris, J. 1992. Agricultural land drainage, land use change and economic performance: Experience in the UK. Land Use Policy Volume 9, Issue 3, July 1992, Pages 185-198.*

And for decision support: *See Chapter 9 Flood Risk Management for Agriculture, in: Penning-Rowsell, E., Priest, S., Parker, D., Morris, J., Tunstall, S., Viavattene, C., Chatterton, J. and Owen D. (2013) Flood and Coastal Erosion Risk Management: A Manual for Economic Appraisal, Routledge, Abingdon, Oxford*

**Answer:** Thank you. In the revised version of the manuscript we tried to explain this nexus as a possible reason of the limited literature on flood damage to agriculture (Section 1).

**RC20 (P2.L5):** I think also there has been a policy shift, especially in Europe post 1980s when agricultural surpluses increased under EU CAP and the subsides to agric were being challenged, and urban flood damage increased in absolute as well as relative importance. Also the drainage link is important here : the emphasis in Europe and N America was on drainage land and reclamation.

**Answer:** Thank you for this comment. We have included these points in the revised version of the manuscript (Section 1).

**RC21 (P2.L15):** Some of the comments here seem premature: we haven't yet explained the approach and the model, but seem to be drawing conclusions, unless these are objectives . The authors might want to consider a clear statement of the objectives of the work reported here, and then subsequently review the extent to which they have been able to meet them

**Answer:** In the lines indicated by the reviewer we briefly introduced AGRIDE-c, the need for its model structure and its usefulness. In order to avoid ambiguity, we have rephrased P2.L17-19 of the original manuscript as follows: "While the model structure aims to be generally valid, the analytical expression of its components must necessarily be specific to the local physical characteristics of the area as well as to the standards of the agricultural practices and to the type of crops under analysis, given the large variability characterising the agricultural sector".

**RC22 (P2.L25):** Should table 1 be part of methods ? What of 'flood risk' and 'drainage' as key search terms ? And using experts to identify sources ?

**Answer:** We think that the use of "flood risk" and "drainage" as key search terms is misleading as it results in different kinds of paper, often not linked to flood damage to agriculture. For this reason, we did not include the results of this research in Table 1 but we commented on the link between the literature on land drainage and that on flood protection in Section 1 (see also response to RC19). We think Table 1 should not be moved in the methodological section of the paper, as we used it only to support our preliminary statement on the need to improve damage modelling for the agricultural sector. Experts were not involved in this literature research (almost all of them were not academics), but they were interviewed for model development and assessment (see also response to RC14).

**RC23 (P3):** Would be good to clarify the perspective and purpose of the assessment of damage costs: ex ante or ex post, and the implications : the term ex post is used later without explanation.

**Answer:** In the Introduction of the revised version of the manuscript we explained that AGRIDE-c is a tool for an ex-ante (i.e. expected) estimation of flood damage to agriculture, while we have replaced the term "ex-post" within the paper with "observed" or "empirical".

**RC24 (P3.L20):** Agree there is paucity of data on actual flood impact costs , recorded during and post flood. This observation is not confined to the agricultural sector (Chatterton et al, for the English cases for example, including agricultural damage)

*Chatterton, J; Clarke, C; Daly, E; Dawks, S; Elding, C; Fenn, T; Hick, E; Miller, J; Morris, J; Ogunyoye, F; Salado R. .2016. The costs and impacts of the winter 2013 to 2014 floods. Report SC140025/R1. Environment Agency, Bristol. http://rpaltd.co.uk/uploads/report_files/the-costs-and-impacts-of-the-winter-2013-to-2014-floods-report.pdf*

There is a large, albeit now dated literature on drainage/water logging impacts on agricultural production that should be referred to, with modelling of the link between soil- water, crop growth and yields, and particularly linked to water level management in the context of land drainage and associated flood control measures.

**Answer:** Thank you. We have included the suggested reference in the revised version of the manuscript. In addition, we included aspects related to drainage and waterlogging impacts in the enhanced version of the conceptual model represented in Figure 2 (see also response to RC4).

**RC25:** See Chapter 9, section 9.5, p336 in Penning-Rowsel, opcit
For FLOODFARM, that assesses the cost of flooding at the farm scale
Where FLOODFARM = (costs associated with flood impacts on) ARABLE+GRASS+LIVESTOCK+OTHER.
See also: *Dunderdale J A L and Morris J. 1997. The Benefit: Cost Analysis of River Maintenance. Water and Environment Journal. Volume11, Issue 6 Pages 423-430 https://doi.org/10.1111/j.1747-6593.1997.tb01375.x*

**Answer:** We thank the referee for the suggested literature that we will include in the new version of the manuscript. Still, our model is only focused on the crops component of flood damage to farms as explained in answer to RC2.

**RC26 (p.3, L25):** I am not sure the assumption of full loss is true here. The Posthumus, and the Morris and Brewin examples, based on farmers reported assessment of damages, incorporated 'partial' losses, and also losses in the following years. And also on farms adapting to flood risk:

*Pivot J.M., Josien E. & Martin P. Farms adaptation to changes in flood risk: a management approach. J Hydrol 2002, 267, 12–25.*

The ex-ante estimation methods described in Penning Rowsel above, for use in the appraisal of flood investments for agriculture, explicitly build in allowance for seasonal variation in yield loss between different crops (including grass) and livestock.

**Answer:** Thank you. We have a included a comment based on the suggested references of the revised version of the paper.

**RC27 (P7.L29):** Should define Gross profit as gross output minus direct costs. The term Gross Margin is widely used in agricultural /farm business accounting circles. (there is an interesting accounting challenge here: what is considered a direct, avoidable cost in the context of flood impacts, especially when lots of field operations are carried out by contractors)

**Answer:** See reply to comment RC9. All agricultural operations have been considered as direct, avoidable costs and priced based on contractors' price lists for the different operations (experts told us that in Lodi province most of field operations are carried out by contractors). This point has been made clearer in the revision of the manuscript and reference to the price books has been included as well.

**RC28 (P9.L7):** Is this a tautology ?

**Answer:** The sentence "the first provides information on the physical damage, while the second converts the physical effects of the flood into monetary terms" has been deleted in the revised manuscript.

**RC29 (P9.L9):** Should this be' and/or': with respect to data source, estimation and valuation methods: eg some models have both physical quantities and unit monetary values.

**Answer:** We have replaced "and" with "and/or" in the revised paper.

**RC30 (P9.L10):** Implies that this would be good idea? Again need to set in context of the purpose of the 'modelling', high level or detailed assessment ? A number of Environmental bodies use very high level 'cost calculators' to derive quick assessments of flood impacts at the large scale , eg using 'standardised' damage costs $/ha, for example to respond to immediate questions by politicians post flood. There is guidance on this > The UK Environment Agency use a Flood Cost Calculator, European Commission are promoting a standard approach to disaster observation, see for example
*http://publications.jrc.ec.europa.eu/repository/bitstream/JRC110489/loss-database-architecture jrc110489.pdf*

**Answer**: We agree with the reviewer that the required level of detail of a model depends on the context and use. So, not always an ultra-detailed, multi-parameter model can be the best option. We think however that it is not appropriate to comment on this point in the methodological part of the paper, but we included some comments on this in the discussion section of the revised paper.

**RC31 (P12.L12):** Says Agenais model is physically, presumably yield based , but then says it uses gross profit (gross output (turnover) less direct costs : isn't this monetary (cost) based. Some further clarity of the distinction between physical and monetary estimation would be useful with definition of terms used

**Answer:** The model implemented in the case study is the physical model included in Agenais et al. (2013), specifically in the annex (pg. 200-202), and reported in Figure 4 of the paper. As explained in Section 4.3, such model supplies an estimation of the relative damage as a percentage of the yield in the Scenario 0. By multiplying this percentage by the crop yield and the unit price of the crops, the reduction in the gross ouput can be calculated. Still, Agenais et al. (2013) includes also absolute damage functions (at pg. 51), at

which Table 1 refers, supplying the absolute damage as the reduction in the net margin, calibrated from the French context. In the new version of the paper, we have specified that we adopted only the physical model of Agenais et al. (2013).

**RC32 (P12.L15):** They both imply that duration is probably more important than depth ?

**Answer:** Yes, for the crop under investigation (maize). But this depends on crop type.

**RC33 (P13.L5):** Some more detail on the methods used to define the boundary of investigation, and the methods used to elicit important parameters and values from experts and other sources. Was a formal research method used? Was the research review for example formally a 'systematic' review, and were the experts 'systematically' engaged? Would be good explain how the research topic was framed and bounded , and the issues arising. What is the implication of an expert based approach here? This is an important methodological aspect, and liable to bias that needs to be managed ?

**Answer:** See response to comment RC14.

**RC34:** How is turnover defined . For the purpose here is it Gross Output (Q x P) specifically for the damage to crop outputs in a given period. Turnover in an accounting sense can be something else. Need to explain.

**Answer:** We have replaced "turnover" with "Gross output" throughout the text of the revised manuscript, with a more detailed definition of the terms used. See reply to comment RC9.

**RC35 (P9.L32):** Need to be explicit on definition of production costs here. Presumably the concern with a costs across the farm business (non revenue items), including replacement and remedial costs, net of savings in uncommitted costs Gross profit is usually after direct costs (or the cost of good sold) , but much depends on how overheads/fixed costs are categorized .How are changes in machinery operating costs, or 'other' damage costs to machinery, buildings and infrastructure being assessed, or are they not included here , given the implied focus on 'field' scale costs?
I think a table to support equation 1 should show the revenue and cost items that are used in the assessment : what is in and what is not ? Lots of jobs are done by contractors : how are these valued ? what of within season reseeding costs, reduction in gross output or profit associated with crop substitution , clean up and remedial works, following year impacts? A list would be good . I see these come later for the Po example, but a classification for the model would be useful; Elements are suggested in figure 1 , but it is not clear which are explicitly measured revenue and cost items

**Answer:** See reply to comment RC2, as AGRIDE-c assesses only damage to annual crops and not to other farm components (machinery, buildings, infrastructure), and RC7 regarding revenue and cost items. Prices of agricultural operations are based on contractors' price lists (experts told us that in Lodi province most of field operations are carried out by contractors and that this would have been the most suitable option for pricing the different operations).

**RC36 (Figure 2):** Useful diagram. Where would salinity fit, and field flooding/waterlogging as it affects field access and timing of operations both within and beyond the immediate flood period? Not all elements are 'valued' in the model

Pri'c'es.

**Answer:** See response to RC2, RC4 and RC5. Figure 2 have been amended with the correct spelling of "prices", thank your for noting that.

**RC37:** Does the model include grassland and associated grassland management and livestock systems? If so, how are flood impacts assessed?

**Answer:** No, AGRIDE-c only estimate damage to crops; this have been clarified in the revised version of the manuscript (see response to RC2).

**RC38:** A summary of estimation parameters and algorithms would be helpful, possibly linked to the table of estimation items referred to earlier, summarizing the estimation basis . Presumably these are listing in the supporting spreadsheets: I tried but had difficulty accessing. See my comment on the Po case later : the approach is one of 'estimation transfer' . And there are some implicit criteria for transfer that could be made more explicit It would be good to say what is not in there: are damage costs to farm infrastructure, crops in store, included ?

**Answer:** Ok, a summary table have been included in the revised version of the manuscript. See also response to RC2.

**RC38 (P13.L2):** Are there thresholds for assumptions on crop switching/reseeding ?

**Answer:** Yes, this was implicitly reported in Table 3 (alleviation strategy vs month) and already specified in the original text of paper (original P12-L30.31).

**RC39:** So the scenario is for a single freshwater flood occurring in a given production year ?

**Answer:** Yes, this has been clarified in the revised manuscript (see response to comment RC2).

**RC40 (P11.L5):** Implications of grassland?

**Answer:** Only damage to crop are considered in AGRIDE-c. See response to RC2.

**RC41 (P11.L10):** What year price base is used ? Were annual price series inflation adjusted to a common year ? similarly with costs?

**Answer:** As stated in P11.L11 of the original manuscript prices and costs were averaged over the last five years (2013-2017: this has been better specified in the new version of the paper) and were not adjusted for inflation (negligible change over the considered period).

**RC42 (P11.L15):** 'annual EU contributions for agriculture as a further income for the farmer and, in detail, the subsidies given to agricultural activities in…' Not clear how these are being treated. Presumably farmers get decoupled income support at the farm scale under CAP and these are unaffected by the flood, so can be left out for a single flood event. What of production subsidies: will not these also continue for the year of the flood, so from a farmers viewpoint costs (and cost savings) are net of subsidies?

**Answer:** Experts explained us that EU contributions do not depend on actual production. If a farmer abandons the production of a year due to a flood, he still receives the contribution.

**RC43 (P11.L17):** consultation of regional price books: reference?

**Answer:** Reference to regional price books has been added in the revised manuscript (APIMA – Associazione Provinciale Imprese di Meccanizzazione Agricola delle Province di Milano, Lodi, Como, Varese: Tariffe 2013-2017 delle lavorazioni meccanico agricole c/terzi).

**RC44 (P11):** Is the assumption that all the costs shown in Fig 3 are direct costs (and therefore included in Gross profit as defined here) and are potentially 'avoidable' . This might be the case if farmers are using contractors , but if they are using own equipment and labour, how much of these are avoidable costs. Some explanation of the treatment of field operations and related costs would be useful. Some costs are more direct than others. The reference to fixed costs on the next page suggests that most costs are regarded as direct. The estimates are very sensitive to assumptions about the treatment and behaviour of costs : a tricky subject. I don't quite follow: I got E927 using the numbers presented , but there may be other costs not shown. Even so, the gross profit as defined for maize seems high > maize farmers in the Po Valley are doing well.

**Answer:** Yes, all field operations are considered as direct costs and priced based on contractors' price lists See also response to RC35. The reviewer is right in obtaining 927 Eur = 175x16.92 + 150+400 – (175x16.92)*0.05 – (sum of production costs). The results in terms of "gross profit" (now defined in the revised paper as "net margin", according to reviewer's suggestion) reflect the ones observed in the Province, as also confirmed by interviewed local experts and farmers.

**RC45 (P12.L10):** This approach should be more fully explained in describing the model above , that algorithms are judiciously 'transferred' from research applications elsewhere according to suitability/relevance, and availability

**Answer:** In the original manuscript we already stated that "local agronomists expressed a favourable opinion on the suitability of this model in the examined region". In the revised manuscript we have stressed this point by including the following sentence "[…] as emerged from discussions held during the interview process".

**RC46 (P12.L16):** Delete first 'nor'

**Answer:** Fixed, thanks.

**RC47 (P12.L25):** According to regional price books, restoration costs have been estimated to be equal to 500 €/ha (see Table 3). Would be good to reference these sources: Were contractors contacted? These seem very high unit costs . As for that matter do field operating costs , eg Harvesting at almost E800 /ha?

**Answer:** See previous response to comments regarding reference to contractors' price lists and experts' opinions. This has been also better clarified in Section 4.4 of the revised manuscript.

**RC48 (P12.L25):** So the damage to soil box in Figure is aspirational?

**Answer:** Yes. See also response to RC1 of Reviewer 1.

**RC49 (P14.L10):** I am surprised that a yield (and possibly price) penalty is not included in the assessment of reseeded crops, given the importance of timing of operations. There are generic yield functions available for timeliness that would support a relative estimate of yield and gross output loss. This is one topic where local experts and farmers would have an empirically based view. The comment about variation and uncertainty in the estimates is valid for the modelling as a whole, and should be made as part of the method critique

**Answer:** For simplicity, in the presentation of results in the original version of the manuscript we did not consider a yield reduction for late planting in case of reseeding (Figure 5). In addition, interviewed experts told us that this is very variable and dependent on many factors (among others, type of late

hybrids used) and difficult to estimate based on few parameters. These considerations are also confirmed by results from the literature (references have been included in the new version of the manuscript – see below). However, as already stated in the original paper, in the AGRIDE-c spreadsheet users have the option to set the most suitable value for the expected yield reduction due to late (re-) planting to take this phenomenon into account. We have clarified this point in the revised manuscript, by also including information on experimental results reported in the literature (generally observed yield reductions: -10% ÷ -30%). In addition, a comment on the possible effect of yield reduction on the results shown in Figure 6 has been included in the revised manuscript: "On the other hand, when flood intensity implies significant yield loss, reseeding (if possible) must be preferred to continuation, limiting the relative damage to 80%; nevertheless, this positive advantage of reseeding over continuation becomes smaller when including a yield penalty for late (re-)planting: results obtained by using the AGRIDE-c spreadsheet indicate a relative damage of 102% and 145% for a yield reduction of 10% and 30%., respectively"

*Abendroth L.J., Woli K.P., Myers A.J., Elmore R.W. (2017) Yield-based corn planting date recommendation windows for Iowa. Crop, Forage & Turfgrass Management, 3(1), 1-7,.*

*Dobor, L., Barcza, Z., Hlásny, T., Árendás, T., Spitkó, T., & Fodor, N. (2016). Crop planting date matters: Estimation methods and effect on future yields. Agricultural and Forest Meteorology, 223, 103-115.*

*Lauer, J. G., Carter, P. R., Wood, T. M., Diezel, G., Wiersma, D. W., Rand, R. E., & Mlynarek, M. J. (1999). Corn hybrid response to planting date in the northern corn belt. Agronomy Journal, 91(5), 834-839.*

*Tsimba, R., Edmeades, G. O., Millner, J. P., & Kemp, P. D. (2013). The effect of planting date on maize grain yields and yield components. Field Crops Research, 150, 135-144.*

**RC49 (P15.L16):** Break stage? There is no crop in the field? Presumably also depends on crop rotation .

**Answer:** Yes, we are considering only a single crop type in field. This has been better explained in the revised version of the manuscript (see also response to comment RC2).

**RC50 (P15.L22):** In my view gross output or gross revenue would be a better term than turnover, throughout. (Turnover refers to total sales in a period, sales may include items from other production periods)

**Answer:** Ok. See response to RC9.

**RC51 (P15.L25):** Seems unlikely that there would be no yield penalty for delayed planting Furthermore, reseeding would probably not be feasible immediately post flood because of field conditions . Penalty delay functions could be used .

**Answer:** See response to comment RC49.

**RC52 (P15.L30):** Finally?

**Answer:** Ok, thanks.

**RC53 (P16. Figure5):** Would be good to make the axes consistent amongst the graphs , and for cost and turnover estimates. Would also be good to indicate net margin (or gross profit) , although this might complicate the graph. If a read it correctly, for a june flood, reseeding will not make sense , especially if there is (likely) yield penalty: I note for this graph the two 'y' scales are common

**Answer:** Figure 5 has been amended in the revised manuscript by taking into consideration reviewer's suggestions.

**RC54 (P17.L9):** This raises the question about likely average annual damaged according to the likelihood of a flood occurring within given months : where information is available on annul flood probability, and seasonal distribution, and to complicate further, whether seasonal distributions vary according the severity of the flood ? I see this is raised later

**Answer:** Yes. The importance of knowing "seasonal" return periods of floods is commented in the Discussion section.

**RC55 (P16. Figure5):** Is this really a table. The title does not explain that it is relative gross profit : this is difficult to interpret when the preceding assessment was made with respect to turnover ad costs, so some clear explanation is required. Is a relative loss of gross profit greater than 100% a helpful measure?

**Answer:** In the revised manuscript, the caption and the text referring to Figure 6 has been made more explicit. The numbers [%] reported in the figure express the relative damage, defined in Equation 2 of the paper, i.e. $d=D/NM_{noflood}$. Reference to Eq.2 has been included in the revised caption of Figure 6.

**RC56 (P17.L15):** The use of the term CBA needs explanation: it implies public choice and assessment of welfare change associated with public investments .

**Answer:** See response to comment RC3.

**RC57 (P17.L16):** Quite consolidated practices. Meaning

**Answer:** We mean that cost assessment in CBA is not very problematic, as all cost data can be easily determined. This sentence was removed in the new version of the manuscript.

**RC58 (P17.L18):** limited to the direct avoided damage to people and some exposed items . this is not clear

Answer: In the revised manuscript we have paraphrased this sentence as follows: "Present damage modelling capacity is mainly focused on direct damage to people and some exposed assets (typically residential buildings) thus preventing the possibility of performing comprehensive flood damage assessments and then reliable CBAs".

**RC59 (P17.L24):**The points here are not clear. I suggest the whole paragraph might be recrafted to advantage, with some examples to support the argument

**Answer:** The whole discussion sections have been redraft in order to better support our critical analysis of the model and its implementation (See Section 5)

**RC60 (P17.L30):** I am not convinced Figure 6 does this. What does the greater than 100% refer to: is this the gross profit estimate in Figure 6. Assumption of no yield loss with (delayed) reseeding probably underestimates losses . There may be opportunities for reseeding with a different crop, especially between winter sown and spring sown crops

**Answer:** 100% refers to the relative damage, as defined in Equation 2, i.e. $d=D/NM_{noflood}$. Reference to Eq.2 has been included in the revised Figure caption, where we have also clarified that "Results shown for the "r" option are obtained by assuming a null yield penalty for late (re-)plating". Regarding reseeding with a different crop, we have better explained in the revised version of the manuscript (see reply to

RC49) that we assume only a single crop type in field (reseeding with a different crop is considered not possible).

**RC61 (P17.L30):** Apart from EU contributions? Not clear

**Answer:** With "apart from EU contributions" we meant "if excluding the EU contributions" (which will still be obtained by the farmer). We have clarified this point in the revised version of the paper.

**RC62 (P17.L30):** Sustained? Already committed/incurred

**Answer:** Ok. This has been changed with "incurred" in the revised manuscript.

**RC63 (P19.L0-10):** These are valid and critical points, and fundamentally concern the underlying variation and uncertainty in the estimates (that have been single values so far). In my view it would be more appropriate to include the treatment of variation and uncertainty in the description of methods and the presentation of results of the case, rather than raise it for the first time here in discussion, where the purpose is to critical discuss the methods and results.

**Answer:** See reply to general comments regarding the reorganization of the paper. In our opinion, this specific point on uncertainty of estimations should remain in the Discussion section of the paper, as here, after the presentation of model structure, description of input parameters and data, we make comments of strengths and weaknesses of the adopted approach.

**RC64 (P19.Fig7):** This is results and should go there above. The figure is presumably for the Po case? The likely effect of a 10% penalty that would most likely arise due to (delayed) planting is apparent : negative gross profit. A figure showing absolute changes in gross profit (as defined here ) might be useful in the results section.

**Answer:** See replies to general comments regarding the reorganisation of the paper. Figure 7 has been removed to increase the readability of the paper

**RC65 (P19.L16):** Rather than saying 'must' it would be better to say why, identifying the advantage of doing so.

**Answer:** We explained the need of developing rapid approximate methods just in previous lines of the original manuscript (P19.L11-16): "The development of AGRIDE-c highlighted some challenges for the hydrology and the hydraulic community. In fact, application of the model requires a relatively detailed set of hazard input variables which are often not supplied in existing flood hazard maps (de Moel et al., 2009). Such knowledge would require a shift from traditional 1D steady hydraulic models to 2D unsteady hydraulic models - coupled with suitable sediment and contaminant transport models - in all flood prone areas, which is not easily achievable in a short time, both for technical and economic constraints".

**RC66 (P19.L18):** Perhaps rather than 'no more' , 'not only…. but also' seasonal probabilities Is the Morris and Hess ref 1988?

**Answer:** The reviewer is right. The sentence and year of Morris and Hess paper have been fixed in the revised manuscript.

**C67:** This paper? The reference to the spreadsheet and to supplementary data needs further support : these are mentioned in passing

**Answer:** "This paper": It is a writing style preference, as already discussed in replies to previous comments. In revising the paper, we have better emphasize reference to the spreadsheet and supplementary data.

**RC68 (P20.L5):** It depends how far the Authors have looked, and with the information presented here it is difficult to judge whether they can substantiate the claim. It might be fair to say they see advantage in developing a generic framework that can potentially be applied across different geographical and economic contexts , and they have made progress in this respect. For example, in more temperate part of Europe, land drainage is a particularly critical component of the land use: flooding nexus, and is particularly critical during post flood periods .

**Answer:** We have revised the sentence according to reviewer's suggestion. The issue of land drainage has been be discussed as well (see reply to general comments).

**RC69:** It would be useful to have a description of the sub models used , as referred to earlier. A summary table showing the estimation methods and sources would be particularly helpful, linked to supplementary data.

**Answer:** See answer to RC8.

**RC70:** Damage mechanisms- Meaning ? Drainage and soils might be important also. And also salinity issues in coastal areas, as referred earlier.

**Answer:** With damage mechanisms we mean the interaction between damage influencing factors and characteristics of exposed elements leading to a loss. This explanation has been included in the new version of the manuscript. Issues related to soil drainage and salinity have been included as well (see answers to general comment RC4-RC5).

**Manuscript nhess-2019-61 "AGRIDE-c, a conceptual model for the estimation of flood damage to crops: development and implementation" – Point by point response to referee 3 comments**

We would like to thank the referee for the work he did on our manuscript. We think that several of his comments can contribute to increase the manuscript robustness and, in general, to improve its quality and readability. Conversely, we partly or do not agree on some specific comments. In the following, we supply a point by point reply to referee comments.

**RC1:** My main concern is on the innovation provided by the article compared to precedent studies. From precedent studies, referred in the state of art section, it can be seen that many bricks presented in the article were already existing. For instance, a "conceptual model" has yet been formalized since the 80's in the USA, combining in a certain way "physical damage" and their "economic implications" in terms of loss of added value. Even if all detail are not given, many specifications are provided (see the user's manual of AGDAM, link provided below). Part of those details may be available from other studies. For instance, in our works (Agenais et al., 2013), we explicit completely how we link "physical damages" with "economic implications". Thus, I recommend that the article should be more specific on what it gets from previous studies, and what it added.

**Answer:** We agree with the referee when he states that "many bricks presented in the article were already existing". In fact, our effort was to organise "this fragmented knowledge" in a "generic" tool that could potentially improve the reliability and ease the procedure of flood damage assessments to crops in future studies. As stated at pg. 7 line 20-25 "AGRIDE-c has been developed by adopting an expert-based approach, encapsulating and systematising all the available knowledge on damage mechanisms triggered by inundation phenomena, as well as on their consequences in terms of income for the farmers. Information has been derived by a thorough investigation of the literature and by consultation with experts (i.e. agronomists and representatives of the authorities responsible for agricultural damage management and compensation). The result is a general, conceptual model, which identifies the different aspects to be modelled for the assessment of flood damage to crops, their (inter)connections as well as the variables at stake".

We would not define contents included in the AGDAM manual as "a conceptual model". AGDAM is a software and what is included in the manual is a diagram showing software inputs and outputs (pg. 4), and the theoretical background of its calculations. We do also not agree that AGDAM combines physical damage with economic implication. Loss functions in AGDAM are cost-based and assume the total loss of the revenue (pg. 16). The only physical aspect that is considered is the percentage of affected crops according to flood duration, on which base loss functions are weighted, but no reference to the reduction in the yield and/or its quality is made, according to different hazard and vulnerability variables. Given that the translation of physical damage into monetary terms can be challenging and not univocal, we strongly support the implementation of both a physical and an economic model. In fact, the distinction/link between physical and economic damage is present in several past works even if not explicitly (beyond Agenais et al. 2013, in Pivot et al. (2002), Morris and Hess (1988), Morris et al. (2014)); they are all quoted in the paper. We embraced this modelling approach in AGRIDE-c as we did in our previous work on flood damage to residential buildings, following a synthetic approach (Dottori et al., NHESS, 2016).

**RC2:** in their conclusion (page 20), the authors state that "According to authors' knowledge, AGRIDE-c represents the first attempt to organize all the available knowledge on flood damage to crops in a usable and consistent tool (i.e. the model integrates physical and economic approaches) that can be implemented to guide the flood damage assessment process, in different geographical and economic contexts." I do not master totally the American approach, but it has been developed to be used in different context (USA is a large country...) and has been developed as a tool used by USACE (ADGDAM). Another example is coming from Hess and Morris (1988), that also organized their work on grasslands in a framework comparable to that of AGRIDE-c and included it in a tool (SCADE). Last example, concerning our work, I can specify that in

chapter 3, the presentation of the methodological framework gives a clear explanation that damage are considered as loss of added value and how to link them to physical effects of flood.

**Answer:** we stress our opinion that AGDAM cannot be considered a conceptual model; moreover, its transferability to other (economic) contexts is limited by the fact that there are not specific guidelines on how to create or adapt loss functions. The work by Hess and Morris (1988) is instead very specific on grassland and is included on a tool for the evaluation of land drainage strategies. We agree that in chapter 3 of Agenais et al., 2013 damage is defined as loss of added value, as other authors did (e.g. Morris and Brewin, 2014; Pivot et al. 2002; Morris and Hess,1988), and that a brief discussion is included on how to link it to physical effects of flood; still, this cannot be considered a conceptual model in our point of view.

**RC3** In chapter 4, we give some precision on modelling of damage to sub-component of farms (which include crops, vegetal material, soil, equipment). For crops, we explain how to take into account farmers' strategies (continuation, abandoning, reseeding for instance), depending on when the flood occurs compared to the crop calendar.

**Answer:** Our model includes knowledge coming from Agenais et al. 2013, as it is stated in the paper. Although, we did not specify in the manuscript which works have been already taken into considerations farmers' strategies (like Agenais et al., 2013 but also Pivot, et. al 2002); we added these specific references in the paper (Section 3). Still, we want to stress that alleviation strategies are not explicitly taken into account in existing damage models, neither in the functions reported in chapter 5 of Agenais et al., 2013. In the implementation of AGRIDE-c in the Po Valley, we make explicit the effect of strategies on flood damage as well as when the different strategies can be implemented or not (Figure 6).

**RC4** In chapter 5 we explain that all this has been implemented in a tool called floodam (now floodam-agri), which aim is to help to adapt damage modelling to different context, including prices, crops calendars, and even also the question of the typology of culture to best fit with typology of GIS.

**Answer:** We cannot appreciate the potentialities of floodam as we could not find it online. According to our understating, chapter 5 of Agenais et al., 2013 is about the adaptation of national French functions to local French contexts.

**RC5** I want also to point that the authors present results for 4 types of crops, whereas both the American and the French approaches deal with many more types (including permanent crops)

**Answer:** After reading the referees' comments, we realised that the "scope of use" of our model was not well specified in the original version of the paper, which could lead to incorrect interpretations of our work. In fact, the conceptual model has been designed to supply an estimation of flood damage:
-       to annual crops (i.e. not including perennial plants)
-       by considering one single culture (i.e. by not considering replacement of one culture with another one)
-       by limiting the time frame of the analysis to one "productive cycle" (i.e. not considering long term damages, e.g. loss of soil productivity in the following cycles/years);
-       for infrequent flooding (i.e. effect of two, or more, consecutive floods is not considered)
Nonetheless, as specified at page 9 line 11-19, AGRIDE-c do not consider damage to other components/elements of the farm that may induce additional damage to crops, as, for instance, damage to machineries and equipment (e.g. the irrigation plant) that may prevent cultivation for a while. Only damage to soil is considered from the evidence that, during a flood, damage to soil and plants occurs always at the same time, differently from damage to the other components which can occur or not, independently from the damage to plants; as a consequence, damage to soil and plants is modelled together, while damage to the other components could be modelled as separated factors, not included in the conceptual model. We

thank the referees to highlight this limit of the original manuscript that we addressed by specifying all these aspects in the new version of the paper (Section 3).

Regarding the implementation in the Po Valley, its objective was not to create a comprehensive model for the estimation of flood damage to crops in the area, rather it was to exemplify how the conceptual model can be implemented in a specific context.

**RC6** I feel uncomfortable with the articulation of the two parts of the article, "conceptual model" and "implementation". From figure 2, it is expected that implementation of AGRIDE-c shall take into account all the phenomena described. But when coming to the implementation, it appears that many of those phenomena are not taken into account (loss of fertility due to sedimentation or loss of quality for instance). This shall be exposed in a clearer way not to induce false expectations on the scope of the study.

**Answer:** We thank the referee to highlight this lack of clarity in the paper. The idea of dividing the conceptual model from its implementation was also conceived with the objective of highlighting the gap between the available knowledge on damage mechanisms and their drivers/explicative variables, and their present modelling capacities. In fact, in order to be generally valid in any specific context, the conceptual model must include all the phenomena which affect the final loss figure. Then, its implementation must take into account not only the specific features of the investigated context (and then the relevant phenomena) but also modelling and data availability. In the Po Valley, data and models are required to properly take into account damage to soil and loss of quality; in their absence we made simplified assumptions, highlighting also research needs (see Conclusions). A table summarising the main elements and sources of revenue and cost estimation processes (i.e. model input data), considered in the application of the model in the Po Valley has been added in the new version of the paper (Table 3)

**RC7** Third, I think that the conceptual model is incomplete, at least for perennial crops (such as vineyard, but also grassland). First, it does not seem clear that the authors get that for some culture it may be useful to separate crops (fruits) and vegetal material (trees). This is not restricted to vineyard and orchards, but may also be important for asparagus, and even certain type of grasslands. If vegetal material is affected by flood, there may be at least two types of effect that last more than one year. For a given plot, if some plants are to be "destroyed" by a flood, it is expected that yield reduction and thus of products, but also variations of charges occur during the following years. This type of effects are for instance implicitly taken into account in Agenais et al.

**Answer:** see answer to RC5

**RC8:** Forth, I think there are not sufficient description of the role of the direct consultation of experts for the current work. It is only said (page 7) that some experts were consulted (agronomists and representatives of the authorities responsible for agricultural damage management and compensation), that this expertise were used to produce the production costs for normal activity (page 11, figure 3), the 3 possible strategies after a flood occur (pages 12-13) and an opinion on the suitability of yield reduction model coming from our works for maize grain (page 12). As many of the implementation seem to rely on the consultation of experts, I think a much more detailed description of this consultation shall be provided: how many experts has been consulted? What were their precise expertise, especially concerning flood impacts? What were their opinions on the data they provided? Have they been consulted on the results? What were their opinion on those results? What were their opinion on the transferability of those results on other context? This would strengthen this part of the work, that is almost invisible at the moment

**Answer:** We really thank the referee to highlight this lack in the original version of the manuscript. Experts were involved with two main objectives. The first one is to support the definition, and validate the quality, of the conceptual model. The second one is to give suggestions/information on the implementation of the model in the Po Valley, above all regarding expected physical damage and costs.

With respect to the first objective, an iterative process was followed. First, a semi-structured interview was conducted, by asking experts about the main damage mechanisms/phenomena in case of flood, possible interconnections among them, important explicative variables. In this phase, results from the literature review were proposed to experts for their judgment. In the following step, experts were asked to evaluate a draft version of the conceptual model we draw according to the literature review and results from first interviews. Then, there was an iterative revision process of improved versions of the model until an agreement on its final structure was reached.

With respect to the second objective, several individual meetings were organised with the aim of asking experts about context-specific information on: crops calendars, yields and prices, type, timing and costs of cultivation practices. In this phase, the transferability of the model by Agenais et al. was also discussed.

Three kinds of experts were involved. One representative of the Regional Authority responsible for agricultural damage management and compensation, with more than 20 years of expertise in the management and compensation of flood damage to farms in the Lombardy Region. Two agronomists of the local association of farmers (Coldiretti Lodi), with specific knowledge on the investigated context and with direct experience in managing floods in the last 20 years. During the work, the two agronomists asked for data/information also to individual local farmers that were flooded in the past years, including also their viewpoint in the process. Finally, an academic economist, with specific expertise in agriculture, has been involved in validating the final model.

This information explaining the whole process of experts' involvement has been included in Section 3 and 5 of the revised manuscript.

**RC9**: I would therefore recommend to be less ambitious in terms of interest of the article and to reorient it on the question of what has been necessary to adapt from previous works to a specific context. I invite the authors to be more precise on what they really include in their model, not on what they would have liked to include, because this makes things unclear for the reader. Another perspective would be to make a clear list of what that have not included. I also invite the authors to be more specific on how they have really implement their "local" model, by precising all the steps about consultations of experts. This seems important, especially for a expert-based approach. For those reasons I recommend a major revision.

**Answer:** we thank the referee for the comment as we think that, by addressing it, we could better specify (in the new version of the paper) which is the added value of the work (i.e. the conceptual model of available knowledge), which are its limits (i.e. the focus on the only crops component, the time frame of the analysis to one "productive cycle", the existing gaps between knowledge and modelling) and the implemented methodologies, above all with respect to expert 's consultation (see also answer to RC5, RC6 and RC7)

**RC10** Given the scope of the article, I am not totally convinced that the state of the art analysis should be done in such a detailed way. This is more convenient for a kind of review articles. If this section should stay in such a detail version, some imprecision needs to be corrected. For instance: - P3-L16. "No model in Table 2 considers instead the behavior of farmers after the occurrence of the flood (e.g. the decision of abandoning the production or to continue with increasing production costs)..." Agenais et al. does (see chapter 4) - P3-L22. The distinction between "physically based" and "cost based". is not clear. As formulated "cost based" models appears as simpler models where yield reduction is always total, which is not the case for "physical based" models. But for "physical based" models consequences in terms of production costs are considered. - Table 2. I am not convinced by the way some works are classified. For instance AGDAM cannot be said as "cost based", as it also consider some physical aspects on flood for yield reduction. I have not the time to check for all the works. Thus, I am not confident by what is presented in table 2. - P5-Table2 Agenais et al. present damage functions for 14 crops type, based on a detailed model of 50 crops types.

**Answer:** We think that the detail of the state of art is required in order to highlight research needs and then the objective of our work. We verified imprecisions suggested by the referee, individual comments are supplied:

- The model by Agenais et al. (i.e. functions in chapter 5) does not explicitly take into account alleviation strategies in damage estimation.
- According to us the distinction between physically-based and cost-based model is clear, laying in considering or not the estimation of physical damage. Accordingly, the AGDAM model must be correctly classified as "cost-based"
- Table 2 refer only to (annual crops); this will be specified in the new version of the paper

**RC11** I haven't seen the demonstration of what is promised in the abstract about "comprehensive cost-benefit analyses of risk mitigation actions". What is said in the discussion (page 17) is just that AGRIDE-c provide a way to estimate direct damage to crops, but in fact, it is only one contributions among others. I also feel that the authors did not get that for CBA purpose, it should be considered a "collective perspective", without considering possible transfers. This is not clear (see remarks on the "spreadsheet" tools). I think that should be reformulated.

**Answer:** The objective of the paper is to present a model and to discuss its potentialities, not demonstrating its usability. In fact, another manuscript is under preparation on the use of AGRIDE-c for the CBA of flood risk mitigation strategies in Lodi. We agree with the referee that including the potentialities of the work for CBA in the abstract can lead to misunderstanding of papers' results and findings. In the new version of the paper, we removed reference to CBA in the abstract, limiting its discussion in the introduction/discussion sections. The CBA of flood risk mitigation strategies would require a comprehensive estimation of benefits linked to the different strategies, i.e. of the avoided loss to all exposed sectors and at different temporal scales (i.e. direct and indirect/long term damages). Present damage modelling capacity prevents comprehensive flood damage assessments, which usually include only direct damage to people and some of the exposed assets (typically residential buildings). In such a context, by allowing the estimation of the expected loss to crops in a specific flood scenario, AGRIDE-c may support more comprehensive CBAs of public risk mitigation strategies. Of course, to meet such an objective, the tool must be critically used, e.g. by considering possible transfers of losses/gains between farmers in an economic perspective, according to the temporal and spatial scales of the analysis. With respect to other available tools, we think that AGRIDE-c, by conceptualising the whole damage estimation process, may lead to more reliable and transparent estimations. More comments have been added on this point in the new version of the paper (Section 1 and Section 5).

**RC12** I haven't seen neither the demonstration that AGRIDE-c is a "a powerful tool to orient farmers' behaviour towards more resilient damage alleviation practices". I do not know in detail what is the context of management of flood and agriculture in Italy. It is not presented in the article. But, this context may have some implications on what strategy would be follow by farmers, independently of what AGRIDE-c shall demonstrate. For instance, in France, if a farmer expect to receive some compensations from "Calamités Agricoles" (a State compensation scheme) or from "Assurances Récoltes" (Private insurance), he shall have to follow some recommendations concerning what he can follow as a strategy. If he does not, he may not receive any compensation. Also, it is not clear that the consequences presented are really those supported by the farmers. If there exists some compensations in Italia, this shall be included to provide a true "financial perspective" (point of view of the farmers).

**Answer:** We stress that the objective of the paper is to present a model and to discuss its potentialities, not demonstrating its usability. By supplying the expected damage for different types of crops and alleviation strategies (according to the expected yield reduction for different flood intensities and period of occurrence), AGRIDE-c may help individual farmers exposed to flood risk in preventing losses by supporting: the choice of the most appropriate crops to be cultivated, the choice of the best alleviation strategy to be followed once flooded, the evaluation of the opportunity to ask for a flood insurance scheme and the definition of the premium. This is a specific finding of the paper that have been better specified/commented in the new version of the manuscript (Section 5).
Insurance in Italy is not compulsory and is not linked to specific recommendations/strategies to be followed in case of flood.

**RC13** I am not convinced by the starting line 19 on page 19, concerning the necessity of "sediment and contaminant transport models" as the authors said before that they did not find available models to estimate the effects of sediment and contamination. This appears not coherent.

**Answer:** The effect of sediments and contaminants transported by flooding water on the yield is well documented in the literature (see Agenais et al., 2013; AGDAM, 1985, The Multi Coloured Manual, 2013, Hussain, 1996), as stressed in the paper. Accordingly, we included this effect in our conceptual model, although (of course) the importance of the phenomenon varies from place to place, being negligible in some areas like the one investigated in the manuscript. However, hazard assessments (i.e. flood hazard maps) usually do not supply estimates of sediments and contaminants load, even in such contexts when the phenomenon plays a crucial role, avoiding the estimation of its effect on flood damage to crops. This is a limit of present tools that we want to highlight in the paper.

**RC14** I think that some of the figures presenting the results shall be changed. In fact, in the title, the authors say that they analyze "damage to crops" but none of the figures clearly present a "net" damage. The reader has to make a mental effort to understand what are the damage from those figures (5 and 7): - make a difference between last point of the curve of scenario 0 and last points of three other curves for the production costs part - make a difference between a value given by a bar for scenario 0 and 3 other values for the "turnover" part - and then make a difference between the difference of turnover and the difference of production costs... Well, this should be done for the reader! This could allow to have a representation of the flood damage depending on the season of occurrence (as a function).

**Answer:** We thank the referee for the comment. We have amended Figure 5 (while Figure 7 was removed to increase the readability of the paper), explicating the value of the absolute damage, by showing also changes in gross profit (i.e. net margin in the revised paper, after reviewers' indication). However, we want to stress that Figure 5 does not represent the damage model but it is only functional to the description of the process leading to it. The final output of the damage model is displayed in Figure 6. From this, the calculation of the damage is immediate, by multiplying the relative damage by the net margin of the specific farm.

**RC15** Concerning figure 1, the authors announce that relative damage is supplied by our works (Agenais et al. 2013), but this is not the case. Our results are expressed in absolute damage (see page 51 of our report). Thus, the figure 1 is an interpretation of what we have done, but this interpretation is not explained. Moreover, this interpretation is necessary incorrect, as our results are presented on a seasonal time step (3 months) whereas the time scale in the figure 1 is one month. Moreover, as seen in our reports, relative damage are maximum only in summer, for long duration, and height over 130 cm, thus it is impossible to have relative damage of 100 % for any other case. I have not verified what is announced about the presentation of the results of Forster et al., but it shall be verified.

**Answer:** The referee is right as we did not use absolute damage functions at pg. 51 of Agenais et al., 2013 but relative "physical damage functions" supplied in the annex (pg. 200-202), and reported in Figure 4 of the paper. In fact, the objective was to highlight differences between physically-based and cost-based approaches. On the other hand, we choose the two models to allow a direct comparison in terms of relative damage, avoiding possible errors in transferring relative damage to absolute damage and vice versa. The physical damage functions in Agenais et al., 2013 are expressed for vegetative stages of the plant (not for three months) that we have linked to the months of the year, according to the Italian climate and cultural calendars (Initial phase: April-May, Growing phase: June, Flowering phase: July-August, Maturation phase: September-October, see Table 3). According to the functions, for a 3 days flood, a 100% physical damage occurs in the Initial phase (for any water depth value) and in the flowering phase for water depth greater than 130 cm. This is reflected in the first row of Figure 1 with a 100% peak of damage in April and July, which are representative, respectively, of the initial and flowering phases. For a 15 days flood, a 100% physical damage occur in the Initial, growing and flowering phase (for any water depth value) and also in the maturation phase, for water depth greater than 130 cm. This is reflected in the second row of Figure 1 with

a 100% damage from April to July (for water depth equal to 0.4 e 0.9 m) and from April to October (for water depth equal to 1.5 m), which are representative, respectively, of the initial, growing and flowering phases, and of the whole cultural cycle. To be more clear, we replaced Figure 1 with the following where 100% damage is reported for the whole duration of the different vegetative stages.

[Figure]

Although an explanation was required to clear referee doubts, we do not think that the detailed explanation of how we implemented the model must be included in the paper, as it is out of the scope of the meaning of Figure 1. Foster et al. was implemented at the best of our understanding.

**RC16** I had a look at the "spreadsheet" tool, and I share some comments on it, as I understood that all the application where made thanks to it: - There is not a manual to help people use the tool, it would be nice (necessary?). - Technically this tool is not designed to produce damage function but to estimate damage for specific value of hazard. This is not very practical for a user interested in a "damage function".

**Answer:** The tool was designed to support analysts in the calculation of damage for a specific hazard and vulnerability context, by implementing the damage functions we derived for the Po Valley, and which are reported in Figure 6 and in the Appendix. Such functions are included in the spreadsheet but can be also partly modified (by changing revenue and costs parameters). A user-manual have been added in the new-version of the paper to explain and increase its usability

**RC17** I have some questions on how the value coming from "Agenais" were filled, as I cannot remember that the authors asked for those values, which are not detailed in our report. - For instance, in some cases (maize, germination) yield reduction may occur for flood with a duration of 0 day, and in other cases (maize, flowering) there is no yield reduction for such flood with duration of 0 day. This may be a misunderstanding of what we developed. This leads to possible damage for a flood of 0 day and 0 cm for maize, which has no sense in our works. Such a flood is typically a flood with no consequences.

**Answer:** The referee is right when he says that the report by Agenais et al. does not detail model's values, which is a limit for its transferability. He is also right in saying that we did not ask for such values. In fact, given the simplicity of the functions, we made some assumptions that were supported by local experts' opinions. In particular, we used the relative physical damage functions reported in the Agenais report at pg. 200-202 (and also reported in our Figure 4) by assuming a linear increase of damage from 0 to 100% when required (see pg. 12 line 15)
According to such functions, in the initial phase, any flood will lead to a 100% damage (see pg. 200 of the Agenais et al. report), and it is in this sense that figure 4 must be read, with respect to the red square related

to the initial phase. On the contrary, in the flowering phase, a flood with a duration of 0 days will not lead to damage, and this is correctly reported in Figure 4 as in pg. 201 of the Agenais et al. report.

While tuning the model for the Po valley we started from the model of Agenais et al., 2013 we kept the same structure, we approximated trends with very simple functions (straight lines) while defining limits of such lines by a comparison of the original model and opinions of local experts. Due to the strong imprinting of Agenais et al. 2013 on our model, we considered as honest to declare the latter as an adaptation of the former. However, if the reviewer (and the editor) consider our model to be too loosely connected with the original one, we have no problem to indicate our model as freely inspired to that of Agenais et al., 2013 thus avoiding claiming any responsibility of the original one on our results.

**RC18** Another thing is that it is not specified that all data used from Agenais et al. are only specified for negligible flow velocity. This is particularly important for maize, wheat, and barley, that are very sensitive to this parameter. This may induce a bad use of the tool.

**Answer:** The referee is right. In fact, this is one of the reason for which we chose the model by Agenais et al., 2013, for the Po Valley, where riverine long-lasting floods are the typical flooding events. We have better specified this in the new version of the manuscript. See section 4.3: "Physical damage to crops is estimated by the physical model developed in France by Agenais et al. (2013). This choice is supported by different considerations. First, the independent hazard variables considered by the authors (for maize: water depth and flood duration) are coherent with the typical flooding characteristics identified for the Po Plain (Section 4.1), i.e. riverine long-lasting floods with low flow velocity.".

**RC19** One of the aspects that may change from site to site is the list of actions inside the crop management sequences. There are many reasons for which those actions may differ, not only in value but also in nature. This aspect is not taken into account, and doesn't seem to be easily taken into account with the provided tool.

**Answer:** The referee is right. By referring to the model we implemented for the Po Valley, the spreadsheet allows changing revenue and costs parameters, but not the type of cultivation practices. Still, as an open tool it can be easily modified to take into consideration of other context-specific cultivation practices.

**RC20** In the tool, "EU contributions for agriculture" are included. This is more oriented for a financial analysis (point of view of a specific farmer, including transfers) than for a Cost-Benefit Analysis (collective point of view, excluding transfers). I think a clear precision on the usage of damage produced should be added. If a financial perspective is to be promoted, all insurance or compensations mechanisms should be also included to give a better view of net consequences for the farmer.

**Answer:** The tool refers to the model implemented for the Po Valley where insurance is not compulsory. However, in order to use the tool in a financial perspective for a specific farm with an insurance, the tool can be easily adapted to include also this form of revenue/cost. Otherwise, if required, the tool can also be used for CBA of public mitigation strategies, by resetting cost parameters that may be included in transfers.

**AGRIDE-c, a conceptual model for the estimation of flood damage to crops: development and implementation**

Daniela Molinari[1], Anna Rita Scorzini[2], Alice Gallazzi[1], Francesco Ballio[1]

5  [1] Department of Civil and Environmental Engineering, Politecnico di Milano, Piazza Leonardo da Vinci 32, 20133, Milano, Italy

[2] Department of Civil, Environmental and Architectural Engineering, Università degli Studi dell'Aquila, Via Gronchi, 18, 67100, L'Aquila, Italy

*Correspondence to*: Daniela Molinari (daniela.molinari@polimi.it)

10  **Abstract.** This paper presents AGRIDE-c, a conceptual model for the assessment of flood damage to crops, in favour of more comprehensive flood damage assessments. Available knowledge on damage mechanisms triggered by inundation phenomena is systematised in a usable and consistent tool, with the main strength represented by the integration of physical damage assessment with the evaluation of its economic consequences on the income of the farmers. This allows AGRIDE-c to be used to guide the flood damage assessment process in different geographical and economic contexts, as demonstrated by the example provided in this study for the Po Plain (North of Italy). The development and implementation of the model highlighted that a thorough understanding and modelling of damage mechanisms to crops is a powerful tool to support more effective damage mitigation strategies, both at public and at private (i.e. farmers) level.

**1 Introduction**

On a global scale, floods are among the most common and damaging natural hazards (EEA, 2017, CRED, 2019). As climate
20  change continues to exacerbate extreme meteorological events, flood prone areas and flood-related damages are expected to grow rapidly in the future (Van Alst, 2006; Wobus et al., 2017; Alfieri et al., 2018; Mechler et al., 2019). To cope with this increasing risk, the EU Floods Directive (Directive 2007/60/EC) requires Member States (and, in particular, River Basin Districts) to periodically develop Flood Risk Management Plans, which are the operational/normative tools for the definition of flood risk mitigation strategies, including a blend of structural and non-structural measures. These measures must be
25  identified on the basis of a reliable and comprehensive assessment of costs and benefits related to the implementation of alternative strategies (Jonkman et al., 2004; Mechler, 2016), i.e. on cost-benefit analyses (CBAs), which implies a public choice based on the assessment of welfare change associated with public investments. In fact, CBAs would require a comprehensive estimation of the benefits produced by the adoption of different strategies (Jonkman et al., 2004; Mechler, 2016), consisting in the avoided losses to all exposed sectors and at different temporal scales (i.e. direct and indirect/long term
30  damages).

Present damage modelling capacity is mainly focused on direct damage to people and some exposed assets (typically residential buildings) thus preventing the possibility of performing comprehensive flood damage assessments and, consequently, CBAs (see e.g. Ballesteros-Cánovas et al., 2013; Saint-Geours et al., 2015; Meyer et al., 2013; Shreve and Kelman, 2014; Arrighi et al., 2018). On the opposite, the importance of developing new and reliable models for more inclusive flood damage assessments has been highlighted in recent investigations of past flood events (Pitt, 2008; Jongman et al., 2012; Menoni et al., 2016), showing that losses to the different sectors weigh differently according to the type of the event and the affected territory. To partially cover this gap, this paper deals with the estimation of flood damage to the agricultural sector, by presenting a new conceptual model for the estimation of flood damage to crops.

In the literature on flood damage modelling, agriculture has received so far less attention than other exposed sectors, as demonstrated in Table 1, showing the number of papers in the Scopus database for different research keywords. Reasons may include: (i) the (perceived) minor importance of agricultural losses compared to those of other sectors, especially because flood damage assessments are usually carried out in urban areas (Förster et al. 2008; Chatterton et al., 2016), (ii) the paucity of empirical data for understanding damage mechanisms and deriving prediction models, and finally, (iii) a policy shift, especially in Europe post 1980s, when the subsides to agriculture were being challenged by the increase of agricultural surpluses under the Common Agricultural Policy, along with the incentivisation of insurance coverage for damage to farms, that led most of public authorities responsible for damage compensation to be less interested in the agricultural sector. However, it must be stressed that flood risk management has been the concern of agricultural policies for many years, as since the 1930s, and probably up to the middle 1980s, agricultural policies were focused on land drainage (i.e. the removal of problems caused by the excess of water on/in the soil) of which flood protection was a critical part (Morris et al. 2008; Morris 1992). Still, literature related to land drainage is often difficult to retrieve and did not converge in the more recent studies on flood damage modelling, as much of the work is reported in the grey literature (see e.g. Hallett et al. 2016).

Available damage models for agriculture are not only few in number, but are also affected by many limitations, the major being the paucity of information/data for their validation and the large variability of the local features affecting damage (i.e. the strong linkage with the context under investigation), which limit their transferability to different contexts more than other exposed sectors as the residential and commercial ones; accordingly, the first requirement for a new damage model is its 
[revised manuscript text omitted]
. More specifically, experts were involved to support the definition of the conceptual model, by following an iterative process. In the first step of the process, a semi-structured interview was conducted, by asking experts about the main damage mechanisms/phenomena for crops in case of flood, important explicative variables and possible interconnections among them; moreover, results from the literature review were proposed for their judgment. In the following step, experts were asked to evaluate a draft version of the conceptual model drawn according to the literature review and results

from first interviews. Then, there was an iterative revision of improved versions of the model until an agreement on its final structure was reached. Three kinds of experts were involved in the process: (i) a representative of one of the Italian regional authorities responsible for agricultural damage management and compensation, with more than 20 years of expertise in the management and compensation of flood damage to farms in the Lombardy Region; (ii) two agronomists of a local association

5 of farmers (Coldiretti Lodi), with specific knowledge on the Po Plain context and with direct experience in managing floods in the last 20 years; the viewpoint of several individual local farmers who experienced flooding in the past years was also included in the analysis, as the two agronomists asked them for direct data and information to support their considerations; (iii) an academic economist, with specific expertise in agriculture.

It must be highlighted that the conceptual model has been designed to supply an estimation of flood damage only to annual

10 crops (i.e., not including perennial crops) under the following assumptions:

- infrequent flooding events (i.e., effect of two, or more, consecutive floods is not considered);

- flooded agricultural plot devoted to a single crop type, with possible reseeding using the same crop type in case of flood;

- time frame of the analysis limited to one productive cycle: long term damages, in particular, loss reduction of soil productivity in the following cycles is not considered;

15 In addition, AGRIDE-c does not consider damage to other components/elements of the farm that, on turn, may induce additional damage to crops, as, for instance, damage to machineries and equipment (e.g. irrigation system) that may prevent cultivation for a while (Dunderdale and Morris, 1997; Posthumus et al., 2009; Agenais et al., 2013; Bremond et al., 2013; Morris and Brewin, 2014). Only short term impacts on soil are included, based on the evidence that, during a flood, damages to soil and crops are concurrent, differently from damages to the other components which can occur or not, independently from

20 the damage to the vegetal material; as a consequence, damage to soil and crops is modelled together, while damage to the other components can be modelled as separated factors.

The model structure is depicted in detail in Figure 2. Absolute damage (D) for an individual farmer is expressed as the difference between the reduction in the gross output ($\Delta GO$) and the increase/decrease in production costs ($\Delta PC$), as a consequence of the flood of a specific crop. This is equal to consider absolute damage as the change in the net margin (NM =

25 GO– PC, where GO = gross output and PC = production costs) due to the flood, compared to the case when no flood occurs (i.e., Scenario 0):

$$D = NM_{noflood} - NM_{flood} = (GO_{noflood}-GO_{flood}) - (PC_{noflood}-PC_{flood}) = \Delta GO - \Delta PC \qquad (1)$$

Accordingly, relative damage (d) can be obtained by dividing the absolute damage by the net margin in the Scenario 0 ($NM_{noflood}$)

30 $$d = D/NM_{noflood} = 1 - NM_{flood}/NM_{noflood} \qquad (2)$$

[Figure]

**Figure 2. Conceptual model of AGRIDE-c**

AGRIDE-c combines a physical and an economic model to evaluate the absolute damage. In this way, the problems of consistency among physically-based and/or cost-based models discussed in Section 2 are overcome, being both aspects explicitly taken into account.

The physical model (identified by the yellow dashed box in Figure 2) is composed of two sub-models, for the evaluation of physical damage to crops (i.e. the plants) and impact on soil, respectively. In fact, as previously stated, among the different components/elements of the farm that may induce damage to crops, only damage to soil is considered in AGRIDE-c.

The model for the assessment of physical damage to soil calculates the amount of soil that is damaged, the kind(s) of damage suffered by the soil and the reduction of soil fertility, as a function of the duration of the flood, the water velocity, the sediment, the salinity (in case of coastal flooding) and the contaminants load. In particular, the model takes into account of processes like erosion, deposition of sediments and contamination (which affect the costs for soil restoration),as well as of the soil fertility (which affects the quality and the quantity of the harvest). In addition, the model estimates the effect of possible waterlogging, as a consequence of an increase in the level of the field water table, in terms of soil fertility reduction and (prolonged) soil saturation, which may increase costs for restoration because of the necessity of land drainage. It must be noted that, although in the European context floods usually have a negative effect on soils, some studies (e.g., Tockner et al., 1999; Hein et al., 2003) pointed out that such events can also have clearly positive effects, namely in the form of an increase of soil fertility, explained by a (re-)distribution of river sediments and organic matter in the course of flooding that replenish carbon and nutrients in topsoil.

[revised manuscript text omitted]

Table 3 summarises the main general data required by the conceptual model and the values / information used in the application for the Po Plain (example of maize).  Data sources are clarified in the following sub-sections.

The implementation of the conceptual model to Po Plain was supported by specific knowledge of local experts. In particular,

20  several individual meetings were organised with the aim of obtaining context-specific information related on crop calendars, yields and prices, type, timing and costs of the different cultivation practices.

**4.1 Hazard and vulnerability features in the Po Plain**

In order to identify the representative features of the floods and the main crops cultivated in the investigated area, we chose the Province of Lodi (Lombardia Region) as representative of hazard phenomena and agricultural activities in the Po Plain.

25  The last significant event occurred in the province, i.e. the flood of the Adda River in November 2002 (AdBPo, 2003; AdBPo, 2004; Rossetti et al., 2010; Scorzini et al., 2018), highlighted riverine long-lasting floods, characterised by medium to high water depths (mean value: 0.9 m), low flow velocities (mean value: 0.2 m/s) and low sediment and pollution loads in the flooded areas as typical of the region; accordingly, main hazard parameters to be included in the analytical expression of AGRIDE-c for the Po Plain are limited to water depth, flood duration and time (month) of flood occurrence.

The analysis of the agricultural cadastral data (supplied by the Regional Authority) in a buffer of 1 km around the Adda River, indicated grain maize, wheat, barley and grassland as the most common crops in the area; the model for maize is discussed hereinafter, while those related to other crops are reported in the supplement.

**Table 3. Summary of input data required by AGRIDE-c: exemplification for the Po Plain**

| Conceptual model | | Implementation for the Po Plain (example of maize) | |
|---|---|---|---|
| | *Input parameters* | *Modelling and input values* | *Data sources* |
| *Physical Model* | | | |
| Damage to crop | As shown in Fig.2 | Transferred and adapted from Agenais et al. (2013) | Agenais et al. (2013) and experts consultation |
| Impact on soil | As shown in Fig.2 | Soil restoration considered as a fixed cost (500€/ha) | APIMA (2013-2017) and experts consultation |
| *Economic Model* | | | |
| *Gross output* | Crop yield | 175 q/ha | Regione Lombardia (2013-2017) |
| | Unit price for crop | 16.9 €/q | Borsa Granaria di Milano (2013-2017) |
| | Other (e.g. EU contributions) | 150 €/ha for crop rotation; 300€/ha for minimum tillage | PSR Regione Lombardia |
| *Production costs* | | | |
| Variable costs | Depend on crop type and cultivations practises / strategies | As shown in Fig.3 and Tab. 4 | APIMA (2013-2017) and experts consultation |
| Fixed costs | | Assumed equal to 5% of the gross output | Experts consultation |

**4.2 Characterisation of the Scenario 0**

The Scenario 0 is characterised in terms of the annual net margin for the farmer, per hectare, in the case no flood occurs; this implies the estimation of the annual gross output and the distribution of production costs over the year.

10    Given that the vegetative cycle of grain maize in the Po Plain covers one year, the gross output is estimated as the product between the average yield and price for grain maize over the period 2013-2017 (data sources: Regione Lombardia and Borsa Granaria di Milano ( Milan Crops Stock Market)), equal to 175 q/ha and 16.92 €/q, respectively. In addition, we also consider the annual EU contributions for agriculture as a further income for the farmer and, in detail, the subsidies given to agricultural activities in case of the application of minimum tillage and crop rotation, equal respectively to 300 and 150 €/ha (data source:

15    PSR - Programma di Sviluppo Rurale, Regione Lombardia: http://www.psr.regione.lombardia.it).

Concerning production costs, the type, period of the year and costs of the different cultivation practices for grain maize were identified with the support of discussions with experts and consultation of regional price books (data source: APIMA – Associazione Provinciale Imprese di Meccanizzazione Agricola delle Province di Milano, Lodi, Como, Varese: Tariffe 2013-2017 delle lavorazioni meccanico agricole c/terzi, i.e., price lists for agricultural operations by contractors). All agricultural

20    operations have been considered as direct, avoidable costs, as interviewed local experts indicated that in Lodi province most

of field operations are carried out by contractors. Figure 3 reports the distribution of costs over the year, with indication of the corresponding vegetative stages of the plant.

[Figure]

 **Figure 3. Po Plain case: production costs over the year for grain maize, in case of minimum tillage technique**

Finally, fixed costs sustained by farmers (like management costs) are assumed to be a portion (5%) of the gross output. Based on these data, the analysis results in a net margin for the famer in case of no flood equal to 1376 €/ha per year.

**4.3 Damage to crops**

Physical damage to crops is estimated by the physical model developed in France by Agenais et al. (2013). This choice is supported by different considerations. First, the independent hazard variables considered by the authors (for maize: water depth and flood duration) are coherent with the typical flooding characteristics identified for the Po Plain (Section 4.1), i.e. riverine long-lasting floods with low flow velocity. Second, their model can be easily transferred to other regions, independently from crop calendars, as they use the vegetative phases of the crop (and not the months of the year) as the time variable for the occurrence of the flood. Finally, local agronomists expressed a favourable opinion on the suitability of this model in the examined region, as emerged from discussions held during the interview process.

An example of the physical damage model for maize is depicted in Figure 4 (adapted from Agenais et al., 2013). The model consists of susceptibility functions giving the yield reduction due to the flood (as a percentage of the yield in the Scenario 0), on the basis of water depth and flood duration, for four different vegetative stages (i.e. seeding, growing, flowering and maturation). Let us consider, for example; the growing stage: for a flood lasting less than 5 days the model gives a null yield loss, independently from the water depth; on the opposite, a flood lasting more than 12 days results in a total loss. For floods with intermediate duration, in absence of specific information in the original model and in accordance with the opinion of local

experts, we assumed a linear yield reduction (from 0 to 100%) between 5 and 12 days, adapting the model to the context under investigation. The use of this model implies that, at present, we do not take into account neither the reduction in the quality of the yield due to the flood nor the effect of damage to soil (i.e. reduction of soil fertility) on yield quality and production; reason for such limitations is simply the lack of literature and data on these topics (see also Section 4.4).

5    ## 4.4 Impact on soil

Concerning the physical impact on soil, only the negative effects of floods were computed as, according to local experts, increase in soil fertility due to floods is infrequent in Northern Italy. Likewise, waterlogging after floods is not relevant in the investigated area and has been neglected.

For the estimation of physical damage to soil, no models were found in the literature investigating the complex chemical and
10   mechanical processes leading to soil erosion, contamination and asphyxiation due to sediment deposition; also interviewed experts were not able to parametrise the possible types of damage, the amount of damaged soil and the reduction in soil fertility as a function of hazard features. For these reasons, at present, the model is based on the simplified assumption that soil always requires restoration in case of flood (consisting in the removal of sediments and in the levelling of terrain) and that no reduction in soil fertility occurs. Indeed, in the context under investigation, erosion and contamination are not expected because of the
15   low velocity and limited contaminant load characterising typical floods in the region (see Section 4.2).

The choice to include the damage to soil component in the implementation of AGRIDE-c, although in this simplified way, was driven by two main reasons: comprehensiveness of the model and importance of this sub-component in the overall flood damage figure to agriculture. In particular, this last point clearly emerged during the interviews with local experts, who pointed out the occurrence of such damages even for flood events characterised by shallow water depths and not particularly high flow
20   velocities. According to estimation of necessary operations supplied by interviewed experts and regional price books (data source: APIMA), restoration costs have been considered here, in a first instance, as fixed costs equal to 500 €/ha.

**4.5 Alleviation strategies**

[revised manuscript text omitted]

5  D = D (month, water depth, flood duration, alleviation strategy) = ΔGO - ΔPC                                    (3)

In detail, ΔGO and ΔPC are calculated on the basis of yield reduction and additional and avoided costs, as reported in Table 4. The resulting damage function has a fixed component due to soil restoration costs, to be added to the costs which varies with the flood characteristics and the alleviation strategy.

As an example of damage estimation, Figure 5 shows changes in production costs and gross output for maize cultivation, for
10  three different flood scenarios. Values of the annual gross output and of cumulative production costs are reported for both Scenario 0 and the flood scenario under investigation, with respect to every alleviation strategy farmers can implement according to the intensity of the flood, its time of occurrence and the physical damage suffered by the plant. Differences of production costs and turnover between "flood" and "no flood" scenarios allow calculating the damage D for the farmer.

The first scenario (Figure 5a) refers to a November flood. In this month, the plant is in the break stage, so no yield loss is
15  expected for any flood intensity (Table 4). Farmers will then continue the production with additional costs limited to those required to restore the flooded soil for a total of 500 €/ha (Table 4), which is the absolute damage sustained by farmers.

The second scenario (Figure 5b) refers to a flood in June, when the plant is in the growing stage. According to the physical model described in Figure 4, in this phase damage depends only on flood duration, while water depth has no effect on it. Figure 5b refers to a 5 days flood which leads, as given by the physical model, to a yield reduction of 12.5%. Given the low physical
20  damage, farmers can decide to continue the production or to reseed. In the first case (green line), the gross output decreases by 12.5% (due to yield reduction), while production costs increase due to additional costs for soil restoration, resulting in an absolute damage for the farmer equal to about 870 €/ha. In the second case (blue line), no reduction in the gross output occurs because reseeding would allow 100% of the yield, while additional production costs include both soil restoration and reseeding costs, resulting in an absolute damage of 1106 €/ha. Figure 5b shows that, although possible in theory, abandoning the
25  production is not a reasonable choice as absolute damage equals 2568 €/ha, due to a yield reduction of 100% (the only income for the farmer consists in the EU contributions for cultivation) against a saving of production costs of about 389 €/ha.

Finally, Figure 5c refers to a flood occurring in September; in this period (i.e. maturation phase of the plant), damage depends on both water depth and flood duration. Figure 5c refers in particular to a 10 days flood with a water depth above 1.30 m. According to the physical model (Figure 4), this flood scenario leads to a 50% yield loss. Farmers have then two choices.

[Figure]

a)  November flood (vegetative stage: break): any flood depth and duration

b)  June flood (vegetative stage: growing): any flood depth and 5 days duration (yield loss 12.5%)

[Figure]

c)  September flood (vegetative stage: maturation): flood depth > 1.30 m and 10 days duration (yield loss 50%)

**Figure 5. Po Plain case: distribution of cumulative production costs for grain maize during the year and annual gross output and net margin in the scenario 0 and in the case of a flood occurring in different months. Colours refers to the different possible strategies the farmer can adopt according to: the time of occurrence of the flood, intensity (water depth and duration) and physical damage. The absolute damage for the farmer (Di) is obtained by the difference of the net margin in the Scenario 0 and in the investigated scenario, as exemplified in Figure 5a.**

If production is continued the gross output decreases by 50% and additional costs are required to restore the flooded soil, resulting in an absolute damage equal to 1980 €/ha. In case of abandoning, absolute damage equals 2677 €/ha, because of a yield reduction of 100% and saving of production costs of 283 €/ha.

Previous considerations can be repeated for the different months of the year and hazard scenarios. Figure 6 displays the
5    ensemble of the results of damage estimation for all the investigated cases, thus defining the AGRIDE-c model for the Po Plain, for grain maize crops. In particular, the figure reports the relative damage with respect to the net margin in case of no inundation, $d=D/NM_{noflood}$, estimated by the model, for the different months of flood occurrence, flood intensities (i.e. water depth and flood duration) and damage alleviation strategies. The "dash" symbol means that the corresponding strategy cannot be adopted or is not reasonable in the flood scenario under investigation. For example, in the "bare field" season, reseeding is
10   not possible because of climatic reasons, nor it is continuation as no cultivation is in place; continuation does not make sense when a 100% yield loss is expected as in the "initial phase" or in the "flowering" stage when $h \geq 1.3$ m; reseeding with late crops is possible only until June, etc. Equivalent tables for the other investigated crops are reported in the Supplement.

**5 Discussion**

The AGRIDE-c model, by enabling the estimation of the expected direct damage to crops in case of flood, represents a
15   powerful tool to support more informed decisions on flood risk management for both public and private stakeholders. AGRIDE-c contributes to overcome the limitations of present CBAs, by providing a more comprehensive estimation of flood damages, thus supporting a better definition and choice of public actions for risk mitigation. In addition, the inclusion of damage to agriculture in CBAs is fundamental, especially when the interventions involve floodplains devoted to agricultural activities, as it is typically the case of river restoration actions, included in "integrated river basin management" projects
20   (Morris and Hess, 1988; Morris et al., 2008; Rouquette et al., 2011; Brémond et al., 2013; Massaruto and De Carli, 2014; Guida et al., 2016). Clearly, the tool must be critically used, e.g. by considering possible transfers of losses/gains between farmers in an economic perspective, according to the temporal and spatial scales of the analysis.

The development of AGRIDE-c and its implementation in the Po Plain highlighted that a thorough understanding and modelling of damage mechanisms to crops (i.e., of the interaction between damage influencing factors and characteristics of
25   exposed elements leading to a loss) is also useful to orient the behaviour of farmers towards more resilient practices, as the selection of the most resilient crops to be cultivated in areas prone to flooding, the choice of the best alleviation strategy to be followed once flooded, the evaluation of the opportunity to ask for a flood insurance scheme and the definition of the premium. For example, for the context and crop types investigated in the case study, Figure 6 highlights that abandoning the production is always the worst strategy, leading to a relative damage greater than 100% in any vegetative stage and for any flood intensity,
30   due to the combined effect of the total loss of the gross output (if excluding the EU contributions, obtained by the farmer also without any yield) and of the costs incurred by the farmer before the flood. On the other hand, when flood intensity implies significant yield loss, reseeding (if possible) must be preferred to continuation, limiting the relative damage to 80%;

nevertheless, the positive advantage of reseeding over continuation becomes smaller when including a yield penalty for late (re-)planting: results obtained by using the AGRIDE-c spreadsheet indicate a relative damage of 102% and 145% for a yield reduction of 10% and 30%, respectively.

The model presents some limitations that must be addressed in future research works and must be carefully taken into account in its implementation. The first is related to data requirements: the number and typology of input parameters may prevent its use in data-scarce areas. However, it must be stressed that high-detailed tools like AGRIDE-c should be adopted only at an advanced stage of the analysis, when the costs of collecting site-specific data may be justified by the expected results (i.e. the choice of the best mitigation strategy); in other cases, like preliminary damage analyses for the identification of priority intervention areas or post-event assessments, rapid tools (e.g. based on standardised damage/costs) should be preferred.

A second limitation concerns the high uncertainty characterising the input data required by AGRIDE-c, even in a specific context. An example is the estimation, based on few parameters (see Section 4.5), of the expected yield reduction due to late (re)seeding, which may be problematic as it is very variable and dependant on many factors (among others, type of late hybrids used). This implies that damage estimation may be affected by significant uncertainty, which is hardly quantifiable due to the limited availability of data for model validation (see Section 2); this uncertainty can be even amplified by the inherent uncertainty of the sub-models implemented in AGRIDE-c, like the economic or physical models for the estimation of flood damage to soil and crops.

This suggests, as for other damage models, the use of AGRIDE-c in a CBA context not in absolute terms (i.e. to evaluate the effectiveness of a specific measure), but as a tool to compare and choose among several alternatives (Scorzini and Leopardi, 2017; Molinari et al. 2019).

Likewise, a sensitivity analysis of input variables should always be performed, to get a flavour of robustness of findings. For example, for maize, the model developed for the Po Plain reveals (not shown here) that even a reduction of 10% of the yield in the Scenario 0 (with respect to the value adopted in the analysis) impacts the damage scenarios, leading to a relative damage greater than 100%, even in the case of reseeding in April and June and continuation in July and September (when yield loss is expected). The same occurs if the selling price decreases more than 12.5%, or EU contribution for the minimum tillage is not considered or production costs increase more than 10%. The "new" damage scenarios change the relative convenience associated to the different mitigation strategies; in particular, continuation may be more convenient that reseeding for short duration floods. Sensitivity analysis allows also investigating the effect on damage of possible changes in the physical and economic context in which the farm is located; in fact, all of the scenarios analysed in the previous example are globally representative of the context under investigation, but they can significantly vary among different farmers and different years: physical productivity is spatially non-uniform within the sub-regions of the Po Plain; prices and costs are highly variable in time and specific locations; only few farmers apply for EU contributions for the minimum tillage.

**Water depth < 130 cm**

| Phase | Month | Strategy | <5 | 5 | 6 | 7 | 8 | 9 | 10 | 11 | >11 |
|---|---|---|---|---|---|---|---|---|---|---|---|
| Bare field | Jan | c | | | | | 36% | | | | |
| | | r | | | | | - | | | | |
| | | a | | | | | - | | | | |
| | Feb | c | | | | | 36% | | | | |
| | | r | | | | | - | | | | |
| | | a | | | | | - | | | | |
| | Mar | c | | | | | 36% | | | | |
| | | r | | | | | - | | | | |
| | | a | | | | | - | | | | |
| Initial phase | Apr | c | | | | | - | | | | |
| | | r | | | | | 80% | | | | |
| | | a | | | | | 158% | | | | |
| | May | c | | | | | - | | | | |
| | | r | | | | | 80% | | | | |
| | | a | | | | | 158% | | | | |
| Growing | Jun | c | 36% | 63% | 90% | 117% | 144% | 171% | 198% | 225% | - |
| | | r | | | | | 80% | | | | |
| | | a | - | | | | 187% | | | | |
| Flowering | Jul | c | 36% | 90% | 144% | 198% | | | - | | |
| | | r | | | | | - | | | | |
| | | a | - | | | | 191% | | | | |
| | Aug | c | 36% | 90% | 144% | 198% | | | - | | |
| | | r | | | | | - | | | | |
| | | a | - | | | | 191% | | | | |
| Maturation | Sep | c | | | | | 36% | | | | |
| | | r | | | | | - | | | | |
| | | a | | | | | - | | | | |
| | Oct | c | | | | | 36% | | | | |
| | | r | | | | | - | | | | |
| | | a | | | | | - | | | | |
| Bare field | Nov | c | | | | | 36% | | | | |
| | | r | | | | | - | | | | |
| | | a | | | | | - | | | | |
| | Dec | c | | | | | 36% | | | | |
| | | r | | | | | - | | | | |
| | | a | | | | | - | | | | |

**Water depth ≥ 130 cm**

| Phase | Month | Strategy | <5 | 5 | 6 | 7 | 8 | 9 | 10 | 11 | >11 |
|---|---|---|---|---|---|---|---|---|---|---|---|
| Bare field | Jan | c | | | | | 36% | | | | |
| | | r | | | | | - | | | | |
| | | a | | | | | - | | | | |
| | Feb | c | | | | | 36% | | | | |
| | | r | | | | | - | | | | |
| | | a | | | | | - | | | | |
| | Mar | c | | | | | 36% | | | | |
| | | r | | | | | - | | | | |
| | | a | | | | | - | | | | |
| Initial phase | Apr | c | | | | | - | | | | |
| | | r | | | | | 80% | | | | |
| | | a | | | | | 158% | | | | |
| | May | c | | | | | - | | | | |
| | | r | | | | | 80% | | | | |
| | | a | | | | | 158% | | | | |
| Growing | Jun | c | 36% | 63% | 90% | 117% | 144% | 171% | 198% | 225% | - |
| | | r | | | | | 80% | | | | |
| | | a | - | | | | 187% | | | | |
| Flowering | Jul | c | | | | | - | | | | |
| | | r | | | | | - | | | | |
| | | a | | | | | 191% | | | | |
| | Aug | c | | | | | - | | | | |
| | | r | | | | | - | | | | |
| | | a | | | | | 191% | | | | |
| Maturation | Sep | c | | | | | 36% | | 90% | 144% | 198% | - |
| | | r | | | | | - | | | | |
| | | a | | | | | - | | 195% | | |
| | Oct | c | | | | | 36% | | 90% | 144% | 198% | - |
| | | r | | | | | - | | | | |
| | | a | | | | | - | | 195% | | |
| Bare field | Nov | c | | | | | 36% | | | | |
| | | r | | | | | - | | | | |
| | | a | | | | | - | | | | |
| | Dec | c | | | | | 36% | | | | |
| | | r | | | | | - | | | | |
| | | a | | | | | - | | | | |

**Figure 6. Po Plain case: relative damage (Eq. 2) to maize crops (in case of minimum tillage) for the different combinations times of occurrence of the flood (i.e. month), flood intensities (i.e. water depth and flood duration) and damage alleviation strategies ("c"=continuation; "r"=reseeding; "a"=abandoning. Results shown for the "r" option are obtained by assuming a null yield penalty for late (re-)plating.**

A third limitation concerns the time frame of the analysis, focused on one productive cycle; this prevents the comprehensiveness of the damage assessment by neglecting long-term indirect damages, like those related to the low productivity of soil in the following years after the flood event. This limitation must be carefully considered when the tool is implemented for the choice of risk mitigation strategies, as the expected damage can be significantly underestimated.

5 Finally, comprehensiveness of damage assessment is limited by the lack of consideration of other farm components which may be damaged in case of flood like damage to perennial plants, livestock, stock, equipment and machineries, buildings, permanent equipment and farm roads (Brémond et al., 2013; Posthumus et al., 2009; Morris and Brewin, 2014) as well as of their systemic interaction (i.e., damage induced to one component by another one). Further research is required on the topic as well as post-event data to calibrate and validate models.

[revised manuscript text omitted]

**Competing interests**

The authors declare that they have no conflict of interest

*Correspondence to*: Daniela Molinari (daniela.molinari@polimi.it)

**Abstract.** This paper presents AGRIDE-c, a conceptual model for the assessment of flood damage to crops, in favour of more comprehensive flood damage assessments. All aAvailable knowledge on damage mechanisms triggered by inundation phenomena is systematised in a usable and consistent tool, with the main strength represented by the integration of physical damage assessment with the evaluation of its economic consequences on the income of the farmersfarmers' gross product. This allows AGRIDE-c to be used to guide the flood damage assessment process in different geographical and economic contexts, as demonstrated by the example provided in this study for the Po Plain (North of Italy). The development and implementation of the model highlighted that a thorough understanding and modelling of damage mechanisms to crops allows for comprehensive cost-benefit analyses of risk mitigation actions, and is a powerful tool to orient farmers' behaviour towardssupport more resilient effective damage alleviation practicesmitigation strategies, both at public and at private (i.e. farmers) level.

**1 Introduction**

On a global scale, floods are among the most common and damaging natural hazards (EEA, 2017, CRED, 2019). As climate change continues to exacerbate extreme meteorological events, flood prone areas and flood-related damages are expected to grow rapidly in the future (Van Alst, 2006; Wobus et al., 2017; Alfieri et al., 2018; Mechler et al., 2019). To cope with this increasing risk, the EU Floods Directive (Directive 2007/60/EC) requires Member States (and, in particular, River Basin Districts) to periodically develop Flood Risk Management Plans, which are the operational/normative tools for the definition of flood risk mitigation strategies, including a blend of structural and non-structural measures. These measures must be identified on the basis of a reliable and comprehensive assessment of costs and benefits related to the implementation of alternative strategies (Jonkman et al., 2004; Mechler, 2016), i.e. on cost-benefit analyses (CBAs), which implies a public choice andbased on the assessment of welfare change associated with public investments. In fact, CBAs would require a comprehensive estimation of the benefits produced by the adoption of different strategies (Jonkman et al., 2004; Mechler,

2016), consisting in the avoided losses to all exposed sectors and at different temporal scales (i.e. direct and indirect/long term damages).

5

present damage modelling capacity is mainly focused on direct damage to people and some exposed assets (typically residential buildings) thus preventing the possibility of performing comprehensive flood damage assessments and, consequently, CBAs (see e.g. Ballesteros-Cánovas et al., 2013; Saint-Geours et al., 2015; Meyer et al., 2013; Shreve and Kelman, 2014; Arrighi et al., 2018). On the opposite, the importance of developing new and reliable models for more

10  inclusive flood damage assessments has been highlighted in recent investigations of past flood events (Pitt, 2008; Jongman et al., 2012; Menoni et al., 2016), showing that losses to the different sectors weigh differently according to the type of the event and the affected territory. To partially cover this gap this paper deals with the estimation of flood damage to the agricultural sector, by presenting a new conceptual model for the estimation of flood damage to crops.

15  ~~devoted to agricultural activities: this is typically the case of river restoration actions, as usually included in "integrated river basin management" projects (Morris and Hess, 1988; Morris et al., 2008; Rouquette et al., 2011; Brémond et al., 2013; Massaruto and De Carli, 2014; Guida et al., 2016). The latter should consider all the direct and indirect costs and benefits that a specific measure brings to the society (Jonkman et al., 2004; Mechler, 2016), with benefits consisting in the avoided damage with respect to a null action.~~

20  ~~In this framework, this paper deals with the estimation of flood damage to the agricultural sector. Indeed, the inclusion of damage to agriculture in CBAs is critically needed, especially when risk mitigation measures involve floodplains devoted to agricultural activities: this is typically the case of river restoration actions, as usually included in "integrated river basin management" projects (Morris and Hess, 1988; Brémond et al., 2013; Massaruto and De Carli, 2014; Guida et al., 2016). Moreover, a thorough understanding of flood damage mechanisms may increase farmers' resilience to floods, by supporting~~

25

In the literature on flood damage modelling, agriculture has received so far less attention than other exposed sectors, as demonstrated in Table 1, showing the number of papers in the Scopus database for different research keywords. Reasons may include: (i) the (perceived) minor importance of agricultural losses compared to those of other sectors, especially because flood damage assessments are usually carried out in urban areas (Förster et al. 2008; Chatterton et al., 2016), (ii) the paucity of

30  empirical data for understanding damage mechanisms and deriving prediction models, and finally, (iii)  a policy shift, especially in Europe post 1980s, when the subsides to agriculture were being challenged by the increase of agricultural surpluses under the Common Agricultural Policy, along with the incentivisation of insurance coverage for damage to farms, that led most of public authorities responsible for damage compensation to be less interested in the agricultural sector.

However, it must be stressed that flood risk management has been the concern of agricultural policies for many years, as since the 1930s, and probably up to the middle 1980s, agricultural policies were focused on land drainage (i.e. the removal of problems caused by the excess of water on/in the soil) of which flood protection was a critical part (Morris et al. 2008; Morris 1992). Still, literature related to land drainage is often difficult to retrieve, and did not converge in the more recent studies on

5    flood damage modelling, as much of the work is reported in the grey literature (see e.g. Hallett et al. 2016). available damage models for agriculture are not only few in number, but are also affected by many limitations, the major being the paucity of information/data for their validation and the large variability of the local features affecting damage (i.e. the strong linkage with the context under investigation), which limit their transferability to different contexts more than other exposed sectors, as the residential and commercial ones

[revised manuscript text omitted]

25   More specifically, experts were involved to support the definition of the conceptual model, by  following an iterative process In the first step of the process, a semi-structured interview was conducted, by asking experts about the main damage mechanisms/phenomena for crops in case of flood, important explicative variables and possible interconnections among them; moreover, results from the literature review were proposed for their judgment. In the following step, experts were asked to evaluate a draft version of the conceptual model drawn according to the literature review and results from first interviews.

30   Then, there was an iterative revision of improved versions of the model until an agreement on its final structure was reached. Three kinds of experts were involved in the process (i) a representative of  regional authorities responsible for agricultural damage management and compensation, with more than 20 years of expertise in the management and

compensation of flood damage to farms in the Lombardy Region; (ii) Two agronomists of a local association of farmers (Coldiretti Lodi), with specific knowledge on the Po Plain context and with direct experience in managing floods in the last 20 years; the viewpoint of several individual local farmers who experienced flooding in the past years was also included in the analysis, as the two agronomists asked them for direct data and information to support their considerations; (iii) An academic economist, with specific expertise in agriculture.

~~The result of this process is a general, conceptual model, which identifies the different aspects to be modelled for the assessment of flood damage to crops, their (inter)connections as well as the variables at stake. Still, as stressed before, the implementation of the model (that is the derivation of an analytical expression for each of its components) must be context specific, as damage to crops depends on many local features that cannot be generalised. An example of the implementation of the model is supplied in Section 4.~~

It must be highlighted that the conceptual model has been designed to supply an estimation of flood damage only to annual crops (i.e., not including perennial crops) under the following assumptions:

- infrequent flooding events (i.e., effect of two, or more, consecutive floods is not considered);

- flooded agricultural plot devoted to a single crop type, with possible reseeding using the same crop type in case of flood;

- time frame of the analysis limited to one productive cycle: long term damages, in particular,  reduction of soil productivity in the following cycles is not considered;

In addition, AGRIDE-c does not consider damage to other components/elements of the farm that, on turn, may induce additional damage to crops, as, for instance, damage to machineries and equipment (e.g. irrigation system) that may prevent cultivation for a while (Dunderdale and Morris, 1997; Posthumus et al., 2009; Agenais et al., 2013; Bremond et al., 2013; Morris and Brewin, 2014). Only short term impacts on soil are included, based on the evidence that, during a flood, damages to soil and crops are concurrent, differently from damages to the other components which can occur or not, independently from the damage to the vegetal material; as a consequence, damage to soil and crops is modelled together, while damage to the other components can be modelled as separated factors.

The mModel structure is depicted in detail in Figure 2. Absolute damage (D) for an individual farmer is expressed as the difference between the reduction in the  gross output ($\Delta$GO) and the increase/decrease in production costs ($\Delta$PC), as a consequence of the flood of a specific crop. This is equal to consider absolute damage as the change in the net margin (NM = GO – PC, where GO = gross output and PC = production costs) due to the flood, compared to the case when no flood occurs (i.e., Scenario 0):

$$D = NM_{noflood} - NM_{flood} = (GO_{noflood}-GO_{flood}) - (PC_{noflood}-PC_{flood}) = \Delta GO - \Delta PC \qquad (1)$$

Accordingly, relative damage (d) can be obtained by dividing the absolute damage by the net margin in the Scenario 0 ($NM_{noflood}$)

$$d = D/NM_{noflood} = 1 - NM_{flood}/NM_{noflood} \qquad (2)$$

[Figure]

[Figure]

**Figure 2. Conceptual model of AGRIDE-c**

5   AGRIDE-c  combines a physical and an economic model to evaluate the absolute damage. In this way, the problems of consistency among physically-based and/or cost-based models discussed in Section 2 are overcome, being both aspects explicitly taken into account.

crop; on the
10  among these, AGRIDE-c considers only the damage to soil.
15   The physical model  (identified by the yellow dashed box in Figure 2) is  composed of two sub-models, for the evaluation of physical damage to crops (i.e. the plants) and impact  on soil, respectively. In fact, as previously stated, among the different components/elements of the farm that may induce damage to crops, only damage to soil is considered in AGRIDE-c.

The model for the assessment of physical damage to soil calculates the amount of soil that is damaged, the kind(s) of damage
20  suffered by the soil and the reduction of soil fertility, as a function of the duration of the flood, the water velocity, the sediment, the salinity (in case of coastal flooding) and the contaminants load. In particular, the model takes into account of processes like erosion, deposition of sediments and contamination (which affect the costs for soil restoration),as well as of the soil fertility (which affects the quality and the quantity of the harvest). In addition, the model estimates the effect of possible waterlogging, as a consequence of an increase in the level of the field water table, in terms of soil fertility reduction and (prolonged) soil
25  saturation, which may increase costs for restoration because of the necessity of land drainage. It must be noted that, although in the European context floods usually have a negative effect on soils, some studies (e.g., Tockner et al., 1999; Hein et al., 2003) pointed out that such events can also have clearly positive effects, namely in the form of an increase of soil fertility, explained by a (re-)distribution of river sediments and organic matter in the course of flooding that replenish carbon and nutrients in topsoil.

[revised manuscript text omitted]

Table 3 summarises the main general data required by the conceptual model and the values / information used in the application
5  for the Po Plain (example of maize). Data sources are clarified in the following sub-sections.

The implementation of the conceptual model to Po Plain was  supported by specific knowledge of local experts. In particular, several individual meetings were organised with the aim of obtaining context-specific information related on crop calendars, yields and prices, type, timing and costs of the different cultivation practices.

**4.1 Hazard and vulnerability features in the Po Plain**

10  In order to identify the representative features of the floods and the main crops cultivated in the investigated area, we chose the Province of Lodi (Lombardia Region) as representative of hazard phenomena and agricultural activities in the Po Plain.

The last significant event occurred in the province, i.e. the flood of the Adda River in November 2002 (AdBPo, 2003; AdBPo, 2004; Rossetti et al., 2010; Scorzini et al., 2018), highlighted riverine long-lasting floods, characterised by medium to high water depths (mean value: 0.9 m), low flow velocities (mean value: 0.2 m/s) and low sediment and pollution loads in the
15  flooded areas as typical of the region; accordingly, main hazard parameters to be included in the analytical expression of AGRIDE-c for the Po Plain are limited to water depth, flood duration and time (month) of flood occurrence.

The analysis of the agricultural cadastral data (supplied by the Regional Authority) in a buffer of 1 km around the Adda River, indicated grain maize, wheat, barley and grassland as the most common crops in the area; the model for maize is discussed hereinafter, while those related to other crops are reported in the supplement.

**Table 3. Summary of input data required by AGRIDE-c: exemplification for the Po Plain**

| Conceptual model | | Implementation for the Po Plain (example of maize) | |
|---|---|---|---|
| | *Input parameters* | *Modelling and input values* | *Data sources* |
| *Physical Model* | | | |
| Damage to crop | As shown in Fig.2 | Transferred and adapted from Agenais et al. (2013) | Agenais et al. (2013) and experts consultation |
| Impact on soil | As shown in Fig.2 | Soil restoration considered as a fixed cost (500€/ha) | APIMA (2013-2017) and experts consultation |
| *Economic Model* | | | |
| *Gross output* | Crop yield | 175 q/ha | Regione Lombardia (2013-2017) |
| | Unit price for crop | 16.9 €/q | Borsa Granaria di Milano (2013-2017) |
| | Other (e.g. EU contributions) | 150 €/ha for crop rotation; 300€/ha for minimum tillage | PSR Regione Lombardia |
| *Production costs* | | | |
| Variable costs | | As shown in Fig.3 and Tab. 4 | APIMA (2013-2017) and experts consultation |

| | | | |
|---|---|---|---|
| Fixed costs | Depend on crop type and cultivations practises / strategies | Assumed equal to 5% of the gross output | Experts consultation |

**4.2 Characterisation of the Scenario 0**

The Scenario 0 is characterised in terms of the annual net margin for the farmer, per hectare, in the case no flood occurs; this implies the estimation of the annual gross output and the distribution of production costs over the year.

5    Given that the vegetative cycle of grain maize  in the Po Plain covers  one year, the gross output is estimated as the product between the average yield and price for grain maize over the period 2013-2017 (data sources: Regione Lombardia and Borsa Granaria di Milano ( Milan Crops Stock Market)), equal to 175 q/ha and 16.92 €/q, respectively. In addition, we also consider the annual EU contributions for agriculture as a further income for the farmer and, in detail, the subsidies given to agricultural activities in case of the application of minimum tillage and crop rotation, equal

10   respectively to 300 and 150 €/ha (data source: PSR - Programma di Sviluppo Rurale, Regione Lombardia: http://www.psr.regione.lombardia.it).

Concerning production costs, the type, period of the year and costs of the different cultivation practices for grain maize were identified with the support of discussions with experts and consultation of regional price books (data source: APIMA – Associazione Provinciale Imprese di Meccanizzazione Agricola delle Province di Milano, Lodi, Como, Varese: Tariffe 2013-

15   2017 delle lavorazioni meccanico agricole c/terzi, i.e., price lists for agricultural operations by contractors). All agricultural operations have been considered as direct, avoidable costs, as interviewed local experts indicated that in Lodi province most of field operations are carried out by contractors. Figure 3 reports the distribution of costs over the year, with indication of the corresponding vegetative stages of the plant.

[Figure]

**Figure 3. Po Plain case: Pproduction costs over the year for grain maize, in case of minimum tillage technique**

Finally, fixed costs sustained by farmers (like management costs) are assumed to be a portion (5%) of the gross output.

5 Based on these data, the analysis results in a net margin for the famer in case of no flood equal to 1376 €ha per year.

**4.3 Damage to crops**

Physical damage to crops is estimated by the physical model developed in France by Agenais et al. (2013). This choice is supported by different considerations. First, the independent hazard variables considered by the authors (for maize: water depth

10 and flood duration) are coherent with the typical flooding characteristics identified for the Po Plain (Section 4.1), i.e. riverine long-lasting floods with low flow velocity. Second, their model can be easily transferred to other regions, independently from crop calendars, as they use the vegetative phases of the crop (and not the months of the year) as the time variable for the occurrence of the flood. Finally, local agronomists expressed a favourable opinion on the suitability of this model in the examined region, as emerged from discussions held during the interview process.

15 An example of the physical damage model for maize is depicted in Figure 4 (adapted from Agenais et al., 2013). The model consists of susceptibility functions giving the yield reduction due to the flood (as a percentage of the yield in the Scenario 0), on the basis of water depth and flood duration, for four different vegetative stages (i.e. seeding, growing, flowering and maturation). Let us consider, for example; the growing stage: for a flood lasting less than 5 days the model gives a null yield loss, independently from the water depth; on the opposite, a flood lasting more than 12 days results in a total loss. For floods

20 with intermediate duration, in absence of specific information in the original model and in accordance with the opinion of local experts, we assumed a linear yield reduction (from 0 to 100%) between 5 and 12 days, adapting the model to the context under investigation. The use of this model implies that, at present, we do not take into account  neither the reduction in the quality

of the yield due to the flood nor the effect of damage to soil (i.e. reduction of soil fertility) on yield quality and production; reason for such limitations is simply the lack of literature and data on these topics (see also Section 4.4).

**4.4  Impact  on soil**

Concerning the physical impact  soil, only the negative effects of floods were computed as, according to
5   local experts, increase in soil fertility due to floods is infrequent in Northern Italy. Likewise, waterlogging after floods is not relevant in the investigated area and has been neglected.
For the estimation of physical damage to soil, no models were found in the literature investigating the complex chemical and mechanical processes leading to soil erosion, contamination and asphyxiation due to sediment deposition; also interviewed experts were not able to parametrise the possible types of damage, the amount of damaged soil and the
10  reduction in soil fertility as a function of hazard features. For these reasons, At present, the model is based on the simplified assumption that soil always requires restoration in case of flood (consisting in the removal of sediments and in the levelling of terrain) and that no reduction in soil fertility occurs. Indeed, in the context under investigation, erosion and contamination are not expected because of the low velocity and limited contaminant load characterising typical floods in the region (see Section 4.2).
15  The choice to include the damage to soil component in the implementation of AGRIDE-c, although in this simplified way, was driven by two main reasons: comprehensiveness of the model –and importance of this sub-component in the overall flood damage figure to agriculture. In particular, this last point clearly emerged during the interviews with local experts, who pointed out the occurrence of such damages even for flood events characterised by shallow water depths and not particularly high flow velocities. According to estimation of necessary operations supplied by interviewed experts and regional price books (data
20  source: APIMA), restoration costs have been considered here, in a first instance, as fixed costs equal to 500 €/ha.

**4.5 Alleviation strategies**

[revised manuscript text omitted]

5    D = D (month, water depth, flood duration, alleviation strategy) = $\Delta$T GO - $\Delta$PC _______________________(3)

In detail, $\Delta$T GO and $\Delta$PC are calculated on the basis of yield reduction and additional and avoided costs, as reported in Table 4. The resulting damage function has a fixed component due to soil restoration costs, to be added to the costs which varies with the flood characteristics and the alleviation strategy.

As an example of damage estimation, Figure 5 shows changes in production costs and  gross output for maize
10   cultivation, for three different flood scenarios. Values of the annual gross output  and of cumulative production costs are reported for both Scenario 0 and the flood scenario under investigation, with respect to every alleviation strategy farmers can implement according to the intensity of the flood, its time of occurrence and the physical damage suffered by the plant. Differences of production costs and turnover between "flood" and "no flood" scenarios allow calculating the damage D for the farmer.

15   The first scenario (Figure 5a) refers to a November flood. In this month, the plant is in the break stage, so no yield loss is expected for any flood intensity (Table 3). Farmers will then continue the production with additional costs limited to those required to restore the flooded soil for a total of 500 €/ha (Table 3), which is the absolute damage sustained by farmers.

The second scenario (Figure 5b) refers to a flood in June, when the plant is in the growing stage. According to the physical model described in Figure 4, in this phase damage depends only on flood duration, while water depth has no effect on it. Figure
20   5b refers to a 5 days flood which leads, as given by the physical model, to a yield reduction of 12.5%. Given the low physical damage, farmers can decide to continue the production or to reseed. In the first case (green line), the gross output  decreases by 12.5% (due to yield reduction), while production costs increase due to additional costs for soil restoration, resulting in an absolute damage for the farmer equal to about 870 €/ha. In the second case (blue line), no reduction in the gross output  occurs because reseeding would allow 100% of the yield, while additional production costs include both soil
25   restoration and reseeding costs, resulting in an absolute damage of 1106 €/ha. Figure 5b shows that, although possible in theory, abandoning the production is not a reasonable choice as absolute damage equals 2568 €/ha, due to a yield reduction of 100% (the only income for the farmer consists in the EU contributions for cultivation) against a saving of production costs of about 389 €/ha.

Finally, Figure 5c refers to a flood occurring in September; in this period (i.e. maturation phase of the plant), damage
30   depends on both water depth and flood duration. Figure 5c refers in particular to a 10 days flood with a water depth above 1.30 m. According to the physical model (Figure 4), this flood scenario leads to a 50% yield loss. Farmers have then two choices.

[Figure]

a) November flood (break): any flood depth and duration

b) June flood (growing): any flood depth and 5 days duration (yield loss 12.5%)

[Figure]

c) September flood (maturation): flood depth > 1.30 m and 10 days duration (yield loss 50%)

[Figure]

a) November flood (vegetative stage: break): any flood depth and duration

b) June flood (vegetative stage: growing): any flood depth and 5 days duration (yield loss 12.5%)

[Figure]

c) September flood (vegetative stage: maturation): flood depth > 1.30 m and 10 days duration (yield loss 50%)

**Figure 5. Po Plain case: dDistribution of cumulative production costs for grain maize during the year and annual turnover gross output and net margin in the scenario 0 and in the case of a flood occurring in different months. Colours refers to the different possible strategies the farmer can adopt according to: the time of occurrence of the flood, intensity (water depth and duration) and physical damage. The absolute damage for the farmer (Di) is obtained by the difference of the net margin in the Scenario 0 and in the investigated scenario, as exemplified in Figure 5a.**

If production is continued the  gross output decreases by 50% and additional costs are required to restore the flooded soil, resulting in an absolute damage equal to 1980 €/ha. In case of abandoning, absolute damage equals 2677 €/ha, because of a yield reduction of 100% and saving of production costs of 283 €/ha.

Previous considerations can be repeated for the different months of the year and hazard scenarios. Figure 6 displays the ensemble of the results of damage estimation for all the investigated cases, thus defining the AGRIDE-c model for the Po Plain, for grain maize crops. In particular, the figure reports the relative damage with respect to the net margin in case of no inundation, $d=D/\text{GP}_{noflood}\text{NM}_{noflood}$, estimated by the model, for the different months of flood occurrence, flood intensities (i.e. water depth and flood duration) and damage alleviation strategies. The "dash" symbol means that the corresponding strategy cannot be adopted or is not reasonable in the flood scenario under investigation. For example, in the "bare field" season, reseeding is not possible because of climatic reasons, nor it is continuation as no cultivation is in place; continuation does not make sense when a 100% yield loss is expected as in the "initial phase" or in the "flowering" stage when $h \geq 1.3$ m; reseeding with late crops is possible only until June, etc. Equivalent tables for the other investigated crops are reported in the Supplement.

**5 Discussion**

The AGRIDE-c model, by enabling the estimation of the expected direct damage to crops in case of flood, represents a powerful tool to support more informed decisions on flood risk management for both public and private stakeholders. By enabling the estimation of the  AGRIDE-c contributes to overcome the limitations of present CBAs, by providing  a more comprehensive estimation of flood damage, thus supporting a better definition and choice of public actions for risk mitigation only direct avoided damage to people and . In addition, the inclusion of damage to agriculture in CBAs is fundamental, especially when the interventions involve floodplains devoted to agricultural activities, as it is typically the case of river restoration actions, included in "integrated river basin management" projects (Morris and Hess, 1988; Morris et al., 2008; Rouquette et al., 2011; Brémond et al., 2013; Massaruto and De Carli, 2014; Guida et al., 2016). Clearly, the tool must be critically used, e.g. by considering possible transfers of losses/gains between farmers in an economic perspective, according to the temporal and spatial scales of the analysis.

~~involve floodplains devoted to agricultural activities as it is typically the case of river restoration actions, included in "integrated river basin management" projects (Morris and Hess, 1988; Morris et al., 2008; Rouquette et al., 2011; Brémond et al., 2013; Massaruto and De Carli, 2014; Guida et al., 2016). On the opposite, the importance of developing new and reliable models for comprehensive flood damage assessments has been highlighted in recent investigations of past flood events (Pitt, 2008; Jongman et al., 2012; Menoni et al., 2016), showing that losses to the different sectors weigh differently according to the type of the event and the affected territory.~~

The development of AGRIDE-c and its implementation in the Po Plain highlighted that a thorough understanding and modelling of damage mechanisms to crops (i.e., of the interaction between damage influencing factors and characteristics of exposed elements leading to a loss) is also useful to orient the  behaviour of farmers towards more resilient practices, as the selection of the most resilient crops to be cultivated in areas prone to flood, the choice of the best alleviation strategy to be followed once flooded, the evaluation of the opportunity to ask for a flood insurance scheme and the definition of the premium.  For example, for the context and crop types investigated in the case study, Figure 6 highlights that abandoning the production is always the worst strategy, leading to a relative damage greater than 100% in any vegetative stage and for any flood intensity, due to the combined effect of the total loss of the gross output  (if excluding the EU contributions, obtained by the farmer also without any yield ) and of the costs incurred by the farmer before the flood. On the other hand, when flood intensity implies significant yield loss, reseeding (if possible) must be preferred to continuation, limiting the relative damage to 80%; nevertheless, the positive advantage of reseeding over continuation becomes smaller when including a yield penalty for late (re-)planting: results obtained by using the AGRIDE-c spreadsheet indicate a relative damage of 102% and 145% for a yield reduction of 10% and 30%, respectively.

 The model presents some limitations that must be addressed in future research works and must be carefully taken into account in its implementation. The first  is related to data requirements: the number and typology of input parameters  may prevent its use in data-scarce  areas. RegardingHowever,  it must be stressed that high-detailed tools like  AGRIDE-c  should be adopted only at an advanced stage of the analysis, when the costs of collecting site-specific data may be justified by the expected results (i.e. the choice of the best mitigation strategy);  in other cases, like  preliminary damage analyses for the identification of priority intervention areas or post-event assessments, rapid  tools (e.g. based on standardised damage/costs) should be preferred.

A second limitation concerns the high uncertainty characterising the input data required by AGRIDE-c, even in a specific context; for example , the estimation, based on few parameters (see Section 4.5), of the expected yield reduction due to late (re)seeding, which may be problematic as it is very variable and depends on many factors (among others, type of late hybrids used) . This implies that damage estimation may be affected by significant uncertainty, which is hardly quantifiable due to the limited availability of data for model validation (see Section 2); this uncertainty  can  be even amplified by the inherent uncertainty of the sub-models

implemented in AGRIDE-c, like , for example, the economic or the physical models for the estimation of flood damage to soil and crops, or the economic mode.

This suggests, as for other damage models, as for other damage models, the variability of parameters required by AGRIDE-c together with the limited availability of data for its validation (see Section 2) suggest the use of the model AGRIDE-c in a CBA context not in absolute terms (i.e. to evaluate the effectiveness of a specific measure), but as a tool to compare and choose among several alternatives (Scorzini and Leopardi, 2017; Molinari et al. 2019).

Likewise, a sensitivity analysis of input variables should always be performed, to get a flavour of robustness of findings. For example, for maize, the model developed for the Po Plain reveals (not shown here) that even a reduction of 10% of the yield in the Scenario 0 (with respect to the value adopted in the analysis) impacts the damage scenarios, leading to a relative damage greater than 100%, even in the case of reseeding in April and June and continuation in July and September (when yield loss is expected). The same occurs if the selling price decreases more than 12.5%, or EU contribution for the minimum tillage is not considered or production costs increase more than 10%. The "new" damage scenarios change the relative convenience associated to the different mitigation strategies; in particular, continuation may be more convenient that reseeding for short duration floods. Sensitivity analysis allows also investigating the effect on damage of possible changes in the physical and economic context in which the farm is located; in fact, all of the scenarios analysed in the previous example are globally representative of the context under investigation, but they can significantly vary among different farmers and different years: physical productivity is spatially non-uniform within the sub-regions of the Po Plain; prices and costs are highly variable in time and specific locations; only few farmers apply for EU contributions for the minimum tillage.

**Water depth < 130 cm**

| Phase | Month | Strategy | <5 | 5 | 6 | 7 | 8 | 9 | 10 | 11 | >11 |
|---|---|---|---|---|---|---|---|---|---|---|---|
| Bare field | Jan | c | | | | | 36% | | | | |
| | | r | | | | | - | | | | |
| | | a | | | | | - | | | | |
| | Feb | c | | | | | 36% | | | | |
| | | r | | | | | - | | | | |
| | | a | | | | | - | | | | |
| | Mar | c | | | | | 36% | | | | |
| | | r | | | | | - | | | | |
| | | a | | | | | - | | | | |
| Initial phase | Apr | c | | | | | - | | | | |
| | | r | | | | | 80% | | | | |
| | | a | | | | | 158% | | | | |
| | May | c | | | | | - | | | | |
| | | r | | | | | 80% | | | | |
| | | a | | | | | 158% | | | | |
| Growing | Jun | c | 36% | 63% | 90% | 117% | 144% | 171% | 198% | 225% | - |
| | | r | | | | | 80% | | | | |
| | | a | - | | | | 187% | | | | |
| Flowering | Jul | c | 36% | 90% | 144% | 198% | | | - | | |
| | | r | | | | | - | | | | |
| | | a | - | | | | 191% | | | | |
| | Aug | c | 36% | 90% | 144% | 198% | | | - | | |
| | | r | | | | | - | | | | |
| | | a | - | | | | 191% | | | | |
| Maturation | Sep | c | | | | | 36% | | | | |
| | | r | | | | | - | | | | |
| | | a | | | | | - | | | | |
| | Oct | c | | | | | 36% | | | | |
| | | r | | | | | - | | | | |
| | | a | | | | | - | | | | |
| Bare field | Nov | c | | | | | 36% | | | | |
| | | r | | | | | - | | | | |
| | | a | | | | | - | | | | |
| | Dec | c | | | | | 36% | | | | |
| | | r | | | | | - | | | | |
| | | a | | | | | - | | | | |

**Water depth ≥ 130 cm**

| Phase | Month | Strategy | <5 | 5 | 6 | 7 | 8 | 9 | 10 | 11 | >11 |
|---|---|---|---|---|---|---|---|---|---|---|---|
| Bare field | Jan | c | | | | | 36% | | | | |
| | | r | | | | | - | | | | |
| | | a | | | | | - | | | | |
| | Feb | c | | | | | 36% | | | | |
| | | r | | | | | - | | | | |
| | | a | | | | | - | | | | |
| | Mar | c | | | | | 36% | | | | |
| | | r | | | | | - | | | | |
| | | a | | | | | - | | | | |
| Initial phase | Apr | c | | | | | - | | | | |
| | | r | | | | | 80% | | | | |
| | | a | | | | | 158% | | | | |
| | May | c | | | | | - | | | | |
| | | r | | | | | 80% | | | | |
| | | a | | | | | 158% | | | | |
| Growing | Jun | c | 36% | 63% | 90% | 117% | 144% | 171% | 198% | 225% | - |
| | | r | | | | | 80% | | | | |
| | | a | - | | | | 187% | | | | |
| Flowering | Jul | c | | | | | - | | | | |
| | | r | | | | | - | | | | |
| | | a | | | | | 191% | | | | |
| | Aug | c | | | | | - | | | | |
| | | r | | | | | - | | | | |
| | | a | | | | | 191% | | | | |
| Maturation | Sep | c | | | | | 36% | | 90% | 144% | 198% | - |
| | | r | | | | | - | | | | |
| | | a | | | | - | | | | 195% | |
| | Oct | c | | | | | 36% | | 90% | 144% | 198% | - |
| | | r | | | | | - | | | | |
| | | a | | | | - | | | | 195% | |
| Bare field | Nov | c | | | | | 36% | | | | |
| | | r | | | | | - | | | | |
| | | a | | | | | - | | | | |
| | Dec | c | | | | | 36% | | | | |
| | | r | | | | | - | | | | |
| | | a | | | | | - | | | | |

**Figure 6. Po Plain case:.  relative damage (Eq. 2) to maize crops (in case of minimum tillage) for the different combinations times of occurrence of the flood (i.e. month), flood intensities (i.e. water depth and flood duration) and damage alleviation strategies ("c"=continuation; "r"=reseeding; "a"=abandoning. Results shown for the "r" option are obtained by assuming a null yield penalty for late (re-)plating.**

words, to perform a sensitivity analysis

5 ~~even in the case of reseeding in April and June (Figure 7) and continuation in July and September (when yield loss is expected). The same occurs (not shown here) if the selling price decreases more than 12.5%, or EU contribution for the minimum tillage is not considered or production costs increase more than 10%; all of these scenarios are realistic in the context under investigation, but they may change in time and space (i.e. lower yields have been observed in other subprices and costs are highly variable, while only few farmers apply for EU contributions for the minimum tillage) highlighting,~~

10 A third limitation concerns the time frame of the analysis, focused on one productive cycle; this prevents the comprehensiveness of the damage assessment by neglecting long-term indirect damages, like those related to the low productivity of soil in the following years after the flood event. This limitation must be carefully considered when the tool is implemented for the choice of risk mitigation strategies, as the expected damage can be significantly underestimated.

15 Finally, comprehensiveness of damage assessment is limited by the lack of consideration of other farm components which may be damaged in case of flood like damage to perennial plants, livestock, stock, equipment and machineries, buildings, permanent equipment and farm roads (Brémond et al., 2013; Posthumus et al., 2009; Morris and Brewin, 2014) as well as of their systemic interaction (i.e., damage induced to one component by another one). Further research is required on the topic as well as post-event data to calibrate and validate models

20 .

[revised manuscript text omitted]

15 **Competing interests**

The authors declare that they have no conflict of interest

---

## Referee Report (RR1)

**Manuscript nhess-2019-61 "AGRIDE-c, a conceptual model for the estimation of flood damage to crops:development and implementation**

**Reviewer 2 , 10th Oct 2019 reviewer reply to author responses and revised manuscript**

I note the responses by the authors to my comments and consider that they have addressed them for the most part satisfactorily.

A few points remain.  I leave it to the editor how best to address them.

Regarding Ordering of Contents.  I note the changes made, thankyou.

In my view table 1 is clearly not part of the introduction but part of methods and should go there, and there is not reference to it before it appears , and figure 6 should go in the results of the case not in discussions (The title to the table refers to results)

I note the response to my comment re listings .  In my view it is not possible to move from a conceptual model to an operational one without a list of metrics.  The chosen metrics define the scope of the model, and vice versa.

The authors might want to consider the approach towards harmonization of damage estimates for sources of hazards and major sectors such as :
Rios Diaz F., Marin Ferrer M., *Loss Database Architecture for Disaster Risk Management* EUR 29063 EN, Publication Office of the European Union, Luxembourg, 2018, ISBN 978-92-79-77752-3, doi:10.2760/647488, JRC110489.. This includes an Italian FLOODcat Application.

A few specifics :
P1 Line 28, insert  -and costs
P2, L1,   injury, loss of life, or their property?
P3 L 14, subsidies
P3, L16 insurance: this has definitely not been the case for crop damage in the UK
P4 L10 not clear what this refers to , you mean that 'in practice a proportion of yield is lost', ?

P9, L24, - perhaps should explain this is the  total value of farm outputs , and this applies to a production cycle ,  typically a year

P13, L13, see comments about the need to express these to a base year using 'constant' prices, eg 2015 values . not done here but should do it.  Guidance usually requires estimates expressed to a constant price base, eg 2018 prices. This is essential if the model outputs are used to guide investment decisions, especially during periods of high inflation.  Furthermore agricultural output prices may be inflating at different rates than flood infrastructure costs.  A comment on this might be useful, that annual prices series need to be adjusted to a constant price base to adjust for inflation if appropriate.

P13,13, make it clear these do not include automatic entitlements to direct farm income support

P13, L17 are these price books better referred to as   'Regional Farm Management Books'?

P20, L19, activities,  as it is typically the case of river restoration actions, included in "integrated river basin management" projects , better to rephrase as '

including integrated river basin management" project and river restoration actions

P20, L30, there may be no choice if reseeding is not feasible and some costs can be avoided

P21, L2, make it clear what the % refer to : NM

P21, L26 convenient = appropriate

Acknowledgements :

Suggest you thank the reviewers (not to name them) in order to recognise the importance and benefit of the review system

---

## Author Response (AR2)

**Manuscript nhess-2019-61 "AGRIDE-c, a conceptual model for the estimation of flood damage to crops: development and implementation" – Final response to comments by referees– second review**

We would like to thank all the referees for their appreciation of our work and for the insightful comments made in the whole review cycle, as they contributed to increase the manuscript robustness and to improve its quality and readability. We also thank the Editor for the precious work done in the coordination and supervision of the review process. In the following, we supply a point by point reply to the last comments raised by referee 2.

General and specific comments

**RC1:** In my view table 1 is clearly not part of the introduction but part of methods and should go there, and there is not reference to it before it appears, and figure 6 should go in the results of the case not in discussions (The title to the table refers to results)

**Answer:** The analysis reported in table 1 was conducted to support the need of the AGRIDE-c conceptual model in the current international panorama on flood damage modelling for the agricultural sector; according to us, this kind of discussion is appropriate for the introduction and is not related to the methods adopted for the development of AGRIDE-c. Table 1 is quoted at pg. 2, before it appears. Regarding Figure 6, it is presented at the end of Section 4, which is related to the case study, and simply recalled in the Discussion section. For these reasons, we think that no modifications are required.

**RC2:**
I note the response to my comment re listings. In my view it is not possible to move from a conceptual model to an operational one without a list of metrics. The chosen metrics define the scope of the model, and vice versa.

**Answer:** we thank the referee for the comment, on which we agree. In fact, Table 3 summarises input data required by AGRIDE-c and its exemplification for the Po Plain.

**RC3**: The authors might want to consider the approach towards harmonization of damage estimates for sources of hazards and major sectors such as: Rios Diaz F., Marin Ferrer M., Loss Database Architecture for Disaster Risk Management EUR 29063 EN, Publication Office of the European Union, Luxembourg, 2018, ISBN 978-92-79-77752- 3, doi:10.2760/647488, JRC110489. This includes an Italian FLOODcat Application.

**Answer:** we thank the reviewer for the suggestion. However, we think that the recommended publication is out of the scope of the manuscript, being related to good practices of harmonisation and standardisation of multi-hazard databases of ex-post event data; for this reason, we prefer not quoting the document in our manuscript. However, if appropriate/required by the Editor, we can quote the document somewhere in the introduction or discussion section.

**RC4:** P1 Line 28, insert -and costs

**Answer:** Added

**RC5:** P2, L1, injury, loss of life, or their property s

**Answer:** Added

**RC6:** P3 L 14, subsidies

**Answer:** Corrected

**RC7:** P3, L16 insurance: this has definitely not been the case for crop damage in the UK

**Answer:** we are aware of this. However, given the international nature of the work (and, in particular, of the potential users of the conceptual model), we think it is not fair/necessary to specify the UK situation in the manuscript. Indeed, by referring to the European context, the paper states "that led most of public authorities responsible for damage compensation to be less interested in the agricultural sector".

**RC8:** P4 L10 not clear what this refers to, you mean that 'in practice a proportion of yield is lost',

**Answer:** yes, the sentence has been corrected and made clearer.

**RC9:** P9, L24, - perhaps should explain this is the total value of farm outputs, and this applies to a production cycle, typically a year

**Answer:** specified

**RC10:** P13, L13, see comments about the need to express these to a base year using 'constant' prices, e.g. 2015 values. not done here but should do it. Guidance usually requires estimates expressed to a constant price base, e.g. 2018 prices. This is essential if the model outputs are used to guide investment decisions, especially during periods of high inflation. Furthermore, agricultural output prices may be inflating at different rates than flood infrastructure costs. A comment on this might be useful, that annual prices series need to be adjusted to a constant price base to adjust for inflation if appropriate.

**Answer:** we thank the referee for this important remark on costs/prices adjustment that, if neglected, could lead to an improper use of the model. We added a sentence on this at the end of Section 4.2.

**RC11:** P13,13, make it clear these do not include automatic entitlements to direct farm income support

**Answer:** According to our knowledge, there are not automatic entitlements in Italy. Still, we have specified in the new version of the paper that EU contributions represents a "potential" income for the farmer.

**RC12:** P13, L17 are these price books better referred to as 'Regional Farm Management Books'?

**Answer**: we preferred to leave the reference as "price lists" (as suggested by the local experts) because we are sure neither on what referee means with "Regional Farm Management Books'" nor we are aware of the existence of such books in Italy.

**RC13:** P20, L19, activities, as it is typically the case of river restoration actions, included in "integrated river basin management" projects, better to rephrase as ' including integrated river basin management" project and river restoration actions

**Answer:** rephrased

**RC14:** P20, L30, there may be no choice if reseeding is not feasible and some costs can be avoided

**Answer**: we agree, this is reflected in the model

**RC15:** P21, L2, make it clear what the % refer to: NM

**Answer:** reference to NM has been added

**RC16 (P1.L20):** P21, L26 convenient = appropriate

**Answer:** corrected

**RC17:** Acknowledgements: Suggest you thank the reviewers (not to name them) in order to recognise the importance and benefit of the review system

**Answer:** we added acknowledgements to the Editor and the Reviewers

Remark for the editor

In order to further increase the accessibility of the supplementary spreadsheet, we have uploaded it on the Mendeley repository. The reference has been updated accordingly.

**AGRIDE-c, a conceptual model for the estimation of flood damage to crops: development and implementation**

Daniela Molinari[1], Anna Rita Scorzini[2], Alice Gallazzi[1], Francesco Ballio[1]

[1] Department of Civil and Environmental Engineering, Politecnico di Milano, Piazza Leonardo da Vinci 32, 20133, Milano, Italy

[2] Department of Civil, Environmental and Architectural Engineering, Università degli Studi dell'Aquila, Via Gronchi, 18, 67100, L'Aquila, Italy

*Correspondence to*: Daniela Molinari (daniela.molinari@polimi.it)

**Abstract.** This paper presents AGRIDE-c, a conceptual model for the assessment of flood damage to crops, in favour of more comprehensive flood damage assessments. Available knowledge on damage mechanisms triggered by inundation phenomena is systematised in a usable and consistent tool, with the main strength represented by the integration of physical damage assessment with the evaluation of its economic consequences on the income of the farmers. This allows AGRIDE-c to be used to guide the flood damage assessment process in different geographical and economic contexts, as demonstrated by the example provided in this study for the Po Plain (North of Italy). The development and implementation of the model highlighted that a thorough understanding and modelling of damage mechanisms to crops is a powerful tool to support more effective damage mitigation strategies, both at public and at private (i.e. farmers) level.

**1 Introduction**

On a global scale, floods are among the most common and damaging natural hazards (EEA, 2017, CRED, 2019). As climate change continues to exacerbate extreme meteorological events, flood prone areas and flood-related damages are expected to grow rapidly in the future (Van Alst, 2006; Wobus et al., 2017; Alfieri et al., 2018; Mechler et al., 2019). To cope with this increasing risk, the EU Floods Directive (Directive 2007/60/EC) requires Member States (and, in particular, River Basin Districts) to periodically develop Flood Risk Management Plans, which are the operational/normative tools for the definition of flood risk mitigation strategies, including a blend of structural and non-structural measures. These measures must be identified on the basis of a reliable and comprehensive assessment of costs and benefits related to the implementation of alternative strategies (Jonkman et al., 2004; Mechler, 2016), i.e. on cost-benefit analyses (CBAs), which implies a public choice based on the assessment of welfare change associated with public investments. In fact, CBAs would require a comprehensive estimation of the costs and benefits produced by the adoption of different strategies (Jonkman et al., 2004; Mechler, 2016), with benefits consisting in the avoided losses to all exposed sectors and at different temporal scales (i.e. direct and indirect/long term damages).

Present damage modelling capacity is mainly focused on direct damage to people (injury, loss of life) and their property (for some exposed assets, (typically residential buildings) thus preventing the possibility of performing comprehensive flood damage assessments and, consequently, CBAs (see e.g. Ballesteros-Cánovas et al., 2013; Saint-Geours et al., 2015; Meyer et al., 2013; Shreve and Kelman, 2014; Arrighi et al., 2018). On the opposite, the importance of developing new and reliable

5  models for more inclusive flood damage assessments has been highlighted in recent investigations of past flood events (Pitt, 2008; Jongman et al., 2012; Menoni et al., 2016), showing that losses to the different sectors weigh differently according to the type of the event and the affected territory. To partially cover this gap, this paper deals with the estimation of flood damage to the agricultural sector, by presenting a new conceptual model for the estimation of flood damage to crops.

In the literature on flood damage modelling, agriculture has received so far less attention than other exposed sectors, as
10  demonstrated in Table 1, showing the number of papers in the Scopus database for different research keywords. Reasons may include: (i) the (perceived) minor importance of agricultural losses compared to those of other sectors, especially because flood damage assessments are usually carried out in urban areas (Förster et al. 2008; Chatterton et al., 2016), (ii) the paucity of empirical data for understanding damage mechanisms and deriving prediction models, and finally, (iii) a policy shift, especially in Europe post 1980s, when the subsidies to agriculture were being challenged by the increase of agricultural surpluses under
15  the Common Agricultural Policy, along with the incentivisation of insurance coverage for damage to farms, that led most of public authorities responsible for damage compensation to be less interested in the agricultural sector. However, it must be stressed that flood risk management has been the concern of agricultural policies for many years, as since the 1930s, and probably up to the middle 1980s, agricultural policies were focused on land drainage (i.e. the removal of problems caused by the excess of water on/in the soil) of which flood protection was a critical part (Morris 1992; Morris et al. 2008). Still, literature
20  related to land drainage is often difficult to retrieve and did not converge in the more recent studies on flood damage modelling, as much of the work is reported in the grey literature (see e.g. Hallett et al. 2016).

Available damage models for agriculture are not only few in number, but are also affected by many limitations, the major being the paucity of information/data for their validation and the large variability of the local features affecting damage (i.e. the strong linkage with the context under investigation), which limit their transferability to different contexts more than other
25  exposed sectors as the residential and commercial ones; accordingly, the first requirement for a new damage model is its possible application in a wide variety of geographical and economic contexts. Experience gained in flood damage assessment for other sectors highlighted that a broad generalisation is often not possible, as damage models must be able to capture the specificities of the investigated area, both in terms of hazard and vulnerability features (Cammerer et al., 2013). Still, a general conceptualisation of the problem is conceivable in terms of main variables influencing the damage mechanisms, cause-effect
30  relationships, etc.

Based on these considerations, this paper presents AGRIDE-c (AGRIculture DamagE model for Crops), a conceptual model for the estimation of expected flood damage to crops (i.e. ex-ante estimation). AGRIDE-c has the ambition of generality, i.e. to be valid in different geographical and economic contexts, supplying a useful framework to be followed any time the estimation of flood damage to crops is required, in which the main components of the problem at stake are identified as well

as its relevant control parameters. While the model structure aims to be generally valid, the analytical expression of its components must necessarily be specific to the local physical characteristics of the area as well as to the standards of the agricultural practices and to the type of crops under analysis, given the large variability characterising the agricultural sector. The implementation of the conceptual framework of AGRIDE-c is exemplified in this paper in relation to the Po Plain - North

5    of Italy. The case study is completed with a spreadsheet (available as supplementary material  in Molinari et al., 2019b) for the calculation of damage to crops, which can be adapted to other contexts.

The paper is organised as follows. Section 2 reviews the state of art on flood damage modelling to crops, as the starting point of the research. Section 3 presents the AGRIDE-c model, while Section 4 describes in detail its implementation in the Po Plain. Section 5 provides a critical discussion on limits and strengths for the effective application of AGRIDE-c and conclusions are

10   finally drawn in Section 6.

Table 1. Papers in the Scopus database for different research keywords (last access: January 2019)

| Keyword search | Number of papers |
|---|---|
| "Flood damage" | 4036 |
| "Flood damage" AND "crop" | 81 |
| "Flood damage" AND "agriculture" | 71 |
| "Flood damage" AND "building" | 284 |
| "Flood damage" AND "infrastructure" | 122 |

**2 State of art on flood damage modelling for crops**

15   Prominent examples of damage models for crops are reported in Table 2. The analysis of the table indicates that main differences among models are related to the input variables describing the inundation scenario (hazard) as well as the response of the exposed elements to flooding (vulnerability). Beyond hazard parameters usually considered in damage modelling for other exposed sectors (i.e., water depth, flow velocity, flood duration, sediment and contaminant load), for crops a key role is played by the period of the year, generally the month of the flood event, as damage is strongly dependent on crop calendars

20   (USACE, 1985; Morris and Hess, 1988; Hussain, 1996; RAM, 2000; Citeau, 2003; Dutta et al., 2003; Förster et al., 2008; Agenais et al., 2013; Shrestha et al., 2013; Vozinaki et al., 2015; Klaus et al., 2016) that, in their turn, depend on the climate of a region: this is one of the reasons which makes damage models for crops strongly context specific. Indeed, crop calendars delineate the vegetative stage of the plants at the time of the flood (which strongly affects the damage suffered by the plants) for any crop type, the latter being the only vulnerability parameter often considered by the models. In the case of meso-scale

25   models (Kok et al., 2005; Hoes and Schuurmans, 2006), this parameter is replaced by the agricultural land-use. No model in Table 2 considers instead the behaviour of farmers after the occurrence of the flood (e.g. the decision of abandoning the

production or to continue with increasing production costs) which has been shown to strongly influence the damage sustained by the farm (Pangapanga et al., 2012; Morris and Brewin, 2014).

With respect to the approach, only few literature models are directly derived from field observations of flood consequences on crops: this is mainly due to the scarcity of observed damage data (Brémond et al., 2013; Chatterton et al., 2016) for models derivation/calibration. In fact, most of the models adopt a synthetic approach based on the expert investigation of causes and consequences of damage. In this regard, some models in Table 2 are labelled as "physically based", i.e., damage is first described in terms of physical susceptibility of the crop and consequent yield reduction, and then converted into economic impact on the income of the farmers. Instead, in "cost based" models damage is assessed only considering production costs sustained by farmers during the year, by implicitly assuming (according to our interpretation) that the yield is totally lost in case of flood, although in practice this not always happens (Posthumus et al., 2009; Penning-Rowsell et al., 2013; Morris and Brewin, 2014). 
[revised manuscript text omitted]
. More specifically, experts were involved to support the definition of the conceptual model, by following an iterative process. In the first step of the process, a semi-structured interview was conducted, by asking experts about the main damage mechanisms/phenomena for crops in case of flood, important explicative variables and possible interconnections among them; moreover, results from the literature review were proposed for their judgment. In the following step, experts were asked to evaluate a draft version of the conceptual model drawn according to the literature review and results

from first interviews. Then, there was an iterative revision of improved versions of the model until an agreement on its final structure was reached. Three kinds of experts were involved in the process: (i) a representative of one of the Italian regional authorities responsible for agricultural damage management and compensation, with more than 20 years of expertise in the management and compensation of flood damage to farms in the Lombardy Region; (ii) two agronomists of a local association
5    of farmers (Coldiretti Lodi), with specific knowledge on the Po Plain context and with direct experience in managing floods in the last 20 years; the viewpoint of several individual local farmers who experienced flooding in the past years was also included in the analysis, as the two agronomists asked them for direct data and information to support their considerations; (iii) an academic economist, with specific expertise in agriculture.

It must be highlighted that the conceptual model has been designed to supply an estimation of flood damage only to annual
10   crops (i.e., not including perennial crops) under the following assumptions:

- infrequent flooding events (i.e., effect of two, or more, consecutive floods is not considered);

- flooded agricultural plot devoted to a single crop type, with possible reseeding using the same crop type in case of flood;

- time frame of the analysis limited to one productive cycle: long term damages, in particular, loss reduction of soil productivity in the following cycles is not considered;

15   In addition, AGRIDE-c does not consider damage to other components/elements of the farm that, on turn, may induce additional damage to crops, as, for instance, damage to machineries and equipment (e.g. irrigation system) that may prevent cultivation for a while (Dunderdale and Morris, 1997; Posthumus et al., 2009; Agenais et al., 2013; Bremond et al., 2013; Morris and Brewin, 2014). Only short term impacts on soil are included, based on the evidence that, during a flood, damages to soil and crops are concurrent, differently from damages to the other components which can occur or not, independently from
20   the damage to the vegetal material; as a consequence, damage to soil and crops is modelled together, while damage to the other components can be modelled as separated factors.

The model structure is depicted in detail in Figure 2. Absolute damage (D) for an individual farmer is expressed as the difference between the reduction in the gross output ($\Delta$GO) and the increase/decrease in production costs ($\Delta$PC), as a consequence of the flood of a specific crop. This is equal to consider absolute damage as the change in the net margin (NM =
25   GO– PC, where GO =is gross output and PC =are production costs over a production cycle, typically a year) due to the flood, compared to the case when no flood occurs (i.e., Scenario 0):

$$D = NM_{noflood} - NM_{flood =} (GO_{noflood}-GO_{flood}) - (PC_{noflood}-PC_{flood}) = \Delta GO - \Delta PC \qquad (1)$$

Accordingly, relative damage (d) can be obtained by dividing the absolute damage by the net margin in the Scenario 0 ($NM_{noflood}$)

30   $$d = D/NM_{noflood} = 1 - NM_{flood}/NM_{noflood} \qquad (2)$$

[Figure]

**Figure 2. Conceptual model of AGRIDE-c**

AGRIDE-c combines a physical and an economic model to evaluate the absolute damage. In this way, the problems of consistency among physically-based and/or cost-based models discussed in Section 2 are overcome, being both aspects explicitly taken into account.

The physical model (identified by the yellow dashed box in Figure 2) is composed of two sub-models, for the evaluation of
5    physical damage to crops (i.e. the plants) and impact on soil, respectively. In fact, as previously stated, among the different components/elements of the farm that may induce damage to crops, only damage to soil is considered in AGRIDE-c.

The model for the assessment of physical damage to soil calculates the amount of soil that is damaged, the kind(s) of damage suffered by the soil and the reduction of soil fertility, as a function of the duration of the flood, the water velocity, the sediment, the salinity (in case of coastal flooding) and the contaminants load. In particular, the model takes into account of processes
10    like erosion, deposition of sediments and contamination (which affect the costs for soil restoration), as well as of the soil fertility (which affects the quality and the quantity of the harvest). In addition, the model estimates the effect of possible waterlogging, as a consequence of an increase in the level of the field water table, in terms of soil fertility reduction and (prolonged) soil saturation, which may increase costs for restoration because of the necessity of land drainage. It must be noted that, although in the European context floods usually have a negative effect on soils, some studies (e.g., Tockner et al., 1999;
15    Hein et al., 2003) pointed out that such events can also have clearly positive effects, namely in the form of an increase of soil fertility, explained by a (re-)distribution of river sediments and organic matter in the course of flooding that replenish carbon and nutrients in topsoil.

[revised manuscript text omitted]

Table 3 summarises the main general data required by the conceptual model and the values / information used in the application for the Po Plain (example of maize).  Data sources are clarified in the following sub-sections.

The implementation of the conceptual model to Po Plain was supported by specific knowledge of local experts. In particular,

20 several individual meetings were organised with the aim of obtaining context-specific information related on crop calendars, yields and prices, type, timing and costs of the different cultivation practices.

**4.1 Hazard and vulnerability features in the Po Plain**

In order to identify the representative features of the floods and the main crops cultivated in the investigated area, we chose the Province of Lodi (Lombardia Region) as representative of hazard phenomena and agricultural activities in the Po Plain.

25 The last significant event occurred in the province, i.e. the flood of the Adda River in November 2002 (AdBPo, 2003; AdBPo, 2004; Rossetti et al., 2010; Scorzini et al., 2018), highlighted riverine long-lasting floods, characterised by medium to high water depths (mean value: 0.9 m), low flow velocities (mean value: 0.2 m/s) and low sediment and pollution loads in the flooded areas as typical of the region; accordingly, main hazard parameters to be included in the analytical expression of AGRIDE-c for the Po Plain are limited to water depth, flood duration and time (month) of flood occurrence.

The analysis of the agricultural cadastral data (supplied by the Regional Authority) in a buffer of 1 km around the Adda River, indicated grain maize, wheat, barley and grassland as the most common crops in the area; the model for maize is discussed hereinafter, while those related to other crops are reported in the supplement.

**Table 3. Summary of input data required by AGRIDE-c: exemplification for the Po Plain**

| Conceptual model | | Implementation for the Po Plain (example of maize) | |
|---|---|---|---|
| | *Input parameters* | *Modelling and input values* | *Data sources* |
| ***Physical Model*** | | | |
| Damage to crop | As shown in Fig.2 | Transferred and adapted from Agenais et al. (2013) | Agenais et al. (2013) and experts consultation |
| Impact on soil | As shown in Fig.2 | Soil restoration considered as a fixed cost (500 €/ha) | APIMA (2013-2017) and experts consultation |
| ***Economic Model*** | | | |
| *Gross output* | Crop yield | 175 q/ha | Regione Lombardia (2013-2017) |
| | Unit price for crop | 16.9 €/q | Borsa Granaria di Milano (2013-2017) |
| | Other (e.g. EU contributions) | 150 €/ha for crop rotation; 300€/ha for minimum tillage | PSR Regione Lombardia |
| *Production costs* | | | |
| Variable costs | Depend on crop type and cultivations practises / strategies | As shown in Fig.3 and Tab. 4 | APIMA (2013-2017) and experts consultation |
| Fixed costs | | Assumed equal to 5% of the gross output | Experts consultation |

**4.2 Characterisation of the Scenario 0**

The Scenario 0 is characterised in terms of the annual net margin for the farmer, per hectare, in the case no flood occurs; this implies the estimation of the annual gross output and the distribution of production costs over the year.

Given that the vegetative cycle of grain maize in the Po Plain covers one year, the gross output is estimated as the product between the average yield and price for grain maize over the period 2013-2017 (data sources: Regione Lombardia and Borsa Granaria di Milano (-Milan Crops Stock Market)), equal to 175 q/ha and 16.92 €/q, respectively. In addition, we also consider the annual EU contributions for agriculture as a further potential income for the farmer and, in detail, the subsidies given to agricultural activities in case of the application of minimum tillage and crop rotation, equal respectively to 300 and 150 €/ha (data source: PSR - Programma di Sviluppo Rurale, Regione Lombardia: http://www.psr.regione.lombardia.it).

Concerning production costs, the type, period of the year and costs of the different cultivation practices for grain maize were identified with the support of discussions with experts and consultation of regional price books (data source: APIMA – Associazione Provinciale Imprese di Meccanizzazione Agricola delle Province di Milano, Lodi, Como, Varese: Tariffe 2013-2017 delle lavorazioni meccanico agricole c/terzi, i.e., price lists for agricultural operations by contractors). All agricultural operations have been considered as direct, avoidable costs, as interviewed local experts indicated that in Lodi province most

of field operations are carried out by contractors. Figure 3 reports the distribution of costs over the year, with indication of the corresponding vegetative stages of the plant.

[Figure]

     **Figure 3. Po Plain case: production costs over the year for grain maize, in case of minimum tillage technique**

Finally, fixed costs sustained by farmers (like management costs) are assumed to be a portion (5%) of the gross output. Based on these data, the analysis results in a net margin for the famer in case of no flood equal to 1376 €ha per year.

It is important to stress that, in case of application of AGRIDE-c as a tool for supporting investment decisions, both costs and
10   prices need to be adjusted to a common price base (year $N$) in order to account for the effect of inflation, if appropriate.

**4.3 Damage to crops**

Physical damage to crops is estimated by the physical model developed in France by Agenais et al. (2013). This choice is supported by different considerations. First, the independent hazard variables considered by the authors (for maize: water depth and flood duration) are coherent with the typical flooding characteristics identified for the Po Plain (Section 4.1), i.e. riverine
15   long-lasting floods with low flow velocity. Second, their model can be easily transferred to other regions, independently from crop calendars, as they use the vegetative phases of the crop (and not the months of the year) as the time variable for the occurrence of the flood. Finally, local agronomists expressed a favourable opinion on the suitability of this model in the examined region, as emerged from discussions held during the interview process.

An example of the physical damage model for maize is depicted in Figure 4 (adapted from Agenais et al., 2013). The model
20   consists of susceptibility functions giving the yield reduction due to the flood (as a percentage of the yield in the Scenario 0), on the basis of water depth and flood duration, for four different vegetative stages (i.e. seeding, growing, flowering and maturation). Let us consider, for example, the growing stage: for a flood lasting less than 5 days the model gives a null yield

loss, independently from the water depth; on the opposite, a flood lasting more than 12 days results in a total loss. For floods with intermediate duration, in absence of specific information in the original model and in accordance with the opinion of local experts, we assumed a linear yield reduction (from 0 to 100%) between 5 and 12 days, adapting the model to the context under investigation. The use of this model implies that, at present, we do not take into account neither the reduction in the quality of the yield due to the flood nor the effect of damage to soil (i.e. reduction of soil fertility) on yield quality and production; reason for such limitations is simply the lack of literature and data on these topics (see also Section 4.4).

**4.4 Impact on soil**

Concerning the physical impact on soil, only the negative effects of floods were computed as, according to local experts, increase in soil fertility due to floods is infrequent in Northern Italy. Likewise, waterlogging after floods is not relevant in the investigated area and has been neglected.

For the estimation of physical damage to soil, no models were found in the literature investigating the complex chemical and mechanical processes leading to soil erosion, contamination and asphyxiation due to sediment deposition; also interviewed experts were not able to parametrise the possible types of damage, the amount of damaged soil and the reduction in soil fertility as a function of hazard features. For these reasons, at present, the model is based on the simplified assumption that soil always requires restoration in case of flood (consisting in the removal of sediments and in the levelling of terrain) and that no reduction in soil fertility occurs. Indeed, in the context under investigation, erosion and contamination are not expected because of the low velocity and limited contaminant load characterising typical floods in the region (see Section 4.2).

The choice to include the damage to soil component in the implementation of AGRIDE-c, although in this simplified way, was driven by two main reasons: comprehensiveness of the model and importance of this sub-component in the overall flood damage figure to agriculture. In particular, this last point clearly emerged during the interviews with local experts, who pointed out the occurrence of such damages even for flood events characterised by shallow water depths and not particularly high flow velocities. According to estimation of necessary operations supplied by interviewed experts and regional price books (data source: APIMA), restoration costs have been considered here, in a first instance, as fixed costs equal to 500 €/ha.

**4.5 Alleviation strategies**

After the recession of the flood, farmers make a choice among the possible strategies that can be adopted to alleviate damage; literature investigation and discussions with experts indicated three main strategies, their feasibility being necessarily linked to the damage suffered by the plants which, in its turn, depends on the flood intensity and the vegetative stage of the plants at the occurrence of the flood: continuing the production, abandoning the production, reseeding. The choice among these strategies influences both yield reduction and production costs, because of additional or avoided cultivation practices consequent the continuation or the abandon of the production; such practices and related costs have been identified for the Po Plain, with the support of experts and regional price books (Table 4).

[Figure]

**Figure 4. Physical damage to maize as a function of vegetative stage, flood depth and duration (adapted from Agenais et al., 2013)**

**Table 4. Yield reduction and change in production costs for grain maize on the basis of damage alleviation strategy adopted by farmer**

| Time of the flood | Vegetative stage | Alleviation strategy | Yield reduction [%] | Additional costs | €/ha | Avoided costs | €/ha |
|---|---|---|---|---|---|---|---|
| November - March | Bare field | Continuation | 0 | Soil restoring | 500 | | |
| April - May | Initial phase | Abandoning | 100 | Soil restoring | 500 | Weeding and fertilising | 387 |
| | | | | | | Irrigation | 110 |
| | | | | | | Harvesting and drying | 783 |
| | | Reseeding | 0 | Soil restoring ( | 500 | | |
| | | | | Strip till and fertilising | 168 | | |
| | | | | Seeds and reseeding | 438 | | |
| June | Growing phase | Continuation | see Fig. 4 | Soil restoring | 500 | | |
| | | Abandoning | 100 | Soil restoring | 500 | Irrigation | 110 |
| | | | | | | Harvesting and drying | 783 |
| | | Reseeding | 0 | Soil restoring | 500 | | |
| | | | | Strip till and fertilising | 168 | | |
| | | | | Seeds and reseeding | 438 | | |
| July - August | Flowering phase | Continuation | see Fig. 4 | Soil restoring | 500 | | |
| | | Abandoning | 100 | Soil restoring | 500 | Irrigation | 55 |
| | | | | | | Harvesting and drying | 783 |
| September - October | Maturation phase | Continuation | see Fig. 4 | Soil restoring | 500 | | |
| | | Abandoning | 100 | Soil restoring | 500 | Harvesting and drying | 783 |

Continuing the flooded crops is suggested when flood damage implies none or minor yield loss; in this case, yield reduction

5   is equivalent to that supplied by the physical model of Figure 4 as a function of hazard features, while additional costs are only due to soil restoring (see Section 4.4). Abandoning the production can be an option when flood damage is severe. This strategy always leads to a 100% yield reduction; soil restoration is still required, but some production costs can be avoided according to the time of the occurrence of the flood (i.e. remaining time to harvest). Reseeding is an alternative strategy to abandoning when flood damage is severe, but it is possible only until June, by using late maize crops. Results presented in this paper are

10   obtained, by adopting the simplified assumption that late reseeding does not imply a yield reduction, neither in quantity nor in quality. In fact, the use of late crops generally implies a yield reduction with respect to traditional crops, reduction that increases as the time of reseeding approaches the maturation phase, and that varies with the different species of late crops and climates, generally ranging from 10% to 30% (Lauer et al., 1999; Tsimba et al., 2013; Dobor et al., 2016; Abendroth et al., 2017). Given the high variability of yield loss with these two variables (i.e. time and species), a reference value was not identified in the

15   literature neither in discussion with experts; however, users of AGRIDE-c have the option to set a proper value of the expected yield reduction for late (re-)planting for the context under investigation, in the spreadsheet supplied as supplementary material (https://tinyurl.com/yyj2arhpMolinari et al., 2019b ). Beyond additional costs required to restore the flooded soil, reseeding implies further additional costs related to the preparation of the terrain, the purchase of new seeds and the seeding operations.

**4.6 Damage estimation**

According to the conceptual model in Section 3 and assumptions described in the previous sub-sections, damage (D) is estimated for different times of occurrence of the flood (i.e. month), flood intensities (i.e. water depth and flood duration) and damage alleviation strategies, as the difference between $\Delta GO$ and $\Delta PC$:

5    $D = D$ (month, water depth, flood duration, alleviation strategy) $= \Delta GO - \Delta PC$         (3)

In detail, $\Delta GO$ and $\Delta PC$ are calculated on the basis of yield reduction and additional and avoided costs, as reported in Table 4. The resulting damage function has a fixed component due to soil restoration costs, to be added to the costs which varies with the flood characteristics and the alleviation strategy.

As an example of damage estimation, -Figure 5 shows changes in production costs and gross output for maize cultivation, for

10    three different flood scenarios. Values of the annual gross output and of cumulative production costs are reported for both Scenario 0 and the flood scenario under investigation, with respect to every alleviation strategy farmers can implement according to the intensity of the flood, its time of occurrence and the physical damage suffered by the plant. Differences of production costs and turnover between "flood" and "no flood" scenarios allow calculating the damage D for the farmer.

The first scenario (Figure 5a) refers to a November flood. In this month, the plant is in the break stage, so no yield loss is

15    expected for any flood intensity (Table 4). Farmers will then continue the production with additional costs limited to those required to restore the flooded soil for a total of 500 €/ha (Table 4), which is the absolute damage sustained by farmers.

The second scenario (Figure 5b) refers to a flood in June, when the plant is in the growing stage. According to the physical model described in Figure 4, in this phase damage depends only on flood duration, while water depth has no effect on it. Figure 5b refers to a 5 days flood which leads, as given by the physical model, to a yield reduction of 12.5%. Given the low physical

20    damage, farmers can decide to continue the production or to reseed. In the first case (green line), the gross output decreases by 12.5% (due to yield reduction), while production costs increase due to additional costs for soil restoration, resulting in an absolute damage for the farmer equal to about 870 €/ha. In the second case (blue line), no reduction in the gross output occurs because reseeding would allow 100% of the yield, while additional production costs include both soil restoration and reseeding costs, resulting in an absolute damage of 1106 €/ha. Figure 5b shows that, although possible in theory, abandoning the

25    production is not a reasonable choice as absolute damage equals 2568 €/ha, due to a yield reduction of 100% (the only income for the farmer consists in the EU contributions for cultivation) against a saving of production costs of about 389 €/ha.

Finally, Figure 5c refers to a flood occurring in September; in this period (i.e. maturation phase of the plant), damage depends on both water depth and flood duration. Figure 5c refers in particular to a 10 days flood with a water depth above 1.30 m. According to the physical model (Figure 4), this flood scenario leads to a 50% yield loss. Farmers have then two choices.

[Figure]

a) November flood (vegetative stage: break): any flood depth and duration

b) June flood (vegetative stage: growing): any flood depth and 5 days duration (yield loss 12.5%)

[Figure]

c) September flood (vegetative stage: maturation): flood depth > 1.30 m and 10 days duration (yield loss 50%)

**Figure 5. Po Plain case: distribution of cumulative production costs for grain maize during the year and annual gross output and net margin in the scenario 0 and in the case of a flood occurring in different months. Colours refers to the different possible strategies the farmer can adopt according to: the time of occurrence of the flood, intensity (water depth and duration) and physical damage. The absolute damage for the farmer (Di) is obtained by the difference of the net margin in the Scenario 0 and in the investigated scenario, as exemplified in Figure 5a.**

If production is continued the gross output decreases by 50% and additional costs are required to restore the flooded soil, resulting in an absolute damage equal to 1980 €/ha. In case of abandoning, absolute damage equals 2677 €/ha, because of a yield reduction of 100% and saving of production costs of 283 €/ha.

Previous considerations can be repeated for the different months of the year and hazard scenarios. Figure 6 displays the ensemble of the results of damage estimation for all the investigated cases, thus defining the AGRIDE-c model for the Po Plain, for grain maize crops. In particular, the figure reports the relative damage with respect to the net margin in case of no inundation, $d=D/NM_{noflood}$, estimated by the model, for the different months of flood occurrence, flood intensities (i.e. water depth and flood duration) and damage alleviation strategies. The "dash" symbol means that the corresponding strategy cannot be adopted or is not reasonable in the flood scenario under investigation. For example, in the "bare field" season, reseeding is not possible because of climatic reasons, nor it is continuation as no cultivation is in place; continuation does not make sense when a 100% yield loss is expected as in the "initial phase" or in the "flowering" stage when h≥ 1.3 m; reseeding with late crops is possible only until June, etc. Equivalent tables for the other investigated crops are reported in the Supplement.

**5 Discussion**

The AGRIDE-c model, by enabling the estimation of the expected direct damage to crops in case of flood, represents a powerful tool to support more informed decisions on flood risk management for both public and private stakeholders. AGRIDE-c contributes to overcome the limitations of present CBAs, by providing a more comprehensive estimation of flood damages, thus supporting a better definition and choice of public actions for risk mitigation. In addition, the inclusion of damage to agriculture in CBAs is fundamental, especially when the interventions involve floodplains devoted to agricultural activities, including "integrated river basin management" projects and river restoration actions (Morris and Hess, 1988; Morris et al., 2008; Rouquette et al., 2011; Brémond et al., 2013; Massaruto and De Carli, 2014; Guida et al., 2016). Clearly, the tool must be critically used, e.g. by considering possible transfers of losses/gains between farmers in an economic perspective, according to the temporal and spatial scales of the analysis.

The development of AGRIDE-c and its implementation in the Po Plain highlighted that a thorough understanding and modelling of damage mechanisms to crops (i.e., of the interaction between damage influencing factors and characteristics of exposed elements leading to a loss) is also useful to orient the behaviour of farmers towards more resilient practices, as the selection of the most resilient crops to be cultivated in areas prone to flooding, the choice of the best alleviation strategy to be followed once flooded, the evaluation of the opportunity to ask for a flood insurance scheme and the definition of the premium. For example, for the context and crop types investigated in the case study, Figure 6 highlights that abandoning the production is always the worst strategy, leading to a relative damage greater than 100% in any vegetative stage and for any flood intensity, due to the combined effect of the total loss of the gross output (if excluding the EU contributions, obtained by the farmer also without any yield) and of the costs incurred by the farmer before the flood. On the other hand, when flood intensity implies

significant yield loss, reseeding (if possible) must be preferred to continuation, limiting the relative damage to 80% (where "relative" refers to NM, according to Eq. 2); nevertheless, the positive advantage of reseeding over continuation becomes smaller when including a yield penalty for late (re-)planting: results obtained by using the AGRIDE-c spreadsheet indicate a relative damage of 102% and 145% for a yield reduction of 10% and 30%, respectively.

5    The model presents some limitations that must be addressed in future research works and must be carefully taken into account in its implementation. The first is related to data requirements: the number and typology of input parameters may prevent its use in data-scarce areas. However, it must be stressed that high-detailed tools like AGRIDE-c should be adopted only at an advanced stage of the analysis, when the costs of collecting site-specific data may be justified by the expected results (i.e. the choice of the best mitigation strategy); in other cases, like preliminary damage analyses for the identification of priority

10   intervention areas or post-event assessments, rapid tools (e.g. based on standardised damage/costs) should be preferred.

A second limitation concerns the high uncertainty characterising the input data required by AGRIDE-c, even in a specific context. An example is the estimation, based on few parameters (see Section 4.5), of the expected yield reduction due to late (re)seeding, which may be problematic as it is very variable and dependant on many factors (among others, type of late hybrids used). This implies that damage estimation may be affected by significant uncertainty, which is hardly quantifiable due to the

15   limited availability of data for model validation (see Section 2); this uncertainty can be even amplified by the inherent uncertainty of the sub-models implemented in AGRIDE-c, like the economic or physical models for the estimation of flood damage to soil and crops.

This suggests, as for other damage models, the use of AGRIDE-c in a CBA context not in absolute terms (i.e. to evaluate the effectiveness of a specific measure), but as a tool to compare and choose among several alternatives (Scorzini and Leopardi,

20   2017; Molinari et al. 2019a).

Likewise, a sensitivity analysis of input variables should always be performed, to get a flavour of robustness of findings. For example, for maize, the model developed for the Po Plain reveals (not shown here) that even a reduction of 10% of the yield in the Scenario 0 (with respect to the value adopted in the analysis) impacts the damage scenarios, leading to a relative damage greater than 100%, even in the case of reseeding in April and June and continuation in July and September (when yield loss is

25   expected). The same occurs if the selling price decreases more than 12.5%, or EU contribution for the minimum tillage is not considered or production costs increase more than 10%. The "new" damage scenarios change the relative convenience associated to the different mitigation strategies; in particular, continuation may be more  appropriate  than reseeding for short duration floods. Sensitivity analysis allows also investigating the effect on damage of possible changes in the physical and economic context in which the farm is located; in fact, all of the scenarios analysed in the previous example

30   are globally representative of the context under investigation, but they can significantly vary among different farmers and different years: physical productivity is spatially non-uniform within the sub-regions of the Po Plain; prices and costs are highly variable in time and specific locations; only few farmers apply for EU contributions for the minimum tillage.

**Water depth < 130 cm**

| Phase | Month | Strategy | <5 | 5 | 6 | 7 | 8 | 9 | 10 | 11 | >11 |
|---|---|---|---|---|---|---|---|---|---|---|---|
| Bare field | Jan | c | 36% | | | | | | | | |
| | | r | - | | | | | | | | |
| | | a | - | | | | | | | | |
| | Feb | c | 36% | | | | | | | | |
| | | r | - | | | | | | | | |
| | | a | - | | | | | | | | |
| | Mar | c | 36% | | | | | | | | |
| | | r | - | | | | | | | | |
| | | a | - | | | | | | | | |
| Initial phase | Apr | c | - | | | | | | | | |
| | | r | 80% | | | | | | | | |
| | | a | 158% | | | | | | | | |
| | May | c | - | | | | | | | | |
| | | r | 80% | | | | | | | | |
| | | a | 158% | | | | | | | | |
| Growing | Jun | c | 36% | 63% | 90% | 117% | 144% | 171% | 198% | 225% | - |
| | | r | 80% | | | | | | | | |
| | | a | - | 187% | | | | | | | |
| Flowering | Jul | c | 36% | 90% | 144% | 198% | - | | | | |
| | | r | - | | | | | | | | |
| | | a | - | 191% | | | | | | | |
| | Aug | c | 36% | 90% | 144% | 198% | - | | | | |
| | | r | - | | | | | | | | |
| | | a | - | 191% | | | | | | | |
| Maturation | Sep | c | 36% | | | | | | | | |
| | | r | - | | | | | | | | |
| | | a | - | | | | | | | | |
| | Oct | c | 36% | | | | | | | | |
| | | r | - | | | | | | | | |
| | | a | - | | | | | | | | |
| Bare field | Nov | c | 36% | | | | | | | | |
| | | r | - | | | | | | | | |
| | | a | - | | | | | | | | |
| | Dec | c | 36% | | | | | | | | |
| | | r | - | | | | | | | | |
| | | a | - | | | | | | | | |

**Water depth ≥ 130 cm**

| Phase | Month | Strategy | <5 | 5 | 6 | 7 | 8 | 9 | 10 | 11 | >11 |
|---|---|---|---|---|---|---|---|---|---|---|---|
| Bare field | Jan | c | 36% | | | | | | | | |
| | | r | - | | | | | | | | |
| | | a | - | | | | | | | | |
| | Feb | c | 36% | | | | | | | | |
| | | r | - | | | | | | | | |
| | | a | - | | | | | | | | |
| | Mar | c | 36% | | | | | | | | |
| | | r | - | | | | | | | | |
| | | a | - | | | | | | | | |
| Initial phase | Apr | c | - | | | | | | | | |
| | | r | 80% | | | | | | | | |
| | | a | 158% | | | | | | | | |
| | May | c | - | | | | | | | | |
| | | r | 80% | | | | | | | | |
| | | a | 158% | | | | | | | | |
| Growing | Jun | c | 36% | 63% | 90% | 117% | 144% | 171% | 198% | 225% | - |
| | | r | 80% | | | | | | | | |
| | | a | - | 187% | | | | | | | |
| Flowering | Jul | c | - | | | | | | | | |
| | | r | - | | | | | | | | |
| | | a | 191% | | | | | | | | |
| | Aug | c | - | | | | | | | | |
| | | r | - | | | | | | | | |
| | | a | 191% | | | | | | | | |
| Maturation | Sep | c | 36% | | | | | 90% | 144% | 198% | - |
| | | r | - | | | | | | | | |
| | | a | - | | | | | 195% | | | |
| | Oct | c | 36% | | | | | 90% | 144% | 198% | - |
| | | r | - | | | | | | | | |
| | | a | - | | | | | 195% | | | |
| Bare field | Nov | c | 36% | | | | | | | | |
| | | r | - | | | | | | | | |
| | | a | - | | | | | | | | |
| | Dec | c | 36% | | | | | | | | |
| | | r | - | | | | | | | | |
| | | a | - | | | | | | | | |

**Figure 6. Po Plain case: relative damage (Eq. 2) to maize crops (in case of minimum tillage) for the different combinations times of occurrence of the flood (i.e. month), flood intensities (i.e. water depth and flood duration) and damage alleviation strategies ("c"=continuation; "r"=reseeding; "a"=abandoning. Results shown for the "r" option are obtained by assuming a null yield penalty for late (re-)plating.**

A third limitation concerns the time frame of the analysis, focused on one productive cycle; this prevents the comprehensiveness of the damage assessment by neglecting long-term indirect damages, like those related to the low productivity of soil in the following years after the flood event. This limitation must be carefully considered when the tool is implemented for the choice of risk mitigation strategies, as the expected damage can be significantly underestimated.

5    Finally, comprehensiveness of damage assessment is limited by the lack of consideration of other farm components which may be damaged in case of flood like damage to perennial plants, livestock, stock, equipment and machineries, buildings, permanent equipment and farm roads (Brémond et al., 2013; Posthumus et al., 2009; Morris and Brewin, 2014) as well as of their systemic interaction (i.e., damage induced to one component by another one). Further research is required on the topic as well as post-event data to calibrate and validate models.

10   The development of AGRIDE-c highlighted some challenges for the hydrology and the hydraulic community. In fact, application of the model requires a relatively detailed set of hazard input variables which are often not supplied in existing flood hazard maps (de Moel et al., 2009). Such knowledge would require a shift from traditional 1D steady hydraulic models to 2D unsteady hydraulic models - coupled with suitable sediment and contaminant transport models - in all flood prone areas, which is not easily achievable in a short time, both for technical and economic constraints. Thus, rapid approximate methods

15   for the estimation of hydraulic variables of interest should be developed (e.g. Scorzini et al., 2018). In addition, a further problem arises with respect to the estimation of the probability of occurrence of the different inundation scenarios. Given the importance of the time of the year, risk estimates should be based not only on annual probabilities, but also on seasonal probabilities (Förster et al., 2008; Klaus et al., 2016; Morris and Hess, 1988; USACE, 1985); this would imply changing present conceptualisation of flood return periods. It is worth noting that the key role played by the time of the event affects

20   also the identification of crops of interest, as the risk analysis should take into account which crops are actually in place when the event occurs. In fact, because of rotation techniques, it may happen that several different crops can exist on the same plot at different times of the year.

**6 Conclusions**

This paper presented AGRIDE-c, a conceptual model for assessing flood damage to crops and its implication for farmers. The

25   model has been exemplified in the Po Plain – North of Italy, for which a spreadsheet (partly customizable by users) for the calculation of damage has been also developed.

By organising the available knowledge on flood damage to crops in a usable and consistent tool that integrates physical and economic approaches, AGRIDE-c constitutes an advancement in flood damage modelling, supplying a general framework that can potentially be applied across different geographical and economic contexts. This aspect is the main strength of the model,

30   given the fragmented and not consolidated literature on the topic. On the other hand, the development of the model highlighted different challenges for the scientific community to achieve reliable estimations of flood damage to crops. Indeed, the exercise carried out for the Po Plain pointed out that further investigations on the modelling of damage mechanisms are required to

fully implement AGRIDE-c in a specific context: at present, (over)simplifications are made, for instance, regarding the physical damage to soil and its effect on crops or the influence of flood intensity on yield quality reduction.

Despite current limitations, the case study demonstrates the usability of the conceptual model; at the same time, it represents an example of how the model can be adapted to different geographical or economic contexts, given that all the assumptions and hypotheses made in the sub-models are clearly described; importantly, the model is based on the vegetative cycle of the crops, allowing its transferability to contexts characterised by different crop calendars or climate conditions. Finally, according to our knowledge, the model represents the first tool for the estimation of flood damage to crops in the Italian context, and in particular in the Po Plain region.

Further research efforts will be focused on three directions: (i) a better understating of damage mechanisms, (ii) the validation of the model, even for other contexts of implementation and (iii) the extension of the model to the other components of a farm.

**Acknowledgements**

This work has been funded by Fondazione Cariplo, within the project "Flood-IMPAT+: an integrated meso & micro scale procedure to assess territorial flood risk".

We wish to thank the Editor and the Reviewers for their constructive comments, which helped us to improve the quality of the manuscript.

**Author Contributions**

Conceptualization, D.M., A.R.S. and F.B.; Methodology, D.M., A.R.S., F.B. and A.G.; Data Management, D.M., A.R.S. and A.G.; Analysis, D.M., A.R.S. and A.G.; Investigation of the results, D.M., A.R.S. and F.B.; Development of the spreadsheet, A.G., Writing – Original Draft, D.M.; Writing - Review, D.M., A.R.S., and F.B.

**Competing interests**

The authors declare that they have no conflict of interest